# Platinum-induced upregulation of ITGA6 promotes chemoresistance and spreading in ovarian cancer

Alice Gambelli [1,8], Anna Nespolo[1,8], Gian Luca Rampioni Vinciguerra [1,7], Eliana Pivetta[1], Ilenia Pellarin[1], Milena S Nicoloso[1], Chiara Scapin[1], Linda Stefenatti[1], Ilenia Segatto[1], Andrea Favero[1], Sara D'Andrea[1], Maria Teresa Mucignat[1], Michele Bartoletti[2], Emilio Lucia [3], Monica Schiappacassi [1], Paola Spessotto [1], Vincenzo Canzonieri[4,5], Giorgio Giorda[3], Fabio Puglisi[2,6], Andrea Vecchione [7], Barbara Belletti [1], Maura Sonego [1,9] & Gustavo Baldassarre [1,9✉]

## Abstract

**Platinum (PT)-resistant Epithelial Ovarian Cancer (EOC) grows as a metastatic disease, disseminating in the abdomen and pelvis. Very few options are available for PT-resistant EOC patients, and little is known about how the acquisition of PT-resistance mediates the increased spreading capabilities of EOC. Here, using isogenic PT-resistant cells, genetic and pharmacological approaches, and patient-derived models, we report that Integrin α6 (ITGA6) is overexpressed by PT-resistant cells and is necessary to sustain EOC metastatic ability and adhesion-dependent PT-resistance. Using in vitro approaches, we showed that PT induces a positive loop that, by stimulating ITGA6 transcription and secretion, contributes to the formation of a pre-metastatic niche enabling EOC cells to disseminate. At molecular level, ITGA6 engagement regulates the production and availability of insulin-like growth factors (IGFs), over-stimulating the IGF1R pathway and upregulating Snail expression. In vitro data were recapitulated using in vivo models in which the targeting of ITGA6 prevents PT-resistant EOC dissemination and improves PT-activity, supporting ITGA6 as a promising druggable target for EOC patients.**

**Keywords** Ovarian Cancer; Metastasis; Integrin; chemoresistance
**Subject Categories** Cancer; Urogenital System

## Introduction

Despite a high initial response rate, the majority of patients with Epithelial Ovarian Cancer (EOC) experience relapse and develop chemoresistance that eventually results in treatment failure and low survival. Primary advanced and relapsed EOCs grow as a metastatic disease, characterized by peritoneal dissemination and carcinosis mainly affecting abdominal and pelvic organs (Jayson et al, 2014; Lheureux et al, 2019). Identifying the pathways necessary for adhesion and spreading of chemo-resistant EOC cells would be of high clinical significance, allowing to identify and, possibly, overcome mechanisms of resistance to current treatments.

EOC patients are defined chemo-resistant when they progress on platinum (PT)-based chemotherapy during or within 6 months from the end of therapy (Jayson et al, 2014; Lheureux et al, 2019). Whether PT treatment itself or the onset of PT-resistance enhances the ability of EOC cells to adhere to the mesothelium and disseminate, it is still debated. However, recent evidences suggest the latter possibility, showing that PT-resistant isogeneic cells commonly acquired a higher ability to adhere and grow on mesothelial cell layer, in vitro, compared to their parental PT-sensitive counterparts (Sonego et al, 2017). Accordingly, unbiased proteomic analyses of PT-resistant cells demonstrated significant changes in the expression of proteins related to Epithelial and Mesenchymal Transition (EMT), cancer stem cells and cytoskeleton pathways, supporting a more invasive phenotype (Smith and Bhowmick, 2016).

Peritoneal dissemination and carcinosis represent a peculiar types of metastasis, quite unique for EOC cells that, shedding from their site of origin, survive as clusters/spheroids in the abdominal cavity and then root and thrive on the mesothelium of peritoneal organs (Thibault et al, 2014). EOC spheroids exhibit slow proliferation and increased resistance to anoikis and cytotoxic factors compared to single cells (Ahmed et al, 2010), akin to what commonly observed in cancer stem-like cells (CSCs). The ability of EOC cells to form spheroids is an accepted measure of the presence of CSCs within the bulk EOC cell population (Sonego et al, 2019). A relevant contribution to peritoneal dissemination is given by the presence of ascites, which is believed to constitute a microenvironment necessary to form and sustain the EOC cancer

[1]Molecular Oncology Unit, Centro di Riferimento Oncologico di Aviano (CRO) IRCCS, National Cancer Institute, Aviano, PN, Italy. [2]Deparment of Medical Oncology, Centro di Riferimento Oncologico di Aviano (CRO) IRCCS, National Cancer Institute, Aviano, PN, Italy. [3]Gynecological Surgery Unit, Centro di Riferimento Oncologico di Aviano (CRO) IRCCS, National Cancer Institute, Aviano, PN, Italy. [4]Pathology Unit, Centro di Riferimento Oncologico di Aviano (CRO) IRCCS, National Cancer Institute, Aviano, PN, Italy. [5]Department of Medical, Surgical and Health Sciences, University of Trieste, Trieste, TS, Italy. [6]Department of Medicine, University of Udine, Udine, UD, Italy. [7]Department of Clinical and Molecular Medicine, Faculty of Medicine and Psychology, Sant'Andrea Hospital, University of Rome "Sapienza", Rome, Italy. [8]These authors contributed equally as first authors: Alice Gambelli, Anna Nespolo. [9]These authors contributed equally as senior authors: Maura Sonego, Gustavo Baldassarre. ✉E-mail: gbaldassarre@cro.it

stem cell niche (Ahmed et al, 2010). Detached EOC cells and spheroids adhere to mesothelial cells by engaging specific cell-cell and cell-extracellular matrix (ECM) interactions, through the expression and activation of adhesion molecules, including cadherins and integrins (Barbolina et al, 2009).

Integrins, transmembrane glycoproteins composed of α and β subunits, recognize specific ECM proteins or ligands on the surface of neighboring cells. Upon engagement, integrins activate several cellular processes, including cytoskeleton remodeling, cell adhesion, proliferation, survival and invasion (Hynes, 2002). Integrins can also contribute to drug resistance, by inducing adaptive pro-survival response pathways (Seguin et al, 2015). In various context integrins expression has been used to identify cells with a CSC phenotype. Integrin α6 (hereafter ITGA6), also known as CD49f, has been extensively used as a marker to identify adult stem cells, progenitor cells, and CSCs (Krebsbach and Villa-Diaz, 2017). ITGA6 can interact with integrin β1 (ITGB1) or β4 (ITGB4) subunits, playing different roles and functions. Overexpression of α6β1 and α6β4 heterodimers is generally associated with tumor spreading, invasion and metastasis (Taddei et al, 2008; Mercurio et al, 2001).

In this study, we investigated whether and how drug resistance and survival were associated with changes in EOC cell adhesion, as reported in other contexts (Damiano et al, 1999; Huang et al, 2005), by studying the expression pattern of integrins in PT-resistant EOC cells and how integrins modulate the response to PT, the formation of a pre-metastatic niche and the metastatic dissemination.

# Results

## ITGA6 is overexpressed in PT-res cells

Using three different isogenic EOC PT-sensitive (PT-sen) and PT-resistant (PT-res) cellular models, we previously reported that all tested PT-res cells adhered better to the mesothelium, compared with their parental PT-sen counterpart (Sonego et al, 2017). We tested if PT-res pools also displayed a different ability to adhere to ECM, plating them onto different ECM components, such as fibronectin, collagen I and IV, laminins and vitronectin. The different EOC PT-sen cell lines displayed different preferences in the adhesion to these ECM substrates, however, all PT-res pools showed increased ability to adhere to laminins (Fig. EV1A), the main ECM component of basement membranes (Hohenester and Yurchenco, 2013). These data suggested that PT-res cells expressed a different repertoire of integrins (Ahmed et al, 2005; Dhaliwal and Shepherd, 2022). Indeed, we observed that ITGA6, encoding for Integrin α6, was upregulated in all PT-res pools compared to PT-sen isogenic counterpart (Fig. EV1B). These data were in line with the notion that α6β1 and α6β4 heterodimers mediate cell adhesion to laminins (Hynes, 2002) and were then confirmed in all PT-res clones derived from pools (Sonego et al, 2020) (Figs. 1A,B and EV1C,D). Using primary cells collected from EOC patients at diagnosis or at disease recurrence, we then observed higher ITGA6 protein expression in the latter group (Fig. 1C, and Appendix Table S1A). By evaluating ITGA6 expression in matched samples from the same patients obtained pre and post chemotherapy we confirmed an increase of ITGA6 in two out of three available cases (Appendix Fig. EV1E). Although

three cases could represent an anecdotical observation, together the collected results suggest that upregulation of ITGA6 could be associated with the acquisition of PT-resistance (or recurrence) both in vitro and in vivo. All PT-res clones tested showed higher ITGA6 mRNA expression in qRT-PCR analyses, and a protein half-life similar to the one of PT-sen cells as demonstrated by cycloheximide (CHX) treatment and WB analysis (Fig. EV1F,G). These data indicated that ITGA6 overexpression in PT-res cells was mainly due to an increased RNA expression.

Intriguingly, when we sorted PT-sen TOV112D cell subpopulations expressing either high or low levels of ITGA6 (ITGA6$^{HIGH}$/ITGA6$^{LOW}$), we observed that ITGA6$^{HIGH}$ cells were as resistant to cis-platin (CDDP) as the PT-res cell clones and expressed higher ITGA6 mRNA levels (Fig. 1D–F).

## Platinum treatment induces ITGA6 expression activating SP1-mediated transcription

The obtained results suggested the possibility that treatment with PT could select pre-existing EOC resistant subclones for the expression of ITGA6 as a mechanism to survive the pressure of PT. We explored this possibility and first tested if CDDP treatment could impact on ITGA6 transcription. Time course analyses demonstrated that ITGA6 mRNA expression significantly increased after CDDP treatment in a time-dependent manner, in PT-sen but not in PT-res cells (Figs. 1G and EV1H). Similarly, upon treatment with CDDP, ITGA6 mRNA increased of about 30 folds in ITGA6$^{LOW}$ cells and of 1.4 folds in ITGA6$^{HIGH}$ cells supporting that PT induced ITGA6 transcription especially in PT-sen cells (Fig. EV1I).

Next, by cloning the ITGA6 promoter (Lin et al, 1997; Nishida et al, 1997) in a luciferase reporter vector (pLUC$^{ITGA6FL}$, Fig. 2A), we verified that CDDP treatment increased of about 4 folds the activity of ITGA6 promoter in PT-sen cells (Fig. 2B). Since c-Myc and SP1 transcription factors are the main regulators of the promoter activity and the expression of ITGA6 (Gaudreault et al, 2007; Groulx et al, 2018), we co-transfected the pLUC$^{ITGA6FL}$ with c-Myc or SP1, in PT-sen cells. Interestingly, SP1 increased while c-Myc reduced the activity of the ITGA6 promoter (Appendix Figure S2A,B). Pharmacological inhibition of SP1 or c-Myc with Mithramycin (MTA, SP1 inhibitor) or 10058-F4 (c-Myc inhibitor) and SP1 overexpression, demonstrated that basal ITGA6 expression was stimulated by SP1 and inhibited by c-Myc expression/activity in PT-sen cells (Figs. 2C–F and EV2C–F). Accordingly, Chromatin Immunoprecipitation (ChIP) assay showed that SP1, but not c-Myc, was enriched on the ITGA6 promoter and this enrichment was strongly increased after CDDP treatment (Fig. 2G). We also observed that, in PT-sen cells, c-Myc co-precipitated with chromatin-bound SP1 and their binding was sharply reduced upon CDDP treatment (Fig. 2H).

Together, these observations supported the idea that SP1 is the main transcription factor driving ITGA6 expression upon CDDP treatment in PT-sen cells and that its activity could be, at least partially, inhibited by the interaction with c-Myc. To verify this hypothesis, we generated two deletion mutants of the ITGA6 promoter: one lacking c-Myc binding site and the other lacking both SP1 and c-Myc binding sites (pLUC$^{ITGA6Mut1}$ and pLUC$^{ITGA6Mut2}$, Fig. EV2G). In luciferase assays, c-Myc overexpression partially suppressed the activity of pLUC$^{ITGA6Mut1}$, while it had no effect on

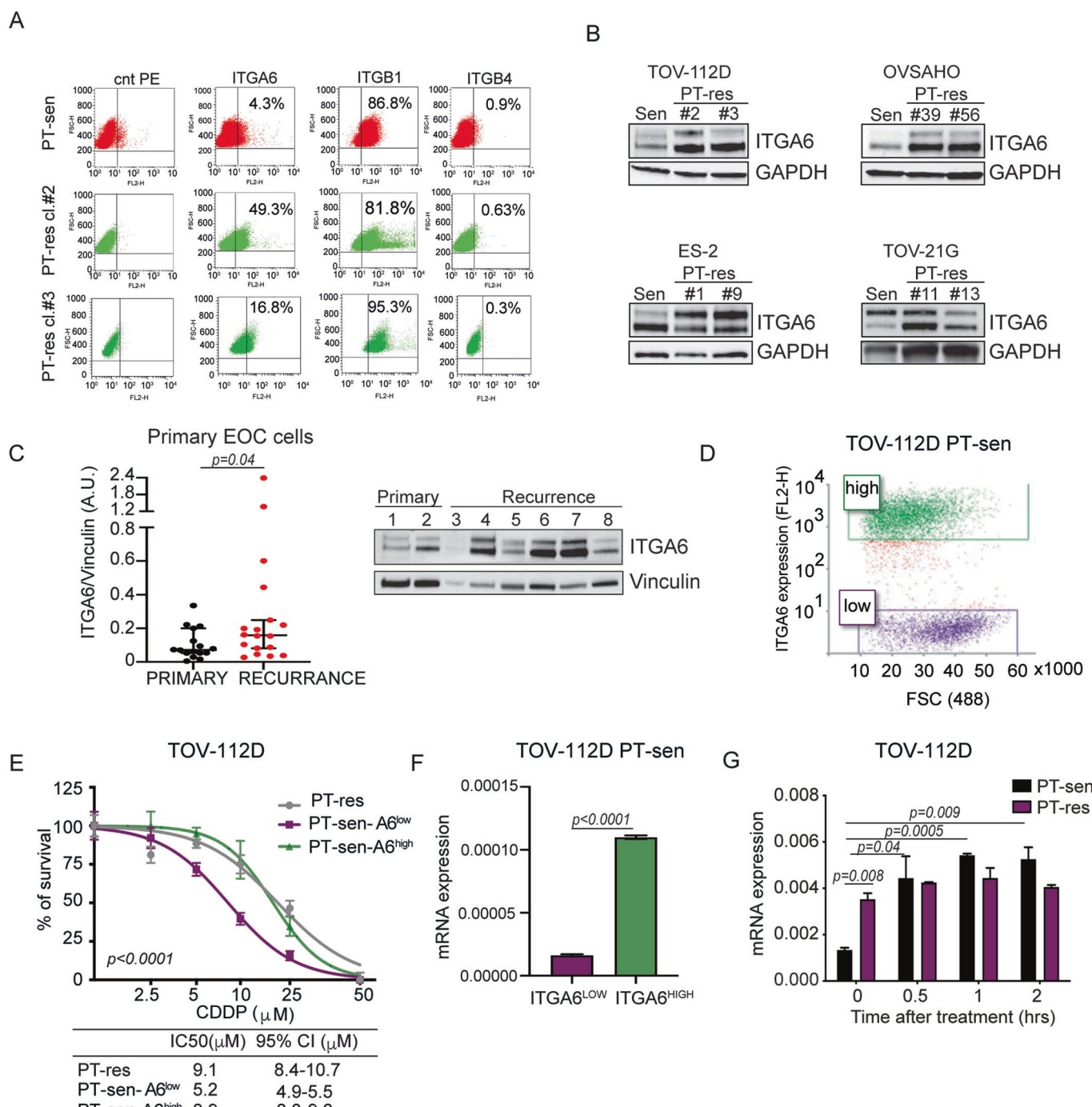

**Figure 1.  ITGA6 is overexpressed in PT-resistant EOC cells and tumors.**

(A) Expression profile of β1 (ITGB1), β4 (ITGB4), and α6 (ITGA6) integrins in TOV-112D PT-sen and PT-res clones analyzed by flow cytometry (FACS). (B) Western blot analysis of ITGA6 expression in the indicated parental and PT-res EOC cells. GAPDH was used as loading control. (C) Graph (left) and representative western blot (right) reporting the expression of ITGA6 protein in total protein extracts of primary cells isolated from EOC patients' ascites collected pre-treatment (primary) or after recurrence (n = 34 from 27 patients). Vinculin was used as loading control. (D) Graph reporting TOV-112D parental cells sorted by FACS for ITGA6$^{HIGH}$ (green) and ITGA6$^{LOW}$ (purple) subpopulations. (E) Non-linear regression analyses of cell viability assay of TOV-112D PT-res, ITGA6$^{HIGH}$, and ITGA6$^{LOW}$ subpopulations (described in D) treated with increasing doses of CDDP for 72 h. Data are expressed as percentage of viable cells with respect to the untreated cells and represent the mean (± SD) of five replicates. The tables below show the IC50 and the Confidence Interval (CI) of each cell type. Fisher's exact test was used to calculate the global p value reported in the graph. (F) Graphs reporting normalized expression mRNA of ITGA6 in ITGA6$^{HIGH}$ and ITGA6$^{LOW}$ subpopulations evaluated by qRT-PCR. (G) Graph reporting the normalized expression of ITGA6 in TOV-112D PT-sen and PT-res cells treated or not with CDDP for 2 h. mRNA was analyzed by qRT-PCR at the indicated time points after CDDP removal. In (F) and (G), mRNA levels were analyzed in triplicate and normalized to actin expression. In (C), (F), and (G), statistical significance was determined by a two-tailed, unpaired Student's t-test (Exact p values were reported on graphs). Bars represent Standard Deviation. Source data are available online for this figure.

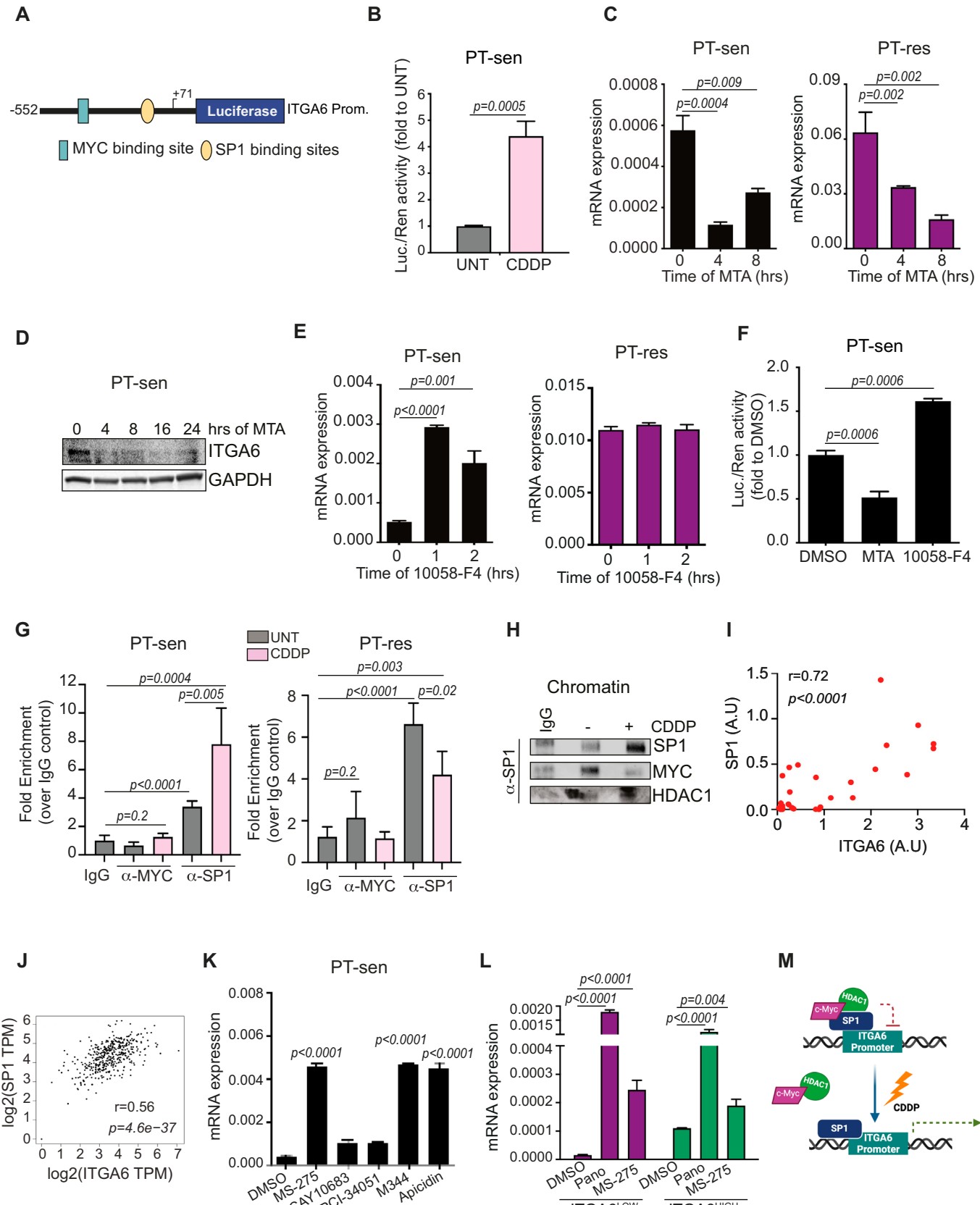

**Figure 2.   PT regulates ITGA6 transcription by modulating SP1 activity.**

(**A**) Schematic representation of ITGA6 Full Length (FL) promoter cloned into pGL3-Luc vector. The c-Myc (blu) and SP1 (yellow) binding sites are indicated. The curved arrow indicates the translation initiation site. (**B**) Graph reporting the normalized luciferase activity measured in TOV-112D PT-sen cells transfected with ITGA6 FL promoter and treated or not (UNT) with CDDP for 24 h. (**C**) Graphs reporting the normalized expression of ITGA6 in TOV-112D PT-sen (left) and PT-res (right) cells treated with the SP1 inhibitor (Mithramycin, MTA) analyzed by qRT-PCR at the indicated time points. (**D**) Western blot analysis of ITGA6 expression in TOV-112D PT-sen cells treated with the SP1 inhibitor, MTA for the indicated time points. GAPDH was used as loading control. (**E**) Graphs reporting the normalized expression of ITGA6 mRNA in TOV-112D PT-sen (left) and PT-res (right) cells treated with the c-Myc inhibitor (10058-F4), analyzed by qRT-PCR at the indicated time points. (**F**) Graph reporting the luciferase activity of ITGA6 FL promoter measured in TOV-112D PT-sen treated with c-Myc inhibitor (10058-F4) or Sp1 inhibitor (MTA) for 24 h. In (**B**) and (**F**), data are expressed as fold value (on untreated in (**B**) and on DMSO in (**F**) of normalized luciferase activity and represent the mean (± SD) of three independent experiments. (**G**) Chromatin immunoprecipitation (ChIP) assay, using either anti-human c-Myc-or anti Sp1-specific antibodies, performed on TOV-112D PT-sen and PT-res cells treated or not with CDDP. Data are expressed as folds enrichment over the IgG, used as negative control and represent the mean of 5 replicates. (**H**) Western blot analysis of SP1-bound proteins in the chromatin fraction of TOV-112D PT-sen cells in the ChIP assay performed as in (**G**). (**I**) Pearson's correlation analysis between ITGA6 and Sp1 protein expression in EOC tumor samples collected in our Institute (see Methods). (**J**) Sperman's correlation analysis between ITGA6 and Sp1 mRNA expression in TCGA ovarian cancer dataset ($n = 489$ samples) using the GEPIA on line tool. (**K**) Graph reporting the normalized ITGA6 mRNA expression in TOV-112D PT-sen cells treated with HDACs inhibitors for 24 h. (MS-275 inhibits preferentially HDAC1; CAY10683 inhibits preferentially HDAC2 and HDAC6; PCI-34051 inhibits preferentially HDAC8, M344 inhibits preferentially HDAC1 and HDAC6; Apicidin inhibits preferentially HDAC1 and HDAC3). (**L**) Graph reporting the ITGA6 mRNA expression in TOV-112D ITGA6^HIGH and ITGA6^LOW subpopulations treated with pan-HDAC inhibitor Panobinostat (Pano) or with the HDAC1 inhibitor MS-275 for 24 h. (**M**) Schematic representation depicting the proposed role of c-Myc/HDAC1/SP1 complex in the regulation of ITGA6 transcription. Under CDDP treatment the repressive c-Myc + HDAC1 complex is displaced leaving SP1 able to activate ITGA6 promoter, inducing its transcription. C-Myc and/or HDAC1 are constitutively displaced from ITGA6 promoter in PT-res cells. In (**C**), (**E**), (**K**), and (**L**), mRNA expression was analyzed in triplicate and normalized to actin. In (**B**), (**C**), (**E**), (**F**), (**G**), (**K**), and (**L**), statistical significance was determined by a two-tailed, unpaired Student's t-test (Exact *p* values were reported on graphs). Bars represent Standard Deviation. Source data are available online for this figure.

the activity of pLUC^ITGA6Mut2, supporting that c-Myc modulated, at least in part, ITGA6 transcription by inhibiting SP1 activity (Fig. EV2H). In line with these findings, the analysis of human EOC samples ($n = 30$, Appendix Table S1B) showed a strong correlation between SP1 and ITGA6 protein expression (r = 0.7 $p < 0.0001$) (Figs. 2I and EV2I). Using the GEPIA on line tool (http://gepia.cancer-pku.cn/index.html) to analyze the TCGA ovarian cancer dataset ($n = 489$ samples), we then confirmed at mRNA level that ITGA6 expression strongly correlated with the one of SP1 (Spearman = 0.56, $p$ $4.6^{-37}$ Pearson = 0.4 $p$ 0) and only weakly with the one of c-Myc (Spearman = 0.18, $p$ $2.1^{-08}$ Pearson = 0.12 $p$ 0.01) (Figs. 2J and EV2J).

In PT-res cells, while SP1 inhibition decreased ITGA6 expression, c-Myc inhibition had no effect (Figs. 2C,E and EV2F). Moreover, SP1 binding to ITGA6 promoter was stably high and slightly reduced by CDDP treatment (Fig. 2G), suggesting that c-Myc inhibitory activity could be impaired in these cells. Since no difference was present in c-Myc and SP1 expression between PT-sen and PT-res cells (Appendix Fig. S2J), we tested if epigenetic modification(s) impacted on their activity. Among nine epigenetic modulator drugs (EMDs) tested, only Panobinostat, a pan-inhibitor of Histone De-Acetylases (HDACs), consistently increased ITGA6 expression in PT-sen models and had no effect on PT-res cells (Fig. EV2L–N). Then, using more specific inhibitors of the different HDAC classes, we observed that only inhibitors of class 1 HDACs (i.e. MS275, M344, and the pan HDAC inhibitor Apicidin) increased ITGA6 expression in PT-sen cells (Fig. 2K). Accordingly, SP1 and HDAC1 readily co-precipitated with c-Myc in PT-sen cells, but this interaction was lost in PT-res ones (Fig. EV2O). Under CDDP treatment, c-Myc expression and interaction with HDAC1 were reduced in PT-sen cells, while the interaction was constitutively low in PT-res cells (Fig. EV2O). Overall, these data indicated that c-Myc, possibly in complex with HDAC1, could act as ITGA6 transcription inhibitor via SP1 regulation, as suggested in other models (Jiang et al, 2007).

Further, HDAC inhibition in ITGA6^LOW cells led to a strong upregulation of ITGA6 mRNA and protein level, while only to a moderate increase of ITGA6 mRNA and protein levels, when

used in ITGA6^HIGH cells (Figs. 2L and EV2P), suggesting that in ITGA6^HIGH cells this layer of regulation was already partially abrogated.

Overall, these data indicated that in PT-sen cells CDDP treatment induced the expression of ITGA6 by activating SP1-mediated transcription, through inhibition of c-Myc/HDAC activity (Fig. 2M). Chronic exposure to CDDP constitutively activated this program in PT-res cells, possibly selecting a pre-existing subpopulation of cells partially insensitive to the HDAC/c-Myc-mediated inhibition of SP1.

## ITGA6 contributes to the acquisition of platinum resistance and stem-like features

Stable upregulation of ITGA6 expression was associated to higher adhesion to ECM and resistance to PT. We then tested whether ITGA6 was necessary for the higher adhesion ability of PT-res cells and whether it played a causal role in the acquisition of the PT-resistant phenotype.

ITGA6 primarily functions as a receptor for laminins (LM), ECM proteins predominantly found in basement membranes (Hohenester and Yurchenco, 2013). Among the different laminin isoforms, LM10 is the most expressed in human EOC (Määttä et al, 2005) and was therefore chosen for subsequent adhesion experiments. Strikingly, adhesion of PT-res cells to LM10 was 10-20 folds higher than the one of PT-sen ones (Fig. 3A; Appendix Fig. S1A). Using a specific anti-ITGA6 blocking antibody (GoH3) or ITGA6 knockout (KO) cells generated by the CRISPR-Cas9 technology, we confirmed that the increased adhesion of PT-res cells to LM10, as well as to the mesothelial cell layer, was primarily mediated by ITGA6 (Fig. 3B–D; Appendix Fig. S1B).

Next, by treating with increasing doses of CDDP PT-sen, PT-res WT and PT-res ITGA6KO cells cultured either on plastic or on a laminin-rich matrices, we observed that PT-sen cells were slightly more sensitive to CDDP when cultured on laminin-rich matrices compared to plastic, while PT-res cells had the same CDDP IC50 in both conditions. Interestingly, when cultured

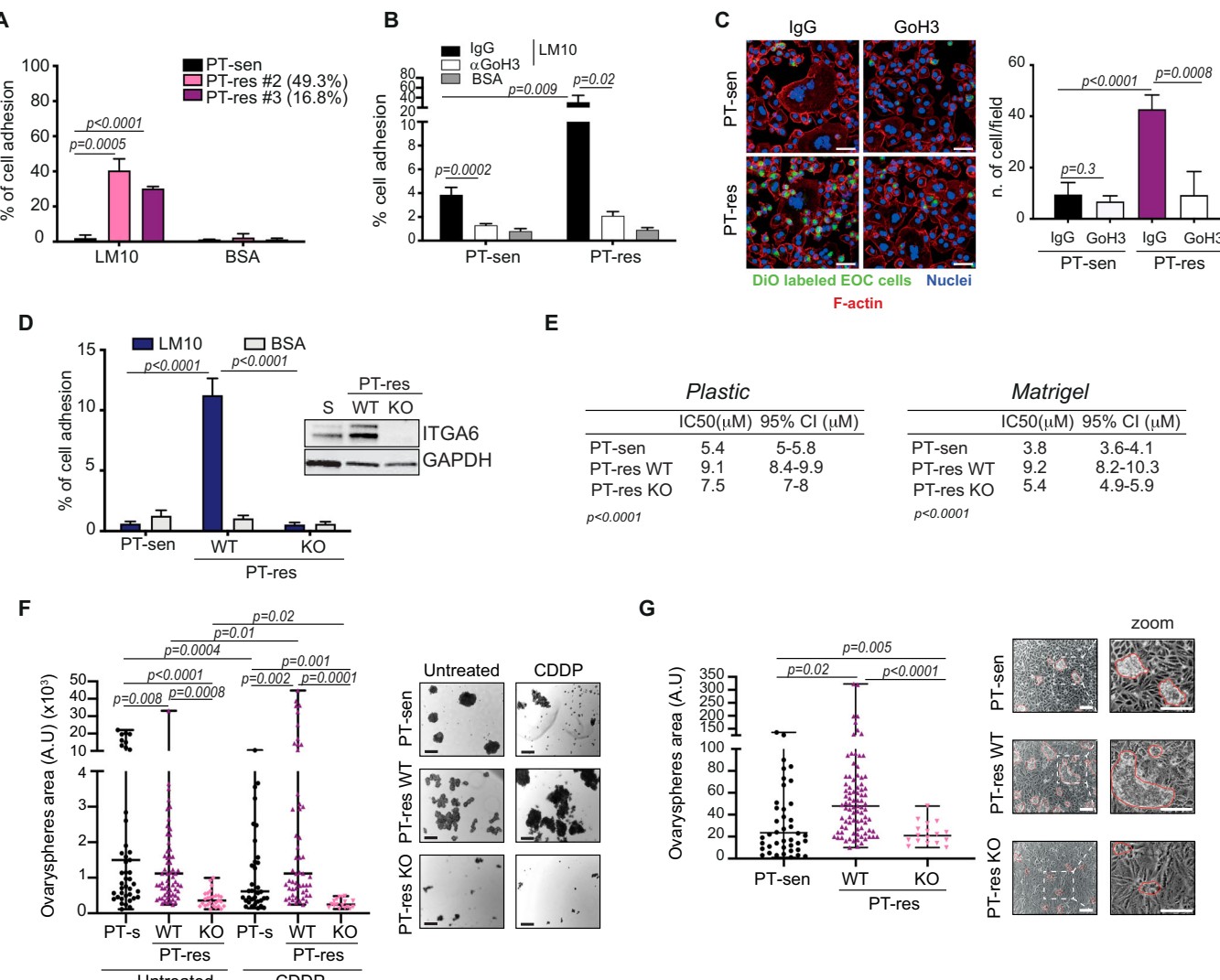

**Figure 3. ITGA6 is responsible of the increased adhesion and invasion of PT-res cells.**

(A) Graph reporting the percentage of PT-sen and PT-res TOV-112D cells adhered on laminin 10 (LM10) coated plates. The percentage of ITGA6 positive cells in each PT-res clone is reported in the legend. (B) Graph reporting the percentage of PT-sen and PT-res TOV-112D cells adhered on LM10 in the presence of IgG (control) or of the specific anti-ITGA6 blocking antibody (GoH3). In (A) and (B), data represent the mean (± SD) of at least 3 replicates and BSA was used as negative control of cell adhesion. (C) Typical images of PT-sen and PT-res TOV-112D cells labeled with the green fluorescent marker DiO (green) and cultured on a monolayer of mesothelial cells for 24 h, in the presence or not of the anti-ITGA6 blocking antibody. Cells were then fixed and stained with Phalloidin (F-Actin, red) and TO-PRO3 (nuclei, blue). Scale bars, 50 μm. The right graph, reports the number (mean ± SD) of cancer cells/field of four independent evaluations. (D) PT-sen, PT-res ITGA6WT, and PT-res ITGA6KO TOV-112D cells adhered on LM10. Inset: western blot analysis reporting ITGA6 expression in the used cells. Data are the mean (± SD) of 2 independent experiments performed in triplicates. BSA was used as negative control for adhesion. (E) Tables reporting the IC50 and the confidence interval (CI) (*n* = 5) of TOV-112D PT-sen, PT-res ITGA6WT and PT-res ITGA6KO cells plated on plastic or in 0.5% Matrigel and treated with increasing doses of CDDP for 72 h. Fisher's exact test was used to calculate the global *p* value reported under the tables. (F) Graph (left) and representative phase-contrast images (right 20X objective) reporting the area of ovaryspheres formed by PT-sen, PT-res ITGA6WT and PT-res ITGA6KO cells treated or not with CDDP (5μM) for 24 h. (G) Graph (left) and representative phase-contrast images (right) reporting the area of ovaryspheres formed by PT-sen, PT-res ITGA6WT and PT-res ITGA6KO cells plated on a mesothelial cell monolayer. White dashed boxes in phase-contrast images highlight the areas magnified in the right panels. In (F) and (G), data represent the median (± SD) of two independent experiments performed in triplicate in which at least 5 randomly selected fields were analyzed. Scale bars, 50 μm. In (A–D) and (F, G). statistical significance was determined by a two-tailed, unpaired Student's t-test (Exact *p* values were reported on graphs). Error bars represent Standard Deviation. Source data are available online for this figure.

on plastic, PT-res ITGA6KO cells had a CDDP IC50 similar to the one of PT-res WT cells, while, when cultured on laminin-rich matrices, their CDDP IC50 was similar to the one of PT-sen cells (Fig. 3E; Appendix Fig. S1C), suggesting that adhesion-dependent resistance to CDDP was mainly due to the expression of ITGA6.

ITGA6 expression has been linked to the acquisition of a cancer stem-cell like (CSCs) status and the overexpression of ITGA6, together with other CSC-associated genes, was observed in PT-resistant EOC models in vivo (Bigoni-Ordóñez et al, 2019; Ricci et al, 2017). Using the ovarysphere formation assay as readout of CSC-like properties of EOC cells (Sonego et al, 2019), we observed

that the spheres formed by PT-res remained resistant to CDDP, as evaluated looking at their number and size, while those formed by PT-sen were still sensitive to CDDP treatment (Fig. 3F, Appendix Fig. S1D,E). The ovarysphere forming ability of PT-Res cells was significantly impaired in ITGA6KO PT-res cells and when an anti-ITGA6 blocking ab was used (Fig. 3F, Appendix Fig. S1D–F). Similarly, ITGA6$^{HIGH}$ subpopulation sorted from parental PT-sen TOV112D cells formed more and bigger sphere when compared to the ITGA6$^{LOW}$ subpopulation (Appendix Fig. S1G). Moreover, spheroids formed by PT-res cells attached and spread better onto a mesothelial cell layer, compared to both PT-sen and PT-res ITGA6KO cells (Fig. 3G). Notably, spheroids from ITGA6KO cells were composed of few cells, unable to invade the mesothelial cell layer, pointing to ITGA6 as a key molecule for this invasive ability acquired by PT-res cells (Fig. 3G, right images). Finally, in three-dimensional (3D) Matrigel evasion assay that PT-sen cells had no (TOV-112D) or very little (OVSAHO) evasion abilities, while PT-res clones efficiently evaded from Matrigel only if ITGA6 was expressed (Appendix Fig. S1H).

## ITGA6 engagement activates Src signaling and Snail expression

The observation that ITGA6 is necessary for adhesion-dependent CDDP resistance, spheroid formation, adhesion and invasion capabilities of PT-res cells, suggests that, upon its engagement and activation, ITGA6 may modulate the epithelial-mesenchymal transition (EMT). To test this possibility, we evaluated the expression of EMT-inducing transcription factors (EMT-TF), essential for the acquisition of CSC-like phenotype (An et al, 2021), in PT-res cells. Upon adhesion to LM10, only Snail protein increased within 1 h of adhesion, in a ITGA6-dependent manner (Fig. EV3A,B). Reintroduction of ITGA6 isoforms A or B similarly restored adhesion-induced Snail protein expression in ITGA6KO PT-res cells (Fig. EV3C). To verify if Snail expression/upregulation could mediate PT-res cells' adhesion and invasion capabilities, we silenced it in TOV-112D PT-res cells and challenged silenced and control cells' ability to form spheres and evade. Consistently, Snail silenced cells formed smaller spheres and had lesser evasion abilities respect to control silenced cells (Fig. EV3D,E).

Adhesion-dependent Snail upregulation was not paralleled by increased mRNA levels, while blocking protein synthesis or proteasome-dependent protein degradation indicated that Snail protein expression was mainly regulated at post-transcriptional level (Fig. EV3F–H). Snail protein proteasomal degradation is mainly regulated by phosphorylation by Glycogen Synthase Kinase 3 beta (GSK-3β) (Zhou et al, 2004). However, using LiCl treatment to inhibit GSK-3β and pSer9 as readout of its activity (Zheng et al, 2013; Bachelder et al, 2005), we excluded that GSK-3β was involved in the regulation of Snail expression upon cell adhesion to LM10 (Fig. EV3I). Therefore, we unbiasedly searched for a kinase that could be responsible for Snail phosphorylation-dependent proteasomal degradation. Looking at the activation/inhibition of 71 kinases (Appendix Table S2) we identified Spleen Associated Tyrosine Kinase (SYK), Focal adhesion kinase 1 (FAK1) and Lck/Yes Novel tyrosine kinase (Lyn) as the most differentially phosphorylated kinases upon adhesion between WT

and ITGA6KO PT-res cells (Appendix Fig. S1I). We focused on Lyn, a member of the Src family, since ITGA6 depletion alters Lyn phosphorylation in acute lymphoblastic leukemia (ALL), increasing sensitivity to chemotherapy (Gang et al, 2020) and since Lyn could regulate Snail and Slug protein stability in breast and prostate cancer cells (Thaper et al, 2017). Indeed, we observed that, upon adhesion to LM10, phosphorylation at Y508 (inhibitory) was increased in PT-res ITGA6KO cells, while phosphorylation at Y397 (stimulatory) was not affected (Appendix Fig. S1J). In addition, ITGA6-mediated adhesion resulted in Lyn and, more generally, Src family members activation that was associated with the stabilization of Snail protein in PT-res cells (Fig. EV3J). Accordingly, treatment with the pan-Src inhibitor Saracatinib largely prevented Snail protein upregulation in PT-res cells adhered to LM10 (Fig. EV3K). Altogether, these experiments suggest that ITGA6 engagement to LM10 in EOC cells increases Snail protein stability through the regulation Src-family kinases leading to the activation of a transcriptional program that drives EMT, invasion and drug resistance (Fig. EV3L).

## ITGA6 secretion by PT-res cells influences the metastatic ability of PT-sen cells

One unresolved question regarding EOC dissemination is whether and how a subpopulation of PT-resistant clones could sustain PT-sen bulk cells in their ability to adhere, grow, and invade the mesothelium and, ultimately resist to chemotherapy. This possibility was supported by the observation that PT-res clones with varying percentage of ITGA6 positive cells showed similar ability to adhere to LM10 (Fig. 3A; Appendix Fig. S1A). It is known that integrins can be secreted and act as paracrine and/or endocrine signals to promote tumor progression (Dhaliwal and Shepherd, 2022; Li et al, 2020). Indeed, we observed that ITGA6 protein was expressed in the conditioned medium (CM) of our PT-res cells (Fig. 4A). Moreover, while CDDP treatment increased ITGA6 mRNA expression in PT-Sen cells (Figs. 1G and EV1G), this increase was not paralleled by an augment of ITGA6 protein levels over a 24 h period of treatment (Appendix Fig. S2A), again suggesting that it could be secreted. Accordingly, we observed that CDDP treatment induced the secretion of ITGA6 and CD63 (extracellular vesicles marker) in a time-dependent manner, peaking at 16 and 24 h of CDDP treatment, in PT-sen and PT-res cells, respectively (Appendix Fig. S2B). To verify if ITGA6 was secreted as soluble protein in extracellular vesicles we isolated exosomes from the conditioned medium of PT-res cells and observed that ITGA6 expression was mostly restricted to the exosome fraction and exosome isolated from CDDP-treated cells express ITGA6 (Fig. 4B,C). PT-sen cells incubated with exosomes isolated from WT PT-res cells formed more and bigger spheres respect to PT-sens cells incubated with exosomes isolated from ITGAKO PT-res cells (Fig. 4D; Appendix Fig. S2C), suggesting that secreted ITGA6 was biologically active.

Based on these results, PT-sen cells were incubated with the CM of WT or ITGAKO PT-res cells for 16 h prior to challenge their ability to adhere on the mesothelial monolayer, form spheroids and activate specific intracellular pathways (Appendix Fig. S2D). PT-sen cells treated with WT PT-res cell CM, formed bigger and more

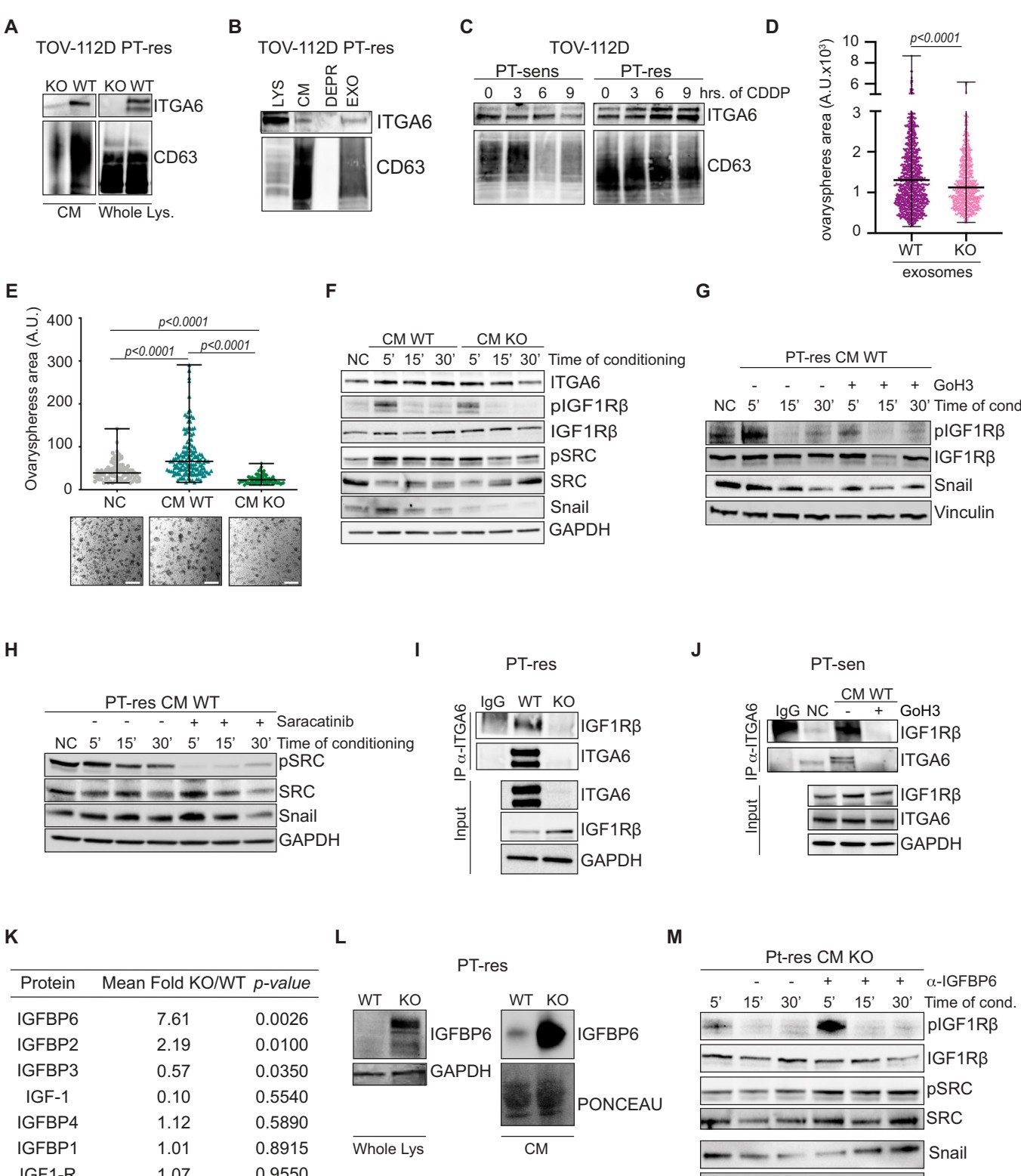

adherent spheroids compared to cells left untreated or conditioned with ITGA6KO PT-res cell CM (Fig. 4E; Appendix Fig. S2E), indicating that PT-res cells could modify the ability of PT-sen cells to adhere and grow on the mesothelium, at least in part, by secreting ITGA6.

## ITGA6 activates IGF1R pathway to promote cancer cell dissemination

We next explored if the stimulation with CM from WT or ITGA6KO PT-res cells differentially activated intracellular signaling pathways in

◄

**Figure 4.   Secreted ITGA6 primes PT-sen cells to adhere and grow on mesothelial cells, by modulating IGF1R pathway.**

(A) Western blot analysis evaluating the expression of ITGA6 and CD63 (marker of extracellular vesicles) in conditioned medium (CM) and whole cell lysates of TOV-112D PT-res cells treated with CDDP for the indicated time points. (B) Western blot analysis to compare ITGA6 and CD63 expression in whole lysates (LYS), CM, exosomes (EXO) and exosomes-depleted CM (DEPR). (C) Western blot analysis of ITGA6 and CD63 in exosomes isolated from CM of the indicated cells treated with CDDP for 3, 6, and 9 h. (D, E) Graphs reporting the area of ovaryspheres formed by TOV-112D parental cells primed with exosomes (D) or CM (E) of TOV-112D PT-res ITGA6WT and KO cells and then plated on a mesothelial cells monolayer (NC = Not Conditioned). In (E), representative phase-contrast images were also reported under the graph. Scale bars, 50 μm. In (D) and (E), data represent the median (± SD) of three independent experiments performed in triplicate in which at least 70 randomly selected cells were analyzed. Statistical significance was determined by a two-tailed, unpaired Student's t-test (Exact *p* values were reported on graphs). (F) Western blot analysis evaluating the expression of the indicated proteins in whole cell lysates of TOV-112D PT-sen cells challenged for the indicated time points with the CM from TOV-112D PT-res ITGA6WT or ITGA6KO cells. NC = Not Conditioned. (G) Western blot analysis evaluating the expression of the indicated proteins in TOV-112D PT-sen cells incubated or not (NC) for the indicated times with the CM from TOV-112D PT-res WT in the presence or not of the anti-ITGA6 blocking antibody, GoH3. (H) Western blot analysis evaluating the expression of the indicated proteins in TOV-112D PT-sen cells incubated or not (NC) for the indicated times with the CM from TOV-112D PT-res WT cells in the presence or not of the Src inhibitor, Saracatinib. (I) Co-immunoprecipitation (Co-IP) analysis of ITGA6 and IGF1Rβ receptor in TOV-112D PT-res ITGA6WT or ITGA6KO cells. (J) Co-IP analysis of ITGA6 and IGF1Rβ receptor in TOV-112D parental cells incubated or not (NC) with the CM from TOV-112D PT-res WT in the presence or not of the anti-ITGA6 blocking antibody, GoH3. In (H) and (I), input shows the expression of the indicated proteins in the lysates used for the IP experiments; IgG represents the control IP using an unrelated antibody. (K) Table reporting the expression of the 7 proteins related to IGF1R pathway present in the used cytokine array. The mean fold (*n* = 3 replicates) reports the ratio between the expression of each cytokine in the CM of PT-res ITGA6KO compared to that in the CM of PT-res ITGA6WT cells. Statistical significance was determined by a two-tailed, unpaired Student's t-test (Exact *p* values were reported on graphs). (L) Western blot analysis evaluating the expression of IGFBP6 in CM and whole lysates of TOV-112D PT-res ITGA6WT or ITGA6KO cells. (M) Western blot analysis of the indicated proteins in cell lysates of TOV-112D PT-sen cells incubated or not (NC) for the indicated times with the CM of TOV-112D PT-res ITGA6 KO cells in presence or not of the specific anti-IGFBP6 blocking antibody. In the figure GAPDH was used as loading control. Source data are available online for this figure.

recipient PT-sen cells. Among different tested signaling pathways, the CM from WT PT-res cells better induced the activating phosphorylation of the beta subunit of the Insulin-like Growth Factors 1 Receptor (IGF1Rβ) compared to the CM from ITGA6KO, PT-res cells. Similarly, activation of Src tyrosine kinases phosphorylation was more and longer expressed after stimulation with CM from WT respect to CM from ITGA6KO PT-res cell. The activation of IGF1R/Src then resulted in increased Snail expression over the time, which could explain the higher adhesion and invasion ability of cells stimulated with CM from WT PT-res cells (Fig. 4F; Appendix Fig. S3A).

A possible reciprocal regulation between IGF1R and ITGA6, has been proposed by others (Basu et al, 2014; Fujita et al, 2012). Accordingly, the use of the anti-ITGA6 blocking antibody GoH3, to block secreted ITGA6 in WT PT-res CM, confirmed that secreted ITGA6 contributed to the activation of IGF1Rβ and to the induction of Snail expression in recipient PT-sen cells (Fig. 4G). The increase in Snail expression was attributed to a post-transcriptional regulation, since its mRNA expression was not affected by ITGA6 presence/activity in the CMs (Appendix Fig. S3B). Conversely, Src inhibition reduced Snail protein levels in PT-sen cells treated with WT PT-res cell CM (Fig. 4H).

It has been reported that full activation of IGF1R intracellular signaling is achieved only when IGF1R forms a trimeric complex with one of its ligands and integrins, particularly IGF2 and ITGA6 (Basu et al, 2014; Cedano Prieto et al, 2017). We observed a readily detectable constitutive binding between ITGA6 and IGF1R in PT-res cells, while this interaction was only slightly appreciable in PT-sen cells. However, in the latter cells ITGA6-IGF1R binding significantly increased upon stimulation with PT-res CM and was prevented by pre-incubation with GoH3 (Fig. 4I,J).

These data were further supported by an unbiased high throughput proteomic screening, evaluating the expression of 174 cytokines in the CM produced by WT or ITGA6KO PT-res cells, 11 of which were differentially expressed (*p* < 0.05, Appendix Table S3). Notably, three of these 11 factors were related to IGF1R pathway, namely the Insulin-like Growth Factor Binding Protein-6 (IGFBP6), IGFBP2 and IGFBP3 (Fig. 4K; Appendix Fig. S3C). The most significantly increased cytokine in the CM of ITGA6KO

PT-res cells was IGFBP6, an O-linked glycoprotein that binds IGF1 and, predominantly, IGF2, reducing their bioavailability, thereby dampening IGF1R pathway activation (Bach, 2015). IGFBP6 protein was overexpressed in both CM and lysate of ITGA6KO PT-res cells (Fig. 4L). We also observed that in different TOV-112D ITGA6KO PT-res clones the IGFBP6 mRNA was increased compared to the expression observed in TOV-112D WT PT-res cells (Appendix Fig. S3D). Further investigation into post-transcriptional mechanisms possibly contributing to overexpression of IGFBP6 protein in ITGA6KO PT-res cells revealed that it could be actively degraded by lysosome since blocking of lysosome-mediated degradation by bafilomycin, but not proteasomal degradation by MG132, resulted in the accumulation of IGFBP6 in both WT and ITGA6KO KO cells. Of note, ITGA6 also appeared to undergone to lysosome- rather than proteosome-mediated degradation in WT PT-res cells (Appendix Fig. S3E,F).

Importantly, phosphorylation of IGF1R and Src and expression of Snail were effectively rescued in PT-sen cells when IGFBP6 was blocked in ITGA6KO PT-res cell CM (Fig. 4M), indicating that IGFBP6 might represent a critical mediator of EMT in this context.

## ITGA6-IGF1R signaling axis promotes EOC dissemination, acting on both tumor and mesothelial cells

Data collected so far demonstrated that PT-res cells secreted bioactive ITGA6 that could, in turn, sustain the spreading and growth of bulk PT-sen cells, through the activation of IGF1R-Src signaling pathway. These findings prompted us to test if PT-res could also "prime" mesothelial cells and favor the formation of a pre-metastatic niche, as proposed in other cancer models (Hoshino et al, 2015). To this end, we stimulated mesothelial cells with WT or ITGA6KO PT-res cell CM, then plated PT-sen cells, displaying low adhesion ability, on primed mesothelial cells (Fig. 5A). When mesothelial cells were treated with WT PT-res cell CM, PT-sen cells formed spheroids that were bigger and more stably anchored to mesothelial cell monolayer, compared to the ones formed on ITGA6KO PT-res cell-conditioned or untreated mesothelial cells (Fig. 5B).

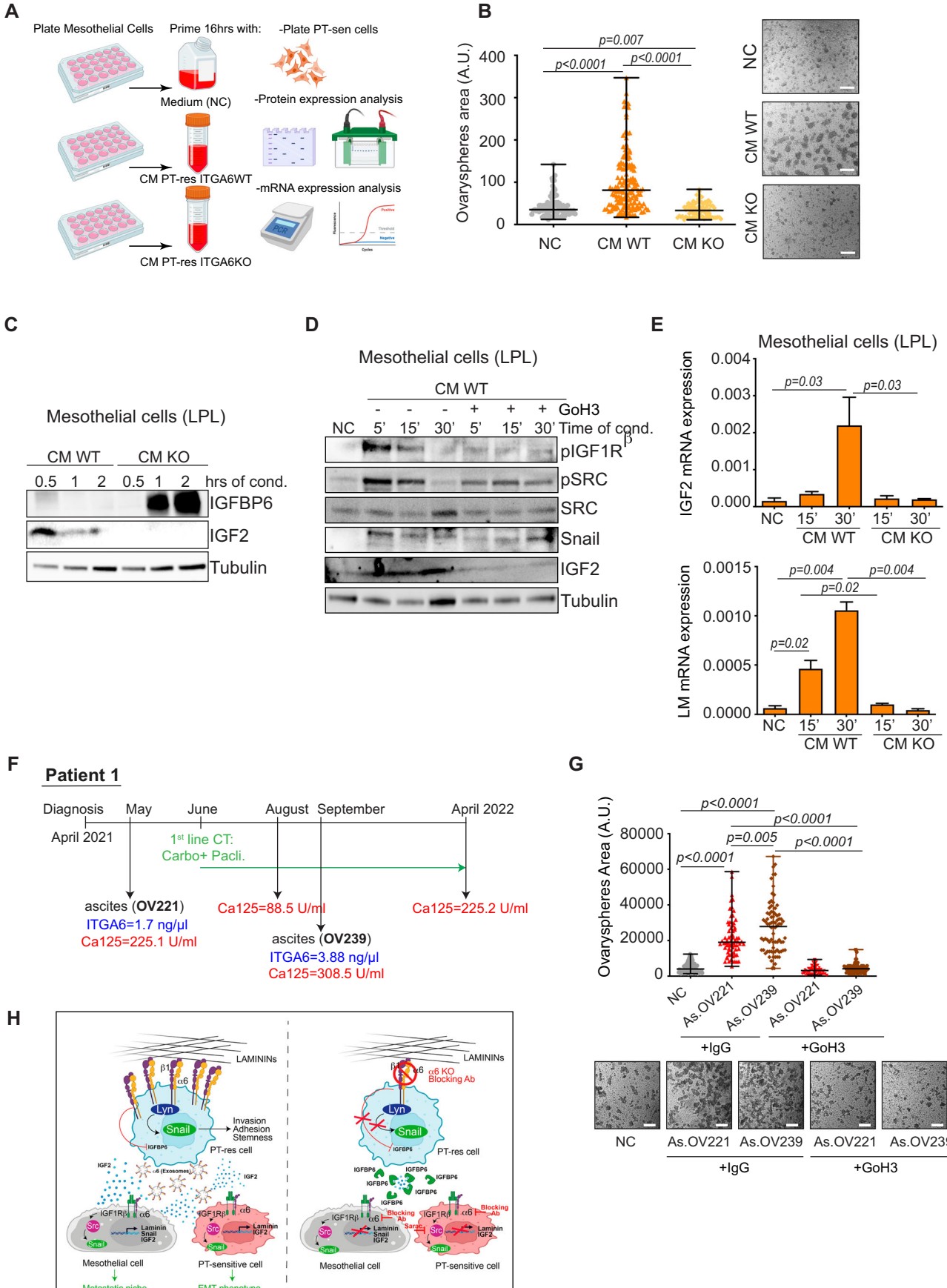

**Figure 5. Secreted ITGA6 primes mesothelial to produce IGF2 and laminins and form a pre-metastatic microenvironment.**

(A) Schematic representation of the experimental procedures used. Mesothelial cells (LPL) cells were incubated or not with CM of TOV-112D PT-res ITGA6WT or ITGA6KO cells for 16 h and then used for adhesion assays with TOV-112D PT-sen cells or for protein and mRNA expression analysis by western blot and qRT-PCR. NC = Not Conditioned. (B) Graph (left) and representative phase-contrast images (right) reporting the area of ovaryspheres formed by TOV-112D PT-sen cells plated on mesothelial cells monolayer primed as described in (A). Data represent the median (± SD) of three independent experiments performed in triplicate in which at least 70 randomly selected cells were analyzed. (C) Western blot analysis evaluating the expression of IGFBP6 and IGF2 in mesothelial whole cell lysates treated as described in (A). (D) Western blot analysis evaluating the expression of the indicated proteins in whole cell lysates of mesothelial cells conditioned or not (NC) for the indicated times with CM of TOV-112D PT-res ITGA6KO cells in presence or not of the specific anti-ITGA6 GoH3 Ab. In (C) and (D), Tubulin was used as loading control. (E) Graph reporting the mRNA expression of IGF2 and LAMA5 (LM) in mesothelial cells incubated for the indicated times as described in (A). NC = Not Conditioned. Data are the mean (± SD) of 3 replicates. (F) Clinical history of patient #1 reporting the timeline of chemotherapy treatments and ascites collections. The amount of CA125 (in blood samples) and ITGA6 (in ascites) were reported in red and blue, respectively. (G) Graph (top) and representative phase-contrast images (bottom) reporting the area of ovaryspheres formed by TOV-112D PT-sen cells plated on mesothelial cells monolayer conditioned or not (NC) for 16 h with ascites samples described in (F), in the presence of the specific anti-ITGA6 blocking antibody GoH3, as indicated. Data represent the median (± SD) of three independent experiments performed in triplicate in which at least 70 randomly selected cells were analyzed. (H) Schematic representation depicting the role of ITGA6 in inducing a PT-resistant phenotype (increased invasion, adhesion, and stemness) both at cell-autonomous and non-cell-autonomous levels, leading to the formation of the pre-metastatic niche. Overexpression of ITGA6 allows a better adhesion of PT-res cells to the mesothelium. ITGA6 engagement induces stabilization of Snail and inhibition of IGFBP6 expression that promote invasion and stem-like behavior. On the other side secreted ITGA6 acts on PT-sen and mesothelial cells to activate the IGF1R pathway and amplify the pro-metastatic signals. ITGA6 KO, or its inhibition with blocking antibodies, prevents the activation of pro-metastatic metastatic pathways both in PT-res cells and in recipient cells and impairs IGF1R signaling by releasing the inhibition on IGFBP6 transcription leading to its overproduction. In (B) and (G), data represent the mean (± SD) of two independent experiments performed in triplicate in which at least 10 randomly selected fields were analyzed. Scale bars, 50 μm. In the figure statistical significance was determined by a two-tailed, unpaired Student's t-test (Exact p values were reported on graphs). Source data are available online for this figure.

Mesothelial cells expressed ITGA6, IGF1R β, and Src family members (Appendix Fig. S4A). Upon stimulation with CM from WT PT-res cells, they expressed IGF2 while stimulation with CM from ITGA6KO PT-res cells resulted higher levels of IGFB6 (Fig. 5C). These data supported the possibility that the IGF1R β -ITGA6 pathway activated in PT-sen by PT-res CM could also be activated in mesothelial cells. Indeed, by priming mesothelial cells with WT PT-res cells CM, we observed an increased phosphorylation of IGFR1 and Src and increased expression of Snail and IGF2 (Fig. 5D). On the other side, using an anti-IGFBP6 blocking antibody in ITGA6KO PT-res cells CM, we observed an anticipated increase of IGFR1β phosphorylation at 5 and 15 min after stimulation, and an increase in Src activation and Snail and IGF2 expression, also in mesothelial cells (Appendix Fig. S4B).

Interestingly, WT PT-res cell CM induced the transcription of both IGF2 and laminins not only in mesothelial but also in PT-sen cells. Their increased transcription was reverted by the use of GoH3 and not observed when ITGA6 was not expressed (Fig. 5E; Fig. S4C), in line with the notion that the induction of a transcriptional mesenchymal program in mesothelial cells leads to the production of ECM proteins, such as laminins, and pro-survival growth factors (Winkler et al, 2020).

### ITGA6-IGF1R signaling axis contributes to the formation of a "pre-metastatic niche"

We next evaluated the clinical relevance of our in vitro findings. First, we assessed the expression of ITGA6 in ascites samples, collected from EOC patients (Appendix Table S1A and Methods section). By following up the same patients and by concomitant evaluation of cancer antigen 125 (CA125) levels in patients' serum, we observed that ITGA6 expression and release in the ascites paralleled tumor progression (Fig. 5F). An increased concentration of ITGA6 was detected in the ascites from patients under chemotherapy treatment, suggesting that PT-treatment could stimulate the production/secretion of ITGA6 also in patients (Fig. 5F; Appendix Fig. S4D). More importantly, ITGA6 secreted in

the ascites was biologically active, as demonstrated by the ability to specifically stimulate PT-sen cells to form spheroids, adhere and grow on ascites-stimulated mesothelial cells (Fig. 5G; Appendix Fig. S4E). This primed mesothelium displayed increased mRNA levels of IGF2 and LAMA5 (encoding for the alpha subunit of LM10), which was prevented by treatment with GoH3 (Appendix Fig. S4F). Altogether, our data indicated that ITGA6, secreted by PT-res cells, modifies the transcriptional program of PT-sen and mesothelial cells, enhancing adhesion, invasion, and metastatic growth of EOC, by acting not only on tumor cells but also on the local microenvironment (Fig. 5H).

### ITGA6 mediates tumor growth and invasion in vivo

All our in vitro data strongly supported that ITGA6 is necessary for adhesion-dependent chemoresistance and PT-res cell dissemination. Therefore, we evaluated whether this holds true in vivo by intraperitoneally injecting WT and ITGA6KO PT-res cells in NSG mice (Fig. 6A). All untreated mice injected with WT PT-res cells developed ascites (volume 0.1–0.4 ml), whereas only 2/5 mice injected with ITGA6KO PT-res cell developed ascites (volume 0–0.05 ml), indicating a less aggressive disease. Upon CDDP treatment, WT PT-res cell injected mice developed even higher volumes of ascites (0.4–3 ml), while, again, only 2/6 mice injected with ITGA6KO cells developed small volumes of ascites (0 to 0.15 ml) (Fig. EV4A). Necroscopy analyses confirmed that ITGA6KO PT-res cells formed significantly less and smaller tumor masses compared to the WT controls (Figs. 6B and EV4B,C). As expected, CDDP treatment had only a minor effect on PT-res cells growth and spreading in vivo (Figs. 6B and EV4A–C). Histological analyses confirmed and reinforced the data collected from macroscopic evaluation, also showing a decreased ability of ITGA6KO cells to infiltrate the peritoneal wall and abdominal organs (Fig. 6C,D). Molecular analyses on tumors and ascites confirmed that the lower invasion ability and aggressiveness of ITGA6KO cells was associated with a decreased expression of Snail and an increased expression/secretion of IGFBP6 (Fig. 6E–G). These data demonstrated that PT-res cells are largely insensitive to

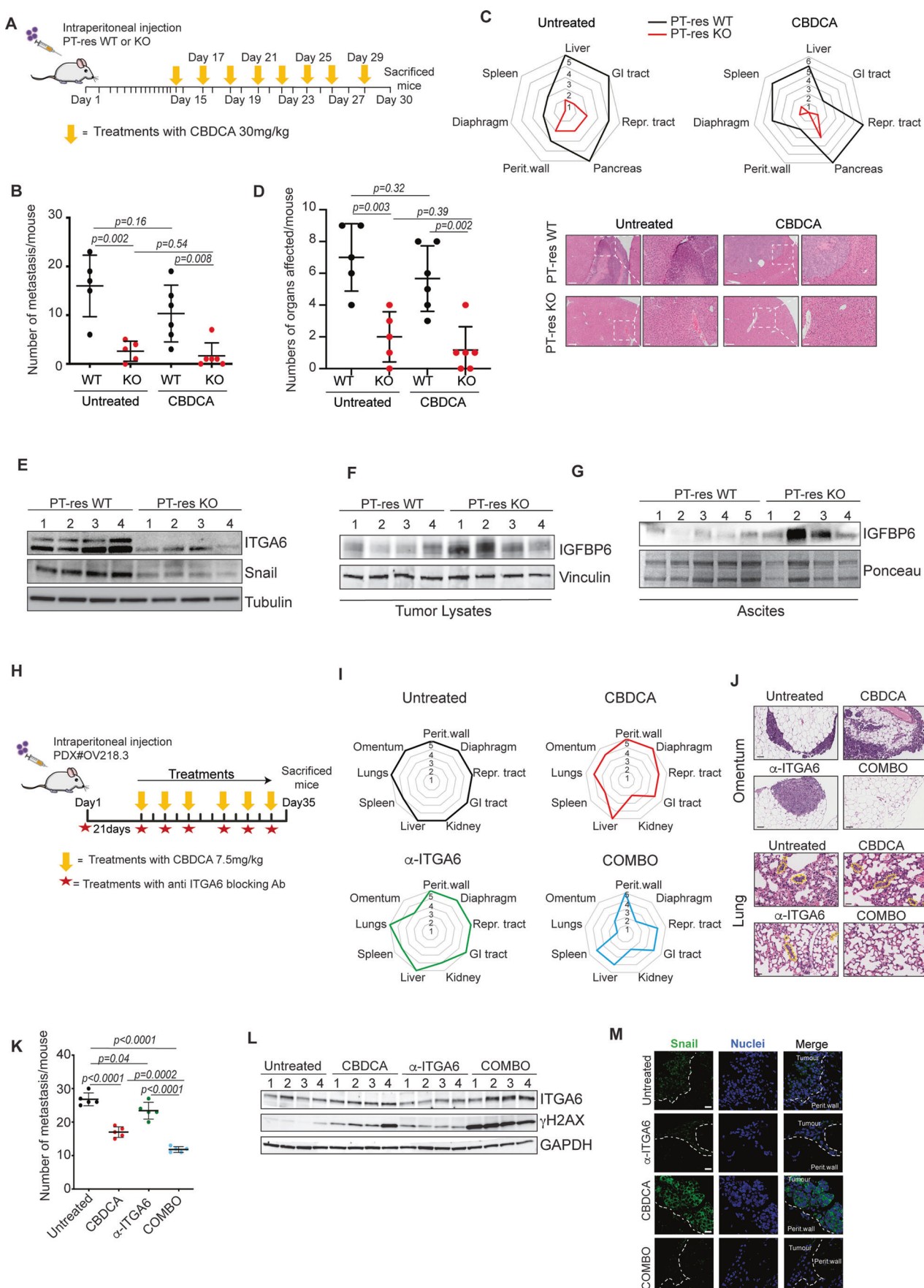

◄ **Figure 6. ITGA6 is a druggable target to block in vivo cell spreading and response to PT.**

(A) Schematic representation of the in vivo experimental procedures. NSG mice injected intraperitoneally (IP) with TOV-112D PT-res ITGA6WT ($n = 11$) or ITGA6KO ($n = 11$) cells, were randomly divided into two group and treated ($n = 6$) or not ($n = 5$) with CBDCA as indicated. Mice were sacrificed 30 days after the injection. (B) Graph reporting total number of tumors in mice described in (A), determined by macroscopic and pathological analyses (total tumor burden). (C, D) Radar (C) and Dot (D) plots reporting the distribution of abdominal metastasis (C) and the total number of organs infiltrated by cancer cells (D) in mice described in (A). In (C), black (WT cells) and red (ITGA6KO cells) lines indicate the number (values 0 to 5 for untreated and 0 to 6 for CBDCA-treated mice) of mice affected for each district, as indicated. Typical images of hematoxylin and eosin (H&E) staining of the showing liver metastasis in mice injected with ITGA6WT PT-res cells and treated as described in (A). 5X (scale bar = 200 μm) and 10X (scale bar = 50 μm) images of the same field are shown. White dashed boxes represent the magnified areas. (E, G) Western blot analysis evaluating the expression of ITGA6 and Snail in tumor masses (E) and IGFBP6 in the tumor masses (F) and ascites (G) of mice described in (A). Tubulin and Ponceau were used as loading controls. Mice 3 and 4 injected with ITGA6KO cells were treated with CBDCA. (H) Schematic representation of the in vivo experimental procedures using EOC PDX model. NSG mice injected IP with PDX OV218.3 ($n = 20$), were randomly divided into four groups and treated or not ($n = 5$) with the specific anti-ITGA6 blocking antibody ($n = 5$), P5G10 ($n = 5$), with CBDCA ($n = 5$) or with the combination of both ($n = 5$), according to the scheme. (I) Radar plots reporting the distribution of abdominal metastasis in mice injected intraperitoneally and treated as in (H), as determined by macroscopic and pathological analyses. Colored bold lines in each plot indicate the number (values 0 to 5) of mice affected for each district. (J) Typical images of H&E analyses of the omentum and lungs of mice described in (H). For each condition, 20X (for omentum, scale bar = 50 μm) and 40X (for lungs, scale bar = 20 μm) images are shown. Yellow dashed lanes in lung images highlighted the tumor metastasis. (K) Graph reporting total number of metastasis/mouse treated as indicated and determined by pathological analyses (L) Western Blot analysis of γH2AX in tumor cells isolated from ascites of mice described in (H). (M) Typical images of Snail (green) expression on peritoneal metastasis collected from mice described in (H) and evaluated by IF analyses (nuclei are in blue). White dashed lines indicate the boundary between tumor masses and the peritoneal wall. Scale bars = 20 μm. In (B), (D), and (K) Statistical significance was determined by a two-tailed, unpaired Student's t-test (Exact $p$ values were reported on graphs). Bars represent Standard Deviation. Source data are available online for this figure.

CDDP also in vivo and rely on ITGA6 to disseminate into the abdominal cavity.

We next asked whether targeting ITGA6 could represent a feasible strategy to overcome PT resistance and prevent EOC dissemination. First we checked if using an anti-ITGA6 blocking antibody (α-ITGA6) in immunocompetent C57/Black6MJ female mice could induce pathological separation of epithelial cells from the laminin-containing basement membrane as suggested by experiment of organ culture of buccal mucosa (Bhol et al, 2001). Mice treated with α-ITGA6 or control IgG, as depicted in Appendix Fig. S5A, were examined macro- and microscopically. No signs of sufferance (e.g. food intake, body weight loss, movements impairments, ascites formation) were observed in control or treated mice. When the detachment of mesothelial cells or epithelial cells of the small and large intestine from their basement membrane was examined by an expert pathologist, we did not observe significant alterations in mice treated with the α-ITGA6 nor in control mice. In particular, no detachment was observed between basal membrane and epithelial cells in any mice and sporadic small detachments were observed in some analyzed peritoneal areas (Appendix Fig. S5B,C).

Once we confirmed the safety of the treatment, we intraperitoneally injected mice with WT PT-res cells and treated them with α-ITGA6 antibody. With only two administrations of α-ITGA6, we observed a reduction in the number of tumor per mouse, a restricted dissemination to the abdominal organs and lower percentage of tumors capable of adhering and infiltrating the peritoneal wall and the abdominal organs (Appendix Fig. S5D–G).

Based on these promising data, we further investigated the activity of α-ITGA6, using a patient-derived xenograft model (PDX#OV215.3) established in our lab by intraperitoneal injection of a high-grade serous ovarian cancer (clinical history in Appendix Fig. S5H). PDX#OV215.3 mice were treated three times with α-ITGA6 and sacrificed after 30 days (Appendix Fig. S5I). We observed a reduced number of tumor masses and diminished tumor dissemination in anti-ITGA6 treated mice, although the difference did not reach statistical significance. Further, treatment with anti-ITGA6 significantly reduced the number of infiltrating tumors, with most of them growing as masses lining on and not infiltrating

the organs abdominal organs (Appendix Fig. S5I–K). These data confirmed, in a PDX model, that ITGA6 plays a central role in promoting EOC invasion and spreading, supporting its possible use as therapeutic agent in combination with chemotherapy, especially in PT-resistant EOC.

To test this possibility, we utilized a second PDX model (OV218.3) established in our lab from an EOC patients relapsed after PT-based chemotherapy during the maintenance treatment with the PARP inhibitor Olaparib (clinical history in Fig. EV4D). This very aggressive PDX forms ascites and macroscopic metastasis within 30-35 days from the injection in NSG mice. NSG mice injected with OV218.3 were treated with intraperitoneal injection of CBDCA and/or α-ITGA6 for two weeks, starting 21 days after PDX injection (Fig. 6H). As expected, untreated mice formed ascites (range 1.5-4 ml/mice), colonized all abdominal and pelvic organs and metastasized to the lung (Figs. 6I and EV4E). Each treatment alone reduced the volume of ascites but only the combination of CBDCA + α-ITGA6 reached the statistical significance. Similarly, the number of tumor spheroids present in the ascites was significantly reduced only in mice treated with CBDCA + α-ITGA6 combination (Fig. EV4E,F). Pathological analyses demonstrated that each treatment reduced the number of metastasis/mice and that CBDCA + α-ITGA6 combination controlled PDX spreading significantly better than the single treatments. Metastasis to omentum and lungs were particularly reduced in combo-treated mice (Fig. 6I±K). Immunofluorescence, western blot, and qRT-PCR analyses were used to verify if α-ITGA6 acted in vivo on the same pathways inhibited in vitro. We confirmed that spheroids formed in vivo by PDX OV218.3 expressed ITGA6 and observed that the used treatments did not significantly impact on its expression, in line with our in vitro observation showing that CDDP treatment has no significant effects on ITGA6 expression in PT-res EOC cells (Fig. 6L). Combination treatment significantly increased the expression of the DNA-damage marker γ-H2AX and of IGFBP6 (Figs. 6L and EV4G). Immunofluorescence analyses showed that Snail expression was consistently upregulated by CBDCA treatment, as we have recently reported (Sonego et al, 2019). Yet, co-administration of α-ITGA6 completely prevented this increase. Single α-ITGA6 treatment also reduced basal expression of Snail again confirming in vitro data (Fig. 6M). Finally, by co-staining peritoneal metastases with anti-ITGA6 and anti-

phosphoIGF1Rβ antibodies we observed clear clusters of active IGF1Rβ and ITGA6 co-localization, especially at the site of tumor-mesothelium contacts. These co-localizations were further increased upon PT-treatment and almost completely abolished when α-ITGA6 was used, mostly for a decreased expression of phospho-IGF1Rβ (Fig. EV4H). Overall, in vivo data collected with PT-res cell lines and PDX models, confirmed the in vitro observations and support the possibility to use anti-ITGA6 treatment(s) as an option to improve PT-based chemotherapy in EOC.

## The expression of ITGA6 targets predicts EOC patients' survival

To understand if ITGA6 expression could also have a prognostic value in EOC patients, we interrogated the KM Plotter dataset available online. In high risk suboptimal debulked advanced patients, high ITGA6 mRNA levels predicted a shorter Progression Free Survival (PFS) [$n = 459$, HR = 1.65, logrank $p = 0.0000021$] and Overall Survival (OS) [$n = 536$, HR = 1.17, logrank $p = 0.15$] (Appendix Fig. S6A and S6B). We also reasoned that the biological activity, more than the expression of ITGA6, might have greater prognostic value. We thus looked at the expression of genes, identified in our study, regulated by ITGA6 engagement, i.e., IGF2 and LAMA5, the alpha subunit of laminin 10. Both high IGF2 or LAMA5 expression predicted shorter PFS in the cohort described above, although the latter did not reach the statistical significance in longrank test (Appendix Fig. S6C and S6D).

Importantly, high LAMA5 and/or IGF2 predicted shorter PFS and OS also in the whole EOC patients' population (Fig. 7A–D). Of note, while IGFBP6, did not have any prognostic value, the expression of the other subunits of laminin 10, LAMA5B1 and LAMA5C1 also predicted poorer patients' prognosis (Appendix Fig. S6E–H).

Together, these data strongly support the preclinical data and indicate that the high expression/activity of ITGA6 predicts worse EOC patients' prognosis.

## Discussion

Gold standard therapeutic approaches for patients with advanced EOC include debulking surgery and PT-based chemotherapy. The vast majority of patients (>70% of the cases) are sensitive to first line PT-based therapy. However, over 75% of stage III-IV EOC patients experience disease recurrence that become resistant to PT, leading to metastatic disease in the abdomen and pelvis (Jayson et al, 2014; Lheureux et al, 2019). No effective therapies are available for the recurrent PT-resistant disease, which still remains largely uncurable.

In our search for molecular mechanisms underlying PT-resistance and mediating the acquisition of increased survival and spreading abilities by EOC cells, we discovered that Integrin α6 (ITGA6) was overexpressed by PT-resistant cells and was necessary to sustain EOC metastatic ability, both in vitro and in vivo. Our data suggest that among PT-sen bulk cells, subclones expressing higher levels of ITGA6 exists and, under the pressure of chemotherapy, could have a growth/survival advantage and therefore positively selected. Of course, this hypothesis should be better experimental validated.

On the other side, we observed that a transcriptional program could be rapidly activated by PT treatment (within 1–2 h) leading to the transcription of ITGA6 in PT-sen cells likely due to epigenetic regulation of its promoter activity. Increased expression of ITGA6 could represent a survival mechanism exploited by PT-sen cells to survive. This possibility is in line with the recent observations that, in breast and colon cancer, the activation of specific transcriptional programs are necessary for cancer cells to survive the pressure of chemotherapy (Lin and Zhu, 2021).

Importantly, our experiments highlighted that targeting ITGA6, using a specific monoclonal blocking Ab, represents a feasible approach and leads to reduced EOC metastatic spreading in different EOC models, including PDXs. It is noteworthy that with this treatment we observed a significant reduction in the growth of infiltrating metastatic lesions, in particular of the diaphragm and lungs, replaced by the presence of small spheroids that were only leaning on the mesothelium of abdominal/pelvic organs. These results have relevant clinical implications, as the presence of infiltrating diaphragmatic lesions poses a major challenge for achieving an optimal cytoreduction in EOC patients (Papadia and Morotti, 2013), while non-infiltrating spheroids are more easily removed during the surgical intervention and more sensitive to chemotherapy. Thus, targeting ITGA6 in combination with standard PT-based chemotherapy could be beneficial both to elicit a better response to PT, and to obtain a better surgical outcome, either during interval surgery or in the surgical treatment of recurrent disease.

Targeting integrins has been tested as anti-cancer therapy in humans. However, the majority of clinical trials investigating the efficacy of integrins-targeting therapeutics in cancer have yielded negative results. Several factors could account for these outcomes, such as a lack of comprehensive understanding of changes in integrin expression during cancer progression in patients, as well as limited knowledge in pharmacological and pharmacodynamic properties of the agents used (Bergonzini et al, 2022). Here, we propose a novel integrin target, ITGA6, which has never been tested in humans and has been specifically investigated based on its expression changes during EOC progression. We also suggest a different route of administration, intraperitoneal injection, which is already utilized in the treatment of advanced EOC patients to deliver chemotherapy alone or in combination with monoclonal antibodies (Kim et al, 2022; Zhang et al, 2023).

Previous observations showed that high ITGA6 protein expression predicted low patients' PFS and OS when analyzed by IHC, but not by gene expression profile (Wei et al, 2019). A low correlation between ITGA6 mRNA and its linearized copy number alteration has been also reported, along with a not significant correlation between ITGA6 mRNA and protein expression (Wu et al, 2020). These data collectively suggest that ITGA6 may hold prognostic value in EOC only when assessed at the protein expression and/or activity level. At this regard, we have identified IGF2 and LAMA5 as two genes transcriptionally regulated by ITGA6 in EOC, and have demonstrated that their high mRNA expression predicts shorter PFS and OS. Investigating whether ITGA6 post-transcriptional modifications (e.g., mRNA splicing, protein glycosylation, and phosphorylation etc.) are also pertinent in determining the stability/activity of the protein and/or its prognostic value will be an interesting avenue for future exploration. Moreover, our data indicate that in the context of PT-res cells, ITGA6 may undergo degradation by lysosomes,

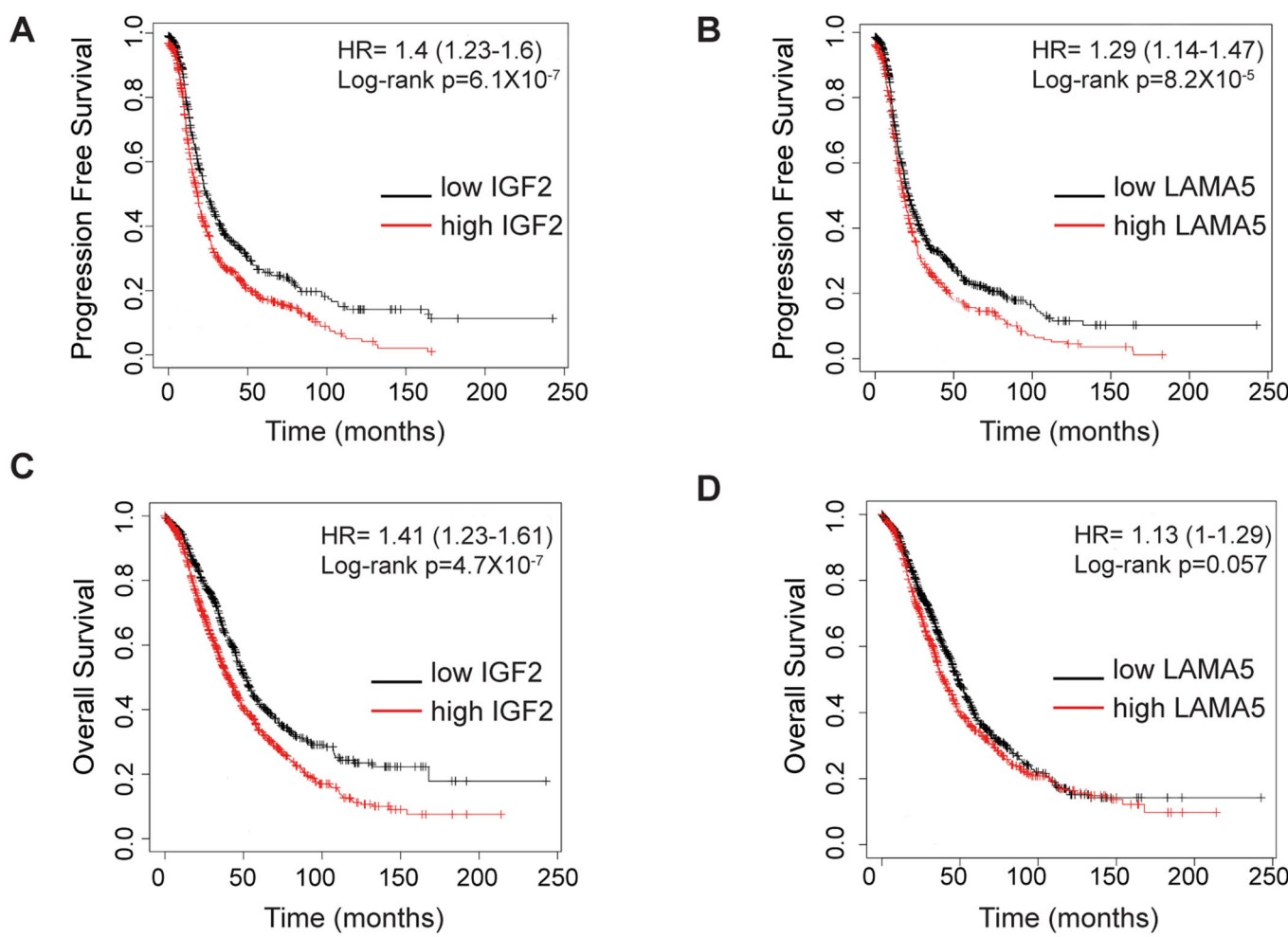

**Figure 7. High ITGA6 expression and activity predict poor prognosis of EOC patients.**

(A–D) Kaplan Meyer estimating the Progression Free (PFS) (n = 1435) (top) or Overall Survival (OS) (n = 1656) (bottom) of EOC patients stratified for IGF2 (A) and LAMA5 (D) using the KM Plotter dataset. HR = Hazard Ratio. The Confidence interval is reported between brackets. The p value was calculated using the logrank test. Source data are available online for this figure.

suggesting a role also for post transcriptional regulation in the contest of recurrent/resistant EOC.

Previous reports have already associated the expression of ITGA6 to a PT-resistant phenotype, in EOC cell and PDX models (Ricci et al, 2017; Wei et al, 2019), independently supporting our observation. However, these studies did not investigate if and how ITGA6 participates in determining the phenotypes of PT-resistance.

In this study, we addressed this gap by uncovering the essential role of ITGA6 in establishing a PT-resistant phenotype, conferring the ability to adhere to the mesothelium and facilitating the formation of spheroids by EOC cells, which subsequently enables their settlement, infiltration, and dissemination. From a molecular point of view, we identified the Src family kinases, as intracellular mediators activated by ITGA6 engagement. By regulating Snail protein phosphorylation and stability, they link the expression of ITGA6 to EMT and the acquisition of a CSC-like phenotype. These findings corroborate our previous demonstration of Snail protein stability regulation as a key molecular switch linking cellular dissemination to the PT response in EOC (Sonego et al, 2019).

Our data also clearly indicate that ITGA6 is secreted as a biologically active protein and its engagement with laminins and/or mesothelial cells induces the formation of a pre-metastatic niche. Recent evidences suggest that bulk *non* metastatic EOC cells can promote metastasis via "hit-and-run" commensal interactions with cells already exhibiting some metastatic ability (Naffar-Abu Amara et al, 2020), supporting the idea that the crosstalk between different clonal EOC populations could foster abdominal/pelvic dissemination. Yet, scant information exists regarding the role of PT and PT-resistance, in this context. A recent report suggests that PT-treatment in EOC cells leads to the release of extracellular vesicles that could induce invasion and increase resistance when up taken by bystander cells. Inhibition of vesicle uptake blocked this adaptive response and sensitized cells to PT (Samuel et al, 2018). These data are in line with our observations, showing that ITGA6 secreted in exosomes, contributes to the metastatic dissemination of bulk PT-sen cells, via the activation of IGF1R pathway. Secreted ITGA6 is then able to activate the IGF1R pathway in recipient cells, serving as a pro-survival and pro-

invasion stimulus. Our results using anti-ITGA6 blocking Ab in vitro and in vivo, suggest that this could be due to the ability of ITGA6 to bind and activate IGF1R-Src pathway, which in turn activates a Snail-dependent transcriptional mechanism which sustains cell survival and invasion. The role of IGFR1 pathway in promoting metastasis and drug resistance is well established, including in EOC, although its targeting has been proved challenging (Fettig and Yee, 2020; Liefers-Visser et al, 2017; Lee et al, 2022). We identified IGFBP6 as a key modulator of ITGA6-IGF1R functional axis and showed that, by modulating IGFBP6 expression/function, it is possible to act on ITGA6 engagement-dependent IGF1R signaling activation. Remarkably, decreased IGFBP6 expression has been observed in the serum of EOC patients compared to healthy donors (Gunawardana et al, 2009). Similarly, the analyses of IGFB6 mRNA expression in normal ovary and EOC samples showed a sharp IGFBP6 reduction in tumor samples, especially metastatic lesions (Piscazzi et al, 2022). Consistent with our data, it has been reported that administration of exogenous IGFBP6 plays a tumor suppressive role in PT-resistant, but not PT-sensitive, EOC cells (Piscazzi et al, 2022) and that IGFBP6 expression is reduced in chemo-resistant glioblastoma (Oliva et al, 2018).

IGFBP6 regulates the availability of IGF2 and, with a lesser extent, of IGF1 (Bach, 2015), explaining at least in part why IGF1R signaling is decreased in ITGA6KO cells. Our data support the possibility that ITGA6 inhibits the expression of IGFBP6, therefore sustaining IGF1R activation. This process could be eventually boosted by EOC exposure to PT, that increases ITGA6 production and secretion.

Very recent evidences demonstrated that PT-resistant ovarian endometrioid A2780 cells disseminate better than their PT-sensitive counterpart, due to an upregulation of IGF1R signaling, resulting in increased ITGA6 expression (Deo et al, 2022). These findings not only support our work, but also raise the possibility that a positive feedback loop, able to sustain EOC spreading and survival, could be established by the ability of ITGA6 to activate, via regulation of IGFBPs, the IGF1R pathway (this work), which, in turn, induces the transcription of ITGA6 (Deo et al, 2022).

Overall, our work describes how chronic exposure to PT influences the acquisition of resistance and metastatic ability by PT-sen cells, both at cell-autonomous and *non*-cell-autonomous levels, leading to the formation of a pre-metastatic niche. We also demonstrated that ITGA6 is a druggable target and identify biomarkers that can predict patients prognosis, thus laying the groundwork for the development of new precision oncology strategies to treat metastatic PT-resistant EOC patients.

# Methods

## Cell lines

TOV-112D (CRL-11731), TOV-21G (CRL-11730), and ES-2 (CRL-1978) cells were obtained from the American Type Culture Collection (ATCC), OVSAHO (JCRB1046) cells was from the JCRB Cell Bank. Cisplatin-resistant (PT-res) isogenic cells were generated as described (Sonego et al, 2017, 2019, 2020). Briefly, EOC parental cells were treated for 2 h with a CDDP dose 10-fold

higher than the calculated IC50 and then allowed to re-grow in drug-free complete medium. In total, PT-res cells received 20 pulse treatments. Mesothelial cells (LPL), kindly provided by Dr. P. Spessotto, were isolated from pleural fluids and were already described in Sonego et al (2017). The isolated cells stained positively for mesothelial markers, such as cytokeratin 5 and Calretinin.

All cell lines were maintained in RPMI 1640 medium (Sigma-Aldrich) supplemented with 10% heat-inactivated fetal bovine serum and 1% penicillin/streptomycin. All cell lines were grown in standard conditions at 37 °C and 5% $CO_2$ and were routinely authenticated in our laboratory using the Cell ID TM System (Promega) protocol and using GeneMapper ID version 3.2.1 to identify DNA short tandem repeat profiles. Mycoplasma contamination was assessed every 15 days using the MycoAlert test (Lonza).

## Mice

Female NOD scid gamma mice (NSG) (4 weeks old) were acquired from Charles River Laboratories and housed in the CRO-Aviano animal facility in cages trolleys, inside cages (up to 4 mice/cage) complete with all necessary accessories in a 12 h light:12 h dark cycle according to the 2010/63/EU regulations. Well-being and the health of the animals were checked daily by dedicated personnel of the facility. Mice were xenografted with $2 \times 10^6$ TOV-112D PT-res ITGA6 WT or KO cells intraperitoneally in sterile phosphate-buffered saline. After 15 days from injection (in accordance with our previous observations of intraperitoneal injection of the same cells), animals were randomly divided into two groups and treated or not intraperitoneally with CBDCA (30 mg/kg) three times for weeks for 2 weeks. For experiment with blocking antibody, in accordance with experimental procedure described above, we injected NSG mice (4 weeks old) with $2 \times 10^6$ TOV-112D PT-res ITGA6 WT. The anti-ITGA6 blocking antibody P5G10 clone was intraperitoneally injected at the final concentration of 30 mg/kg at the injection time and according to the procedure schemes reported in the figures. Patient-derived xenograft (PDX) tumor was generated by injecting primary cells from High Grade Serous Ovarian Cancer (HGSOC) patient's ascite (OV215 and OV218). For PDX OV215.3, NSG mice were injected intraperitoneally with $2 \times 10^6$ of tumor cells and then treated with the anti-ITGA6 blocking antibody P5G10 (30 mg/kg) according to the schemes reported in the Appendix Fig. 5D. To evaluate the effectiveness of the combination between anti-ITGA6 blocking antibody (P5G10) and carboplatin (CBDCA) treatment, NSG mice were intraperitoneally injected with PDX OV218.3 cells. Mice were treated by intraperitoneal injection with P5G10 (30 mg/kg) and CBDCA (7.5 mg/kg), as single agents or in combination according to the scheme reported in Fig. 6H. For all in vivo analysis, dissemination was recorded and annotated by photographs and charted on the basis of organ dissemination for all the described experiments. Ascitic fluids were collected: spheroids were counted and lysed in cold RIPA buffer with protease inhibitors to evaluate the levels of proteins of interest by western blot analysis.

To evaluate the possible basement membrane separation in small and large intestine following anti-ITGA6 blocking antibody (P5G10) treatment C57/Bl6MJ female mice acquired from Charles

River Laboratories were treated twice/week for two weeks with P5G10 (30 mg/kg) and then sacrificed after 48 h from the last injection. Vital well-being signs were collected during the course of the experiment. Tissues were fixed in formalin before dehydration in ethanol and xylene before paraffin embedding and then stained with hematoxylin and eosin and then subjected to pathological analyses by an expert pathologist in blind. The H&E-stained sections were digitalized using an Aperio ScanScope CS2 (Leica Biosystems) and analyzed with Aperio ImageScope software (Leica Biosystems). In each case, three different tracts for both small and large bowel (each tract = 1 mm) were considered, and the detachment of intestinal epithelium and mesothelium from their respective basement membranes were measured and expressed in micrometers.

Formalin-fixed paraffin-embedded tissues' sections were also used for immunofluorescence staining. After deparaffinization, sections were permeabilized with Triton 0.5% for 5 min (for Snail staining) or not permeabilized (for ITGA6 and pIGF1R staining) and then blocked in BSA 1%. Primary antibodies were incubated overnight at 4 °C: anti Snail/Slug (ab18071 1:50) from Abcam, anti-ITGA6 (P5G10, 3 μg/ml) from DSHB hybridoma and anti pIGF1Rβ (#3918, 1:50) from Cell Signalling. Alexa-fluor secondary antibodies were incubated 1 h at room temperature. Coverslips were placed on slides with MOWIOL with DABCO 2.5% and images were acquired through the TCS-SP8 Confocal Systems (Leica Microsystems) interfaced with the Leica Confocal Software (LCS) (version 3.5.5.19976) or the Leica Application Suite (LAS) software (version 6.1.1).

Animal experimentation was reviewed and approved by the Centro di Riferimento Oncologico di Aviano (CRO) Institutional Organism for Animal Wellbeing (OPBA) and by the Italian Ministry of Health (authorizations no. 1261/2015-PR and 753/2021-PR released to G. Baldassarre). All animal experiments were conducted in adherence to international and institutional committees' ethical guidelines.

## Human samples

Biospecimens including tumor samples, patient-derived primary cells, and ascitic fluids were obtained from patients who gave their informed consent, under protocols approved on 10.07.2019 by the Ethics Committee (OutCoME protocol - CRO-2019-53 approval CEUR 2019-Sper-084). The experiments were conformed to the principles set out in the WMA Declaration of Helsinki and the Department of Health Services Belmont Report. Patient data were pseudonymized and annotated in a prospective database.

## Flow cytometry (FACS)

Cells were grown in 100 mm tissue culture plates to 90–95% confluence and harvested with 5 mM EDTA. For measurement of integrin expression, once harvested all samples were maintained at 4 °C to maintain the expression of integrins on the cell surface. Thus, cells were washed and resuspended in 4 °C Tyrode-Hepes Buffer containing 1 mM $CaCl_2$, 1 mM $MgCl_2$, 5.5 mM Glucose, and 1 mg/ml BSA. Cells were incubated with the appropriate primary antibodies (PE anti CD49f, APC anti CD49c, PE anti CD49e; PE anti CD29, PE anti CD104 from BD Pharmingen) for 15 min at 4 °C, washed three times with ice-cold Tyrode-Hepes Buffer and

incubated with PE or Alexa Fluor-488 labeled secondary antibody for another one hour at 4 °C. Cells were washed, resuspended in 0.5 ml of ice-cold Tyrode-Hepes Buffer and kept on ice until analyzed by flow cytometry. Flow cytometry was performed using FACs LSFortessa (BD Bioscience) and data were analyzed using DIVA software (BD Bioscience). Isotype-matched monoclonal antibodies were used as controls.

Separation of ITGA6[High] and ITGA6[Low] subpopulation from TOV-112D PT-sen cells was perfomed by the Cytometry facility of CRO-Aviano, IRCCS using a FACSAriaIII sorter instrument (BD Bioscience) following routine procedures. Briefly, $1 \times 10^7$ TOV-112D PT-sen cells were incubated with the anti-CD49f-FITC conjugated antibody as described above. Resuspend the cells in culture medium and determine the cell concentration using a vital dye such as Trypan blue and adjusted at a concentration of $20 \times 10^6$ cells/ml. Gating of the cells were set using the negative and positive controls to define the populations of interest. Determined gates was selected for sorting into external collection tubes at 4 °C and $2.5 \times 10^6$ cells were sorted into a 15 ml conical tubes.

## Adhesion assay (CAFCA)

The quantitative cell adhesion assay was performed using the CAFCA (Centrifugal Assay for Fluorescence-based Cell Adhesion) methods (Spessotto et al, 2009). Briefly, 6-well strips of flexible polyvinyl chloride were coated with 10 μg/ml Laminin 10 (LM10) (iMatrix-511, ReproCell), or with other indicated substrates: Collagen I (Corning), Collagen IV, Fibronectin and Vitronectin (Sigma-Aldrich), resuspended in calcium carbonate (CaCO3) buffer pH 6.8, for 16 h at 4 °C. EOC parental and PT-res cells were detached with 5 mM EDTA, washed with PBS to remove all EDTA and then labeled with the vital fluorochrome calcein acetoxymethyl (2 μM/$10^6$ cells) (Invitrogen) for 15 min at 37 °C. EOC fluorescently-labeled cells were then collected through a centrifugation step, rinsed twice with cell culture medium and resuspended in culture medium at the concentration of 50,000 cells/well and then dispensed into each well of the coated strips of the bottom CAFCA miniplates. The bottom CAFCA miniplates were then centrifuged for 5 min to synchronize the contact of the cells with the substrate and incubated for 20 min at 37 °C. Subsequently, the bottom CAFCA miniplates were mounted together with a similar CAFCA miniplates (top miniplate) to create communicating chambers for subsequent reverse centrifugation, and centrifuged for 5 min at 1000 rpm. We measured the fluorescence signal emitted by cells in wells of the top (non-bound cells) and bottom (substrate-bound cells) sides of the CAFCA miniplates using the Infinite® M1000 Pro microplate reader (Tecan Group Ltd.) capable of detecting the fluorescence emanated from both the top and bottom side of the microplate. The percentage bound cells, out of the total amount of cells introduced into the system, were calculated as: bottom fluorescence value/bottom fluorescence + top fluorescence values.

## Adhesion on mesothelial cells and Immunofluorescence Analyses

Mesothelial cells (LPL) were seeded on glass coverslips and allowed to grow until they reached complete confluence. EOC parental and PT-res cells ($1.5 \times 10^5$) were detached with 5 mM EDTA and then

treated with anti-ITGA6 blocking antibody (GoH3) or anti-IgG control antibody for 20 min. The cells were then labeled with the vital green fluorescent lipophilic tracer DiO (Invitrogen), washed with PBS and finally plated on mesothelial layer in serum-free medium for 24 h. For immunofluorescence (IF) analyses cells plated on coverslips were fixed in PBS-4% paraformaldehyde (PFA) at room temperature (RT), blocked in PBS-1% bovine serum albumin (BSA). TO-PRO-3 iodide (Invitrogen) were used to visualize nuclei and Alexa-Fluor 647-Phalloidin (Invitrogen) for F-actin staining. Coverslips were mounted with 10% glycerol/0.25% DABCO and analyzed using the TCS-SP8 Confocal Systems (Leica Microsystems Heidelberg GmbH) interfaced with the Leica Application Suite (LAS) software.

## Drugs treatment

TOV-112D ($7 \times 10^3$) PT-sen or PT-res ITGA6 WT or KO cells were plated in 96-well plates coated or not with 2.5 mg/ml of Matrigel (Cultrex) and treated with increasing doses of Cisplatin (CDDP) for 72 h with a minimum of five technical replicates per concentration per cell line. Cell viability was determined using the CellTiter 96 Aqueous kit (MTS) (Promega) and absorbance was detected at 492 nm using a microplate reader (Infinite® M1000 Pro, Tecan). Absolute viability values were converted to percentage viability versus the untreated condition, and then non-linear fit of log versus response was performed in GraphPad Prism v6.0 to obtain an IC50 values. SP1 inhibitor Mithramycin (MTA) (Sigma-Aldrich, M6891) and MYC-MAX inhibitor 10058-F4 (Sigma-Aldrich, F3680) were used at 0.5 μM and at 10 μM respectively at the indicated time points. ITGA6 and Snail protein stability was evaluated by treating EOC cells with cycloheximide (CHX) (10 μg/ml), with MG-132 (10 μM) or Bafilomycin (0,2 μM) (Sigma-Aldrich) for the time indicated.

## Ovarysphere forming assay

Coated dishes were prepared to establish primary ovaryspheres. Poly-HEMA was dissolved in 95% Ethanol at 25 mg/ml final concentration and leave to dry. Once ready, EOC PT-sen and PT-res cells ($8 \times 10^3$) were plated on poly-HEMA coated 6-well plates as single cell suspension in phenol red-free DMEM/F12 (GIBCO), containing B27 supplement (without vitamin A; ThermoFisher), recombinant hEGF (10 ng/ml), recombinant h-bFGF (10 ng/ml) (PeproTech), and 1% penicillin/streptomycin (Sigma-Aldrich). Ovaryspheres were leave to growth for a total of 10 days. In a subset of experiments, cells were plated as usual and leave to growth for 5 days then CDDP (2 μM) was added to the medium and cells were leaved under treatments until the end of the experiment. In another set of experiments with the GoH3 blocking antibody, we coated a 24-well plate as above. TOV-112D ($2 \times 10^3$) cells were plated as a single cell and the day after GoH3 or IgG control antibodies were added at the concentration of 2.5 μg/ml and refreshed every day until the end of the experiment. The ovaryspheres were photographed with a Nikon Eclipse TS100 inverted microscope. The number and the sphere area were measured with the ImagJ program.

## Evasion assay

For Evasion assays, TOV-112D or OVSAHO cells ($7.5 \times 10^3$) were included in Matrigel drops (Cultrex, R&D Systems) at the final

concentration of 8 mg/ml (12 μl of matrix volume per drop). Matrigel was diluted in RPMI 1640 and 0.1% BSA. The drops, were dispensed in cell culture dishes with a minimum of 5 drops per condition and maintained for 1 h at 37 °C upside down to jellify. Then, the dishes were turned up, and the drops were incubated in complete medium. The evasion ability was evaluated 6 days after inclusion by measuring the distance covered by crystal violet-stained cells exiting from the drops (five drops/cell lines per experiment). Images were collected using a stereo microscope Leica M205FA and processed as first described in (Baldassarre et al, 2005).

## Preparation of conditioned medium (CM) and growth on mesothelium

For detection of extracellular ITGA6, confluent PT-sen and PT-res WT or ITGA6 KO cells were cultured for 24 h in serum-free medium in presence or absence of CDDP (25 μM). Conditioned Medium (CM) from all the cell lines were harvested at different time points, centrifugated to eliminates eventual residual cells, and freshly used to challenge other cell type or processed by the addition of Triton X-100 and trichloracetic acid (TCA) to precipitate proteins. Equal amounts of proteins were mixed with Laemmli buffer, separated in 4–20% SDS-PAGE (Criterion Precast Gel, Biorad) and blotted onto a nitrocellulose membrane to visualize secreted proteins. For functional assays with PT-sensitive cells, TOV-112D PT-sen were plated on 24-well plate and, after 24 h, challenged or not with CM from ITGA6 WT or KO PT-res cells for 16 h. After the removal of CMs, TOV-112D PT-sen cells were detached and co-cultured ($5 \times 10^4$) on mesothelial cell monolayer in complete medium and leave to growth for 10 days or plate as single cells on poly-HEMA coated dishes to ovarysphere formation. For functional assays with mesothelial cells, LPL mesothelial cells were plated on 24-well plate to 90% of confluence. After 24 h, LPL were challenged with CM from ITGA6 WT or KO for 16 h. After treatment with CM, TOV-112D PT-sen cells were plated on conditioned LPL monolayer and leave to growth for 10 days. For all these experiments, we evaluated TOV-112D PT-sen ovaryspheres forming ability, as above. In both functional assays, the challenging of TOV-112D PT-sen cells or mesothelial cells with CM from ITGA6 WT or KO cells, could be also in presence or absence of anti-ITGA6 blocking antibody, GoH3 (10 μg/ml) or anti-IGFBP6 blocking antibody (15 μg/ml) (AF876, R&D System) as indicated. Finally, to evaluate signaling activation TOV-112D PT-sen ($6 \times 10^5$) or LPL cells ($8 \times 10^5$) were plated in 6 mm dishes and 24 h after, they were exposed to CM from PT-res ITGA6 WT or KO cells for 5, 15, or 30 min and protein were collected.

## Processing ascites and conditioning mesothelial cells

Ascites were collected in our Institute from EOC patients with different histotype and stages (Table S1). First, primary tumor cells were separated from ascitic fluid by centrifugation at 1200 rpm for 10 min. Then, red blood cells were removed from the pellet by applying the lysis buffer (Lysing Buffer, BD Bioscience) for 5 min at 37 °C and re-centrifuged at 1200 rpm for 5 min. Primary cells were plated in 150 cm² Flasks in OCMI medium (Ince et al, 2015). Ascitic fluids collected after separation of primary cells were centrifuged at 3000 rpm form 10 min, filtered reducing the size of the filter mesh (45 μM and 22 μM) and conserved at −80 °C

(Table S1 for Patients' data). For experiments on mesothelial cells, LPL were plated at 90% of confluence in 24-well plate. After 24 h, they were challenged or not with 2% of ascitic fluids (described above) in serum-free medium for 16 h in presence or not with GoH3 blocking antibody or IgG (as negative control) (2.5 µg/ml). After 16 h of challenging, TOV-112D PT-sen cells ($5 \times 10^4$) were plated on the conditioned LPL in normal media and leave to growth for 10 days. We evaluated TOV-112D PT-sen sphere forming ability, as described above.

## Exosomes extraction

Exosomes were isolated from conditioned media of PT-sen and PT-res ITGA6WT and KO cells following manufacturer's indications for Total Exosome Isolation (from cell culture media) Kit (Invitrogen, 4478359). Briefly, conditioned media were centrifuged to remove cells and debris, then the supernatant was transferred to a new tube. Half volumes of Total Exosome Isolation reagent was added to the cell-free culture media: the mixture was thoroughly mixed and incubated overnight at 4 °C. The day after, the mixture was centrifuged to precipitate exosomes that will be contained in the pellet. The exosomes resuspended in 1X PBS were then used for characterization and functional assays. To evaluate ITGA6 levels, exosomes were lysed using cold RIPA buffer with protease inhibitors, protein concentration was determined by Bio-Rad protein assay (Bio-Rad) and then protein levels were evaluated by western blot analysis. ITGA6 proteins levels in exosomes upon CDDP treatment was evaluated adding the drug directly to serum-free media for the indicated time points before conditioned media harvesting. To evaluate the functional role of ITGA6 contained in exosomes, the resuspension was added to serum-free media to challenge TOV-112D PT-sen cells for 16 h which then were used to evaluate ovaryspheres formation ability on mesothelial cells, as described above.

## Preparation of cell lysates, immunoblotting, immunoprecipitation, and drugs treatments

Cell lysates were prepared using cold RIPA buffer [150 mM NaCl, 50 mM tris-HCl (pH 8), 0.1% SDS, 1% Igepal, and 0.5% NP-40] containing protease inhibitor cocktail (Roche) phosphatase inhibitors, 1 mM Na3VO4 and 10 mM NaF (Sigma-Aldrich) plus 1 mM DTT. Protein concentrations were determined using the Bio-Rad protein assay (Bio-Rad). Proteins were separated in 4 to 20% SDS–polyacrylamide gel electrophoresis (SDS-PAGE) (Criterion Precast Gel, Bio-Rad) and blotted onto a nitrocellulose membrane (Amersham, GE Healthcare). The complete list of used antibodies is reported in Appendix Table S4. Immunoprecipitations were performed using cell lysates in HNTG buffer (20 mM Hepes, 150 mM NaCl, 10% glycerol, and 0.1% Triton X-100) and the indicated primary antibody were incubated overnight at 4 °C. The immunocomplexes were precipitated by adding protein G or protein A agarose conjugated for an additional 1 h and 30 min at 4 °C. IPs were then washed in HNTG buffer, resuspended in 3× Laemmli Sample Buffer [5× Laemmli buffer composition: 50 mM tris-HCl (pH 6.8), 2% SDS, 10% glycerol, 0.05% bromophenol blue, and 125 mM β-mercaptoethanol], and finally separated on SDS-PAGE for Western blot analysis. Membrane strips were blocked with EveryBlot Blocking

Buffer (Bio-Rad) and incubated at 4 °C overnight with primary antibodies. Antibodies were visualized with appropriate horseradish peroxidase (HRP)-conjugated secondary antibodies (Bethyl Laboratories) for chemiluminescent detection (LiteUP, EuroClone) or with Alexa-conjugated secondary antibodies for Odyssey infrared detection (LI-COR Biosciences).

## Generation of ITGA6KO clones

Generation of ITGA6KO cells were performed using the protocol previously described (Sonego et al, 2019). Two specific ITGA6 guide RNAs (gRNAs) were designed using the CRISPR Design online tool (https://zlab.bio/guide-design-resources) to target the exon 1 of ITGA6 (NC_000001.11). The complete list of used oligonucleotides is reported in Table S4. The pairs of annealed oligonucleotides were cloned in the gRNA expression vector pSpCas9(BB)-2A-GFP plasmid (pX458, Addgene #48138) (www.addgene.org/). We transfected 2 µg of this plasmid in the TOV-112D and OVSAHO PT-res cells with Lipofectamine 2000 (Invitrogen) according to the manufacturer's protocol. Seventy-two hours after transfection, GFP-positive cells were sorted by flow cytometry (Becton Dickinson) and pooled together. Pooled cells were seeded as single colonies (0.5 cells per well) in ten 96-well plates. After 2 to 3 weeks of expansion, the cells in each colony were divided in half; one half was subjected to propagation and the other half was used to assess the expression of ITGA6 using Western blot analysis.

## Luciferase assay and transfection

To generate ITGA6 promoter containing plasmids, primers were designed based on the sequence of the 5' untranslated region of human ITGA6 promoter (Nishida et al, 1997). XhoI and BglII restriction sites were introduced into the forward and reverse primers, respectively. The genomic DNA isolated from TOV-112D cells was used as the template to amplify the ITGA6 promoter region. PCR products (of full length and deletion mutant 1, Mut1) were first subcloned in the pGEM-T easy vector (Promega) and then digested with XhoI and BglII restriction enzymes to be cloned in the pGL3-basic vector (Promega) at the 5' of the luciferase gene. To generate the deletion mutant 2 (Mut2), we digested the ITGA6 promoter full length containing vector with KpnI restriction enzyme and cloned in the pGL3-basic vector. Luciferase assay was performed to validate MYC and SP1 putative target sites on ITGA6 promoter essentially as described (Citron et al, 2021). TOV-112D cells were co-transfected with 500 ng of pGL3-ITGA6 promoter reporter and 50 ng of pRL Renilla Luciferase Control Reporter (pRL-TK) (Promega) (internal control for normalization of transfection efficiency) in 24-well plate using FuGENE® HD Transfection Reagent (Promega) according to manufacturer's recommendations. After 24 h, transfected cells were treated or not with 10058F4 (Sigma-Aldrich) 5 µM or with MTA (Sigma-Aldrich) 0,5 µM for 24 h or with DMSO (for control condition). Where indicated, pCMV-MYC (#40900, Addgene) vector was co-trasfected with pGL3-ITGA6 promoter and pRL-TK as described above. Luciferase activity was measured using the Dual Luciferase Reporter Assay System (Promega) according to the manufacturer's instructions. The relative luciferase activity was calculated as the ratio of firefly luciferase to Renilla luciferase.

**Box 1. Clinical timeline history of analyzed patients**

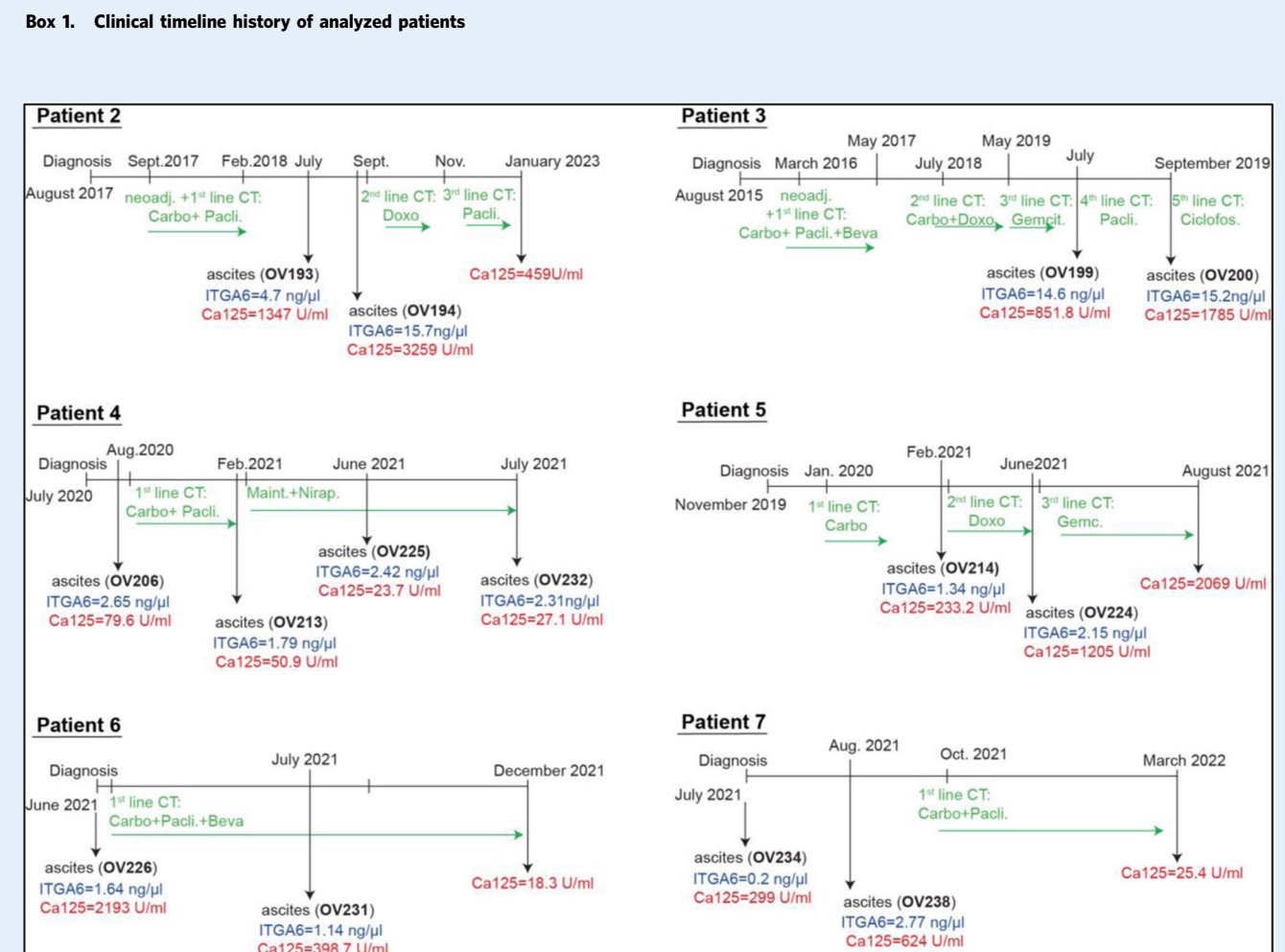

**Clinical timeline history** of 6 EOC patients analyzed by ELISA, reporting the timeline of diagnosis, surgery, chemotherapy treatments (in green) and ascites collections. The concentration of CA125 (in blood samples) and ITGA6 (in ascites) were reported in red and blue, respectively.

pRc-a6A and pRC-a6B plasmids encoding for ITGA6 isoforms were a kindly gift of A. Mercurio, RSV-SP1 vector was from Addgene (#12098).

## RNA isolation and real-time polymerase chain reaction (qRT-PCR)

For RNA expression analysis, TOV-112D or OVSAHO PT-sensitive or PT-resistant cells were plated in 6 mm plates and after 24 h h treated with CDDP (25 μM), or MTA (1 μM) or with 10058F4 (10 μM) for the indicated time points. Total RNA for qRT-PCR analyses was then isolated from cell cultures using Trizol solution (Roche Applied Science Mannheim, Germany) according to manufacturer protocol. Total RNA was quantified using NanoDrop (Thermo Fisher Scientific Inc., USA) and 500 ng of RNA was retro-transcribed with GoScript reverse transcriptase to obtain cDNAs, according to provider's instruction (Promega). 1/20 of the obtained cDNAs was amplified using primers specific for each gene as described in Appendix Table S5. Absolute quantification was evaluated by qRT-PCR, with EvaGreen

dye-containing reaction buffer (SsoFastTM EvaGreen®, BioRad) using the CFX96 TM Real-Time PCR Detection System (Bio-Rad). Data normalization was performed using ACTIN as housekeeping gene.

## Chromatin immunoprecipitation assay (ChIP)

TOV-112D PT-sensitive and PT-Resistant cells were treated with 50 μM of CDDP for 16 h. Then, treatment was removed, cells treated with 1% formaldehyde for the crosslinking and chromatin was prepared via MNase enxymatic digestion according to the protocol. Chromatin IP was performed using SimpleChIP Enzymatic Chromatin IP kit (Magnetic Beads) #9003 from Cell Signaling Technology (CST). ChIP was performed with recommended dilutions of antibodies using the rabbit monoclonal anti-MYC (#5605, Cell Signalling) or the mouse monoclonal anti SP1 (clone 4C8, Merk-Sigma) antibodies. After IPs, DNA was purified and analyzed by qRT-PCR (see above) using the primers described in Appendix Table S5. Results were reported as ITGA6 promoter enrichment of putative ITGA6 promoter fragment (SQ mean) folded on unrelated IP (control IgG).

## Human receptor tyrosine kinase phosphorylation antibody array

For the human receptor tyrosine kinase (RTK) phosphorylation antibody (Ab) array (RayBiotech Inc., Norcross, GA), TOV-112D PT-res ITGA6 WT and KO have been adhered on laminin for 30 min or 1 h and protein lysates were collected. Then the lysates were incubated with the array membranes and protein signal was visualized using a chemiluminecence detection system (Bio-Rad), according to the manufacturer's protocol. Relative density of specific protein expression was determined using ImageLab software (Bio-Rad).

## Cytokines array

For the human cytokine antibody (Ab) array (RayBiotech Inc., Norcross, GA), TOV-112D PT-res ITGA6 WT and KO cells have been cultured for 24 h in serum-free medium. Conditioned Medium (CM) from the cell lines were harvested and incubated with the array membranes according to the manufacturer's protocol and protein signal was visualized using a chemifluorescence detection system (Bio-Rad, Hercules, CA). Relative density of specific protein expression was determined using ImageLab software (Bio-Rad, Hercules, CA).

## ELISA

To detect ITGA6 levels in 7 EOC patients' coupled samples (see Clinical timeline history (Box 1) and Fig. 5F). 20 µl of purified ascitic fluid were diluted in 180 µl of PBS 1x and processed using commercially available ELISA kit (MBS2020560, MyBioSource, San Diego, CA) following the manufacturer's protocol.

## Statistical analysis

For in vitro experiments, at least three biological replicates were performed, according to good laboratory practice. For in vivo experiments a priori error probabilities were set $\alpha = 0.05$ and $\beta = 0.20$ to calculate the number of animals needed. No sample/ animal was excluded from the study. Animal were randomly assigned to each group of treatment. Pathological analyses were performed in blind by two expert pathologists.

Graphs and data analyses were carried out utilizing PRISM software (version 9, GraphPad, Inc.). Where the means of two data sets were compared, and significance was determined by a two-tailed Students t-test, as indicated in each figure. Differences were considered significant at $p < 0.05$. Exact $p$ values are reported in each figure panel, as necessary.

For Kaplan–Meier curves longrank test was used to calculate significance using the on line tool KM-Plotter (Győrffy, 2023).

## For more information

For clinical and translational research in gynecology oncology:
Multicenter Italian Trials in Ovarian cancer and gynecologic malignancies (MITO Group): https://www.mito-group.it/. European Network for Gynaecological Oncological Trial groups (ENGOT): https://engot.esgo.org/.

### The paper explained

#### Problem
Ovarian cancer remains a lethal disease mostly for the frequent development of chemo-resistant recurrences that metastasize in the abdomen and pelvis of patients. Understanding the molecular mechanisms behind resistance to chemotherapy and metastasis would help in identifying novel therapeutic approaches.

#### Results
We generated several in vitro models of PT-resistant (PT-res) ovarian cancer cells. PT-res cells displayed higher adhesion and invasive abilities compared to sensitive cells, which was mostly due to the higher expression of the adhesion molecule integrin α6 (ITGA6). In sensitive cells, ITGA6 expression was induced by PT treatment, while it was constitutively highly expressed in resistant cells. ITGA6 was involved in the formation of a pre-metastatic niche, acting both on bulk PT-sensitive cells and on mesothelial cells. Mechanistically, overexpressed ITGA6 in PT-res cells was secreted in the extracellular space and activated the IGF1R-Src pathway, leading to Snail protein stabilization in both PT-sensitive and mesothelial cells.

Genetic or pharmacological inhibition of ITGA6 reduces PT-res metastatic abilities and increases their sensitivity to PT both in vitro and in several in vivo PDX models of ovarian cancer.

#### Impact
We propose that ITGA6 targeting, by specific blocking antibody, could be exploited to improve the effects of PT-based chemotherapy in high risk ovarian cancer patients and prevents the appearance of recurrent/ resistant diseases.

For patients information:
ACTO: https://www.acto-italia.org/it. The World Ovarian Cancer Coalition (WOCC): https://worldovariancancercoalition.org/. The International Gynecological Cancer Society (IGCS): https://igcs.org/. European Network of Gynecological Cancer Advocacy (EnGAGE): https://engage.esgo.org/.

## Data availability

This study includes no data deposited in external repositories.

The source data of this paper are collected in the following database record: biostudies:S-SCDT-10_1038-S44321-024-00069-3.

## Peer review information

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

## Acknowledgements

We like to thank the patients who donated their samples, all members of the OM Unit for critical discussion of the data, Mr. Roberto Cirombella for technical support with mouse tissue processing and the Cytometry and Animal facilities of CRO-Aviano for support. This work was supported by CRO Aviano Ricerca Corrente core grant (linea 1) of Ministero della Salute, Ministero della Salute (CO-2018-12367051), Ministero degli Affari Esteri e della Cooperazione Internazionale (PGR01036), Ministero dell'Università (ARS01_00568), Associazione Italiana Ricerca sul Cancro (AIRC) (IG 26253), and by CRO-Aviano 5‰ grant to G. Baldassarre; by Associazione Italiana Ricerca sul Cancro (AIRC) (IG 20061) to B. Belletti; by Ministero della Salute (GR-2016-02361041) and Associazione Italiana Ricerca sul Cancro (AIRC) (MFAG 24321) to M. Sonego. Biorender.com was used to create some figure panels.

## Author contributions

Alice Gambelli: Data curation; Investigation; Visualization; Methodology; Writing—original draft. Anna Nespolo: Data curation; Investigation; Visualization; Methodology; Writing—review and editing. Gian Luca Rampioni Vinciguerra: Investigation; Methodology. Eliana Pivetta: Investigation. Ilenia Pellarin: Investigation; Methodology. Milena S Nicoloso: Resources; Investigation. Chiara Scapin: Investigation. Linda Stefenatti: Investigation. Ilenia Segatto: Investigation. Andrea Favero: Investigation. Sara D'Andrea: Investigation. Maria Teresa Mucignat: Investigation. Michele Bartoletti: Resources. Emilio Lucia: Resources. Monica Schiappacassi: Resources. Paola Spessotto: Resources. Vincenzo Canzonieri: Resources. Giorgio Giorda: Resources. Fabio Puglisi: Resources. Andrea Vecchione: Resources; Investigation. Barbara Belletti: Supervision; Funding acquisition; Writing—original draft; Writing—review and editing. Maura Sonego: Conceptualization; Data curation; Funding acquisition; Validation; Investigation; Visualization; Writing—original draft; Writing—review and editing. Gustavo Baldassarre: Conceptualization; Supervision; Funding acquisition; Validation; Writing—original draft; Project administration; Writing—review and editing.

Source data underlying figure panels in this paper may have individual authorship assigned. Where available, figure panel/source data authorship is listed in the following database record: biostudies:S-SCDT-10_1038-S44321-024-00069-3.

## Disclosure and competing interests statement

The authors declare no competing interests.

# Expanded View Figures

**Figure EV1. ITGA6 is commonly overexpressed in EOC PT-res cells.**

(A) Graphs reporting the percentage of cell adhesion on different extracellular matrices of TOV-112D and OVSAHO PT-sen and PT-res pools measured by the quantitative cell adhesion CAFCA assay ($n = 2$ in sextuplicate). Coll IV = Type IV Collagen; Coll I = Type I Collagen; FN = Fibronectin; LNs = Laminins; VN= Vitronectin. (B) Tables reporting the percentage of cells positive for the indicated integrins as measured by flow cytometry (FACS) in PT-sen and PT-res pools of TOV-112D and OVSAHO cells (ITGA1 = Integrin α1, ITGA2 = Integrin α2, ITGA5 = Integrin α5, ITGA6 = Integrin α6, ITGAV = Integrin αV, ITGB1 = Integrin β1, ITGB3 = Integrin β3, ITGB4 = Integrin β4). (C) FACS analyses of the expression profile of ITGB1, ITGB4, and ITGA6 integrins in OVSAHO PT-sen and PT-res clones. (D) Graphs reporting the percentage of cell adhesion on different extracellular matrices of the indicated TOV-112D and OVSAHO PT-sen and PT-res clones. ($n = 2$ in sextuplicate). (E) Western blot analysis evaluating the expression of ITGA6 protein in tumor samples from the same patients taken at diagnosis (Pre) or at recurrence after chemotherapy (Post). Tubulin was used as loading control. (F) qRT-PCR analysis of ITGA6 mRNA expression in different EOC PT-sen and PT-res cells. (G) Western blot analysis of ITGA6 expression in TOV-112D PT-sen and PT-res clone exposed to a time-course treatment with CHX for the indicated times. Vinculin was used as loading control. (H) Graph reporting the expression of ITGA6 in OVSAHO PT-sen and PT-res cells treated or not with CDDP for 2 h. mRNA was analyzed by qRT-PCR at the indicated time points after CDDP removal. (I) Graphs reporting qRT-PCR analysis of mRNA expression of ITGA6 in TOV-112D ITGA6^HIGH and ITGA6^LOW subpopulations treated or not with CDDP. In (F), (H), and (I), mRNA levels were analyzed in triplicate and normalized to actin housekeeping gene and expressed in arbitrary units. In all the graph of the figure statistical significance was determined by a two-tailed, unpaired Student's t-test (Exact *p* values were reported on graphs). Bars represent Standard Deviation.

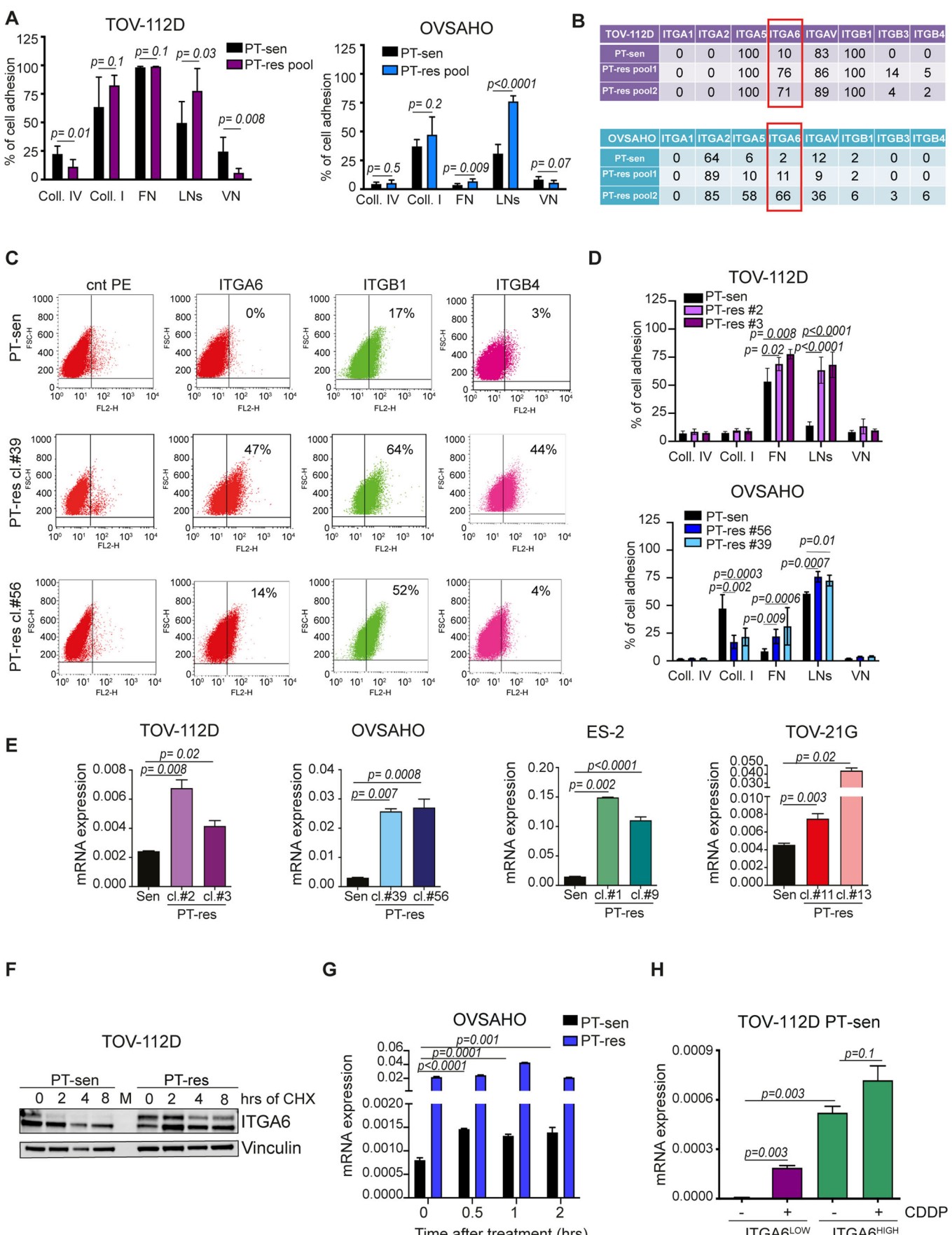

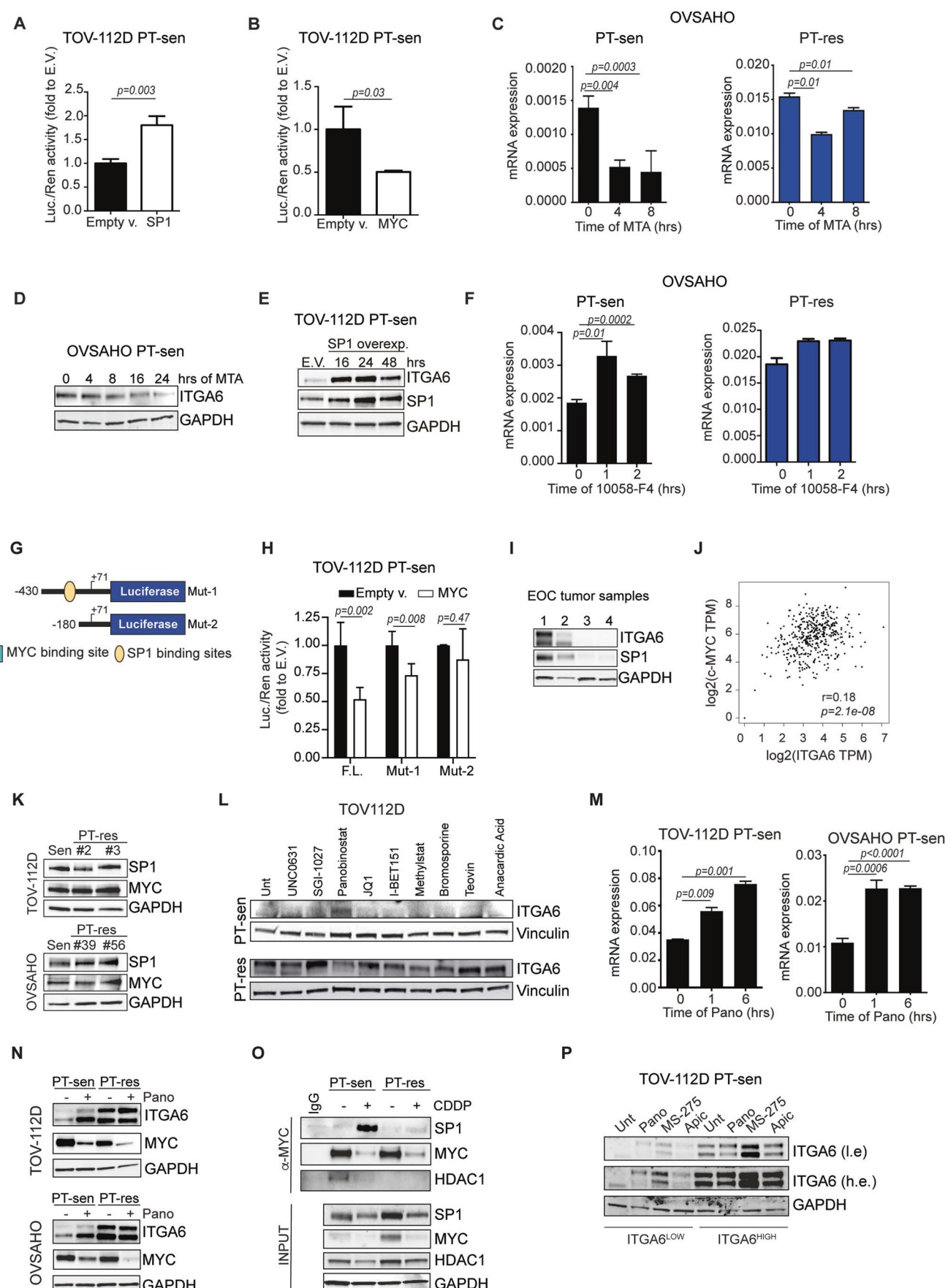

◄

**Figure EV2.  SP1/c-Myc/HDAC1 axis regulates ITGA6 transcription after CDDP treatment.**

(**A, B**) Graphs reporting the luciferase activity measured in TOV-112D PT-sen transfected with ITGA6 promoter full length (FL) together with SP1 (**A**) or c-Myc (MYC) (**B**) vectors. In A and B data are the mean (±SD) of three independent experiments. (**C**) Graphs reporting the expression of ITGA6 in OVSAHO PT-sen (left) and PT-res (right) cells treated with SP1 inhibitor (Mithramycin, MTA). mRNA expression was analyzed by qRT-PCR at the indicated time points. (**D**) Western blot analysis of ITGA6 expression in OVSAHO PT-sen cells treated with SP1 inhibitor, MTA for the indicated time points. (**E**) Western blot analysis of ITGA6 expression in TOV-112D PT-sen cells transfected with SP1 vector. Whole lysates were collected at the indicated time points after transfection. (**F**) Graphs reporting the expression of ITGA6 in OVSAHO PT-sen (left) and PT-res (right) cells treated with the c-Myc inhibitor (10058-F4)). mRNA was analyzed by qRT-PCR at the indicated time points. (**G**) ITGA6 promoter deletion mutants lacking c-Myc (Mut-1) or both c-Myc and SP1 binding sites (Mut-2) generated for this study. (**H**) Graph reporting the luciferase activity measured in TOV-112D PT-sen co-transfected with the full length (F.L.) or deletion mutants and the empty (black bars) or c-Myc (MYC, white bars) expression vectors. Data are the mean (±SD) of three independent experiments. (**I**) Western blot analysis of ITGA6 and SP1 protein expression in EOC tumor samples collected in our Institute (see Appendix Table S1). (**J**) Spearman's correlation analysis between ITGA6 and c-Myc mRNA expression in TCGA ovarian cancer dataset ($n = 489$ samples) using the GEPIA online tool. (**K**) Western blot analysis of c-Myc and SP1 protein expression in TOV-112D and OVSAHO PT-sen and PT-res clones. (**L**) Western blot analysis of ITGA6 protein expression in TOV-112D PT-sen and PT-res cells treated with a panel of epigenetic modifiers inhibitors for 24 h (UNC0631 and SGI-1027 = methyl transferase inhibitors; Panobinostat = pan-HDAC inhibitor; JQ1, iBET151 and Bromosporine = BET inhibitors; Methylstat = de-methyltransferase inhibitor; Tenovin= SIRTs inhibitor; Anacardic acid = HATs inhibitor). (**M**) Graph reporting the mRNA expression of ITGA6 in indicated PT-sen EOC cells treated with Panobinostat (Pano) for 1 and 6 h. mRNA expression was analyzed in triplicate in two biological replicates. (**N**) Western blot analysis of c-Myc (MYC) and ITGA6 protein expression in TOV-112D and OVSAHO PT-sen and PT-res clones treated or not with Panobinostat for 24 h. (**O**) Co-immunoprecipitation (Co-IP) analysis of c-Myc, HDAC1, and SP1 in TOV-112D PT-sen and PT-res cells treated or not with CDDP. Input shows the expression of the indicated proteins in the lysates used for IP experiments; IgG represents the control IP using an unrelated antibody. (**P**) Western blot analysis of ITGA6 expression in ITGA6^HIGH and ITGA6^LOW subpopulations treated with different HDACs inhibitors for 24 h (MS-275 preferentially inhibits HDAC1; Apicidin preferentially inhibits HDAC1 and HDAC3). In all western blots of the figure, GAPDH or Vinculin were used as loading control, as indicated. In (**C**), (**F**), and (**M**), mRNA expression was analyzed in triplicate and normalized to actin housekeeping gene and expressed in arbitrary units. In all the graphs of the figure, statistical significance was determined by a two-tailed, unpaired Student's t-test (Exact *p* values were reported on graphs). Bars represent Standard Deviation.

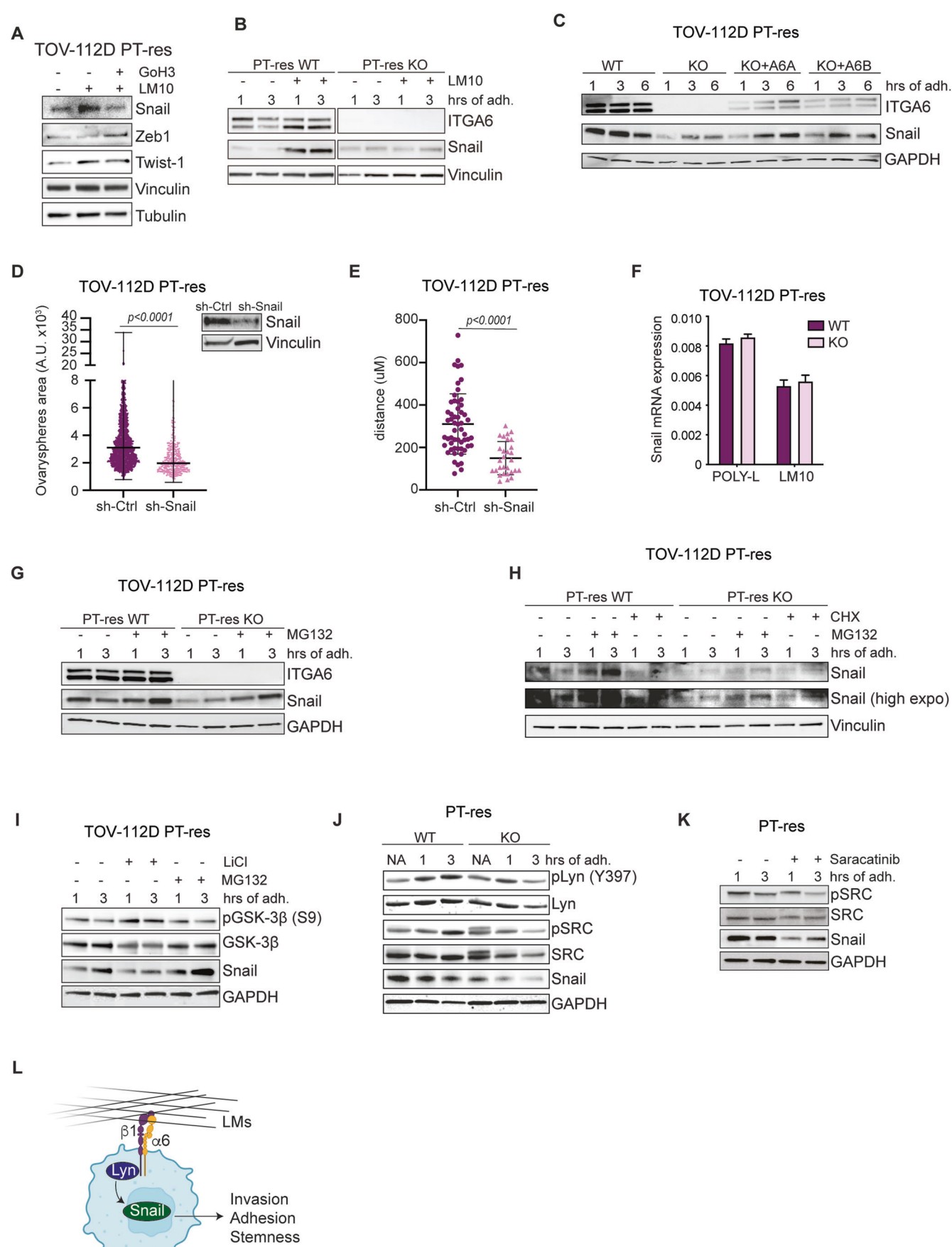

◀ **Figure EV3. ITGA6 engagement modulates Snail expression.**

(A) Western blot analysis of the indicated proteins in whole lysates of TOV-112D PT-res cells plated on LM10 coated dishes in the presence of IgG (as control) or of GoH3 Ab. (B) Western blot analysis evaluating ITGA6 and Snail expression in whole lysates of TOV-112D PT-res ITGA6WT and ITGA6KO cells plated or not on LM10 coated dishes for 1 and 3 h. (C) Western blot analysis evaluating the expression of ITGA6 and Snail in TOV112D PT-res ITGA6WT and KO cells and ITGA6KO transfected with vectors expressing the two isoforms of ITGA6 (isoform A, A6A and isoform B, A6B). Whole lysates were analyzed at the indicated time points after LM10 adhesion. (D) Graph reporting the area of ovaryspheres formed by TOV-112D PT-res cells stably transduced with control shRNA (sh-ctrl) or Snail shRNAs and plated on a mesothelial cell monolayer. In the inset Western blot reports Snail expression in the used cells. Data represent the median (±SD) of three independent experiments performed in triplicate in which at least 150 randomly selected cells were analyzed. (E) Graph reporting the distance covered by the individual cells described in (D), in Matrigel evasion assay calculated starting from the edge of the drop (mean ± SD of three independent experiments in which at least 10 randomly selected fields were analyzed). In D and E, statistical significance was determined by a two-tailed, unpaired Student's t-test (Exact $p$ values were reported on graphs). (F) qRT-PCR evaluating the mRNA expression of Snail in TOV-112D PT-res cells plated on Poly-lysine (negative control) or on LM10 coated dishes. mRNA expression (mean ± SD) was analyzed in triplicate and normalized to actin housekeeping gene expression. (G, H) Western blot analysis evaluating the expression of ITGA6 and Snail in TOV112D PT-res WT and KO cells plated on LM10 for 1 and 3 h and treated or not with the proteasome inhibitor, MG132 alone (G) or with CHX (H). (I) Western blot analysis evaluating the expression of ITGA6 and Snail in TOV112D PT-res ITGA6WT cells plated on LM10 for 1 and 3 h and treated or not with MG132 or with the GSK3β inhibitor, LiCl. (J) Western blot analysis evaluating the expression of the indicated proteins in lysates of TOV-112D PT-res ITGA6WT and ITGA6KO cells plated or not on LM10 coated dishes for 1 and 3 h (NA = not adherent). (K) Western blot analysis evaluating the expression of the indicated proteins in lysates of TOV-112D PT-res ITGA6WT cells plated on LM10 coated dishes for 1 and 3 h in the presence or not of the pan-SRC inhibitor Saracatinib. (L) Schematic representation of obtained results. After adhesion to LM10, ITGA6 activates Lyn/Src pathway and regulates Snail protein stability to mediate invasion, adhesion, and stemness of tumor cells. In all western blots of the figure, GAPDH, Tubulin, or Vinculin were used as loading control, as indicated.

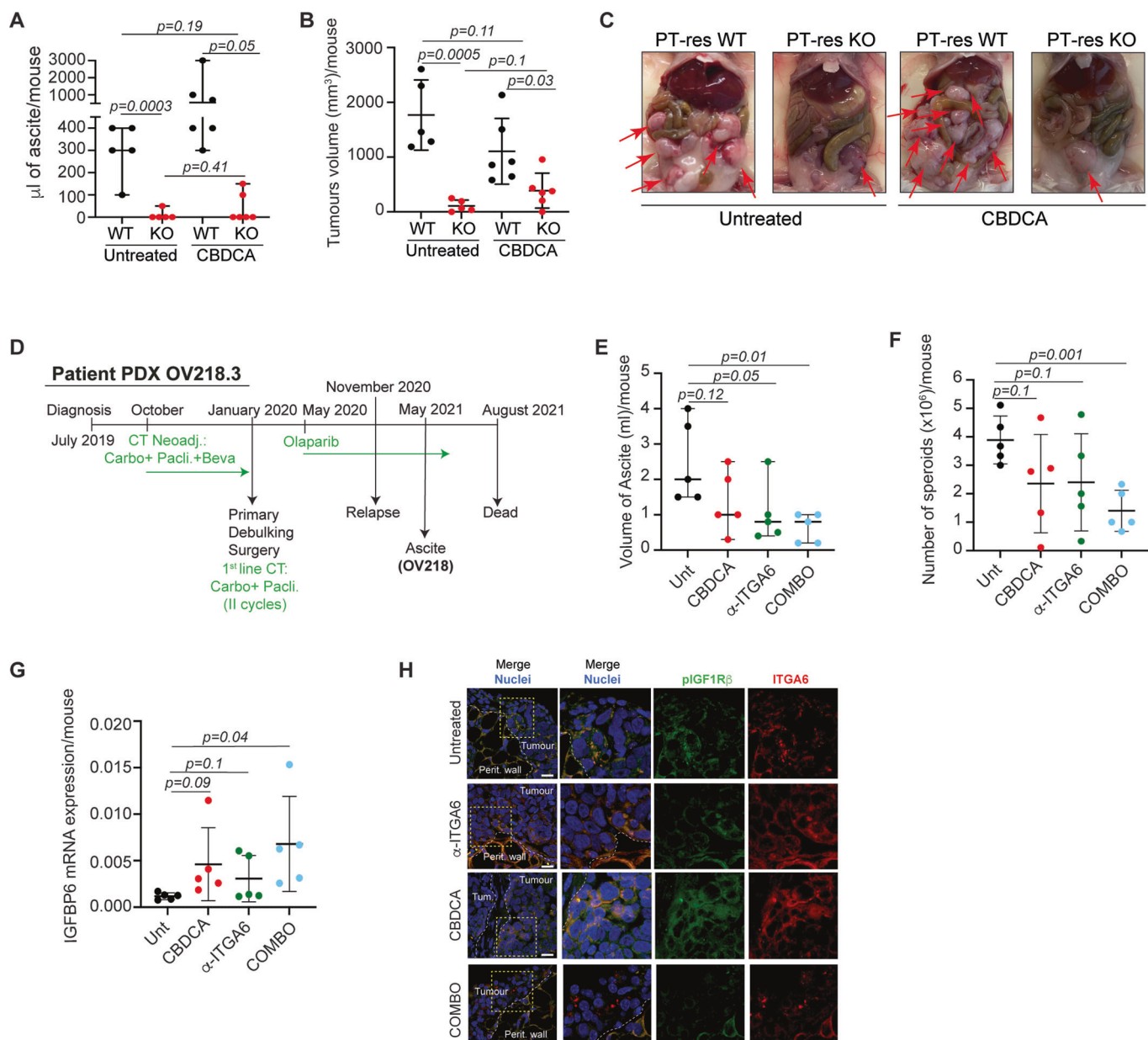

**Figure EV4.  ITGA6 is a druggable target necessary for in vivo cell spreading and response to PT.**

(A, B) Graph reporting the volume of ascitic fluids (A) and of explanted macroscopically identified tumors of NSG mice injected intraperitoneally with PT-res ITGA6 WT ($n = 11$) and KO cells ($n = 11$) and treated ($n = 6$) or not ($n = 5$) with CBDCA 30 mg/kg 3 times per week for 2 weeks. (C) Typical images of mice described in (A) and (B). Red arrows indicated the presence of macroscopically visible tumors. (D) Clinical history reporting the timeline of surgery, chemotherapy treatments (in green) and ascites collection of EOC patient who donate her ascites to establish PDX OV218.3 (see Methods section). (E, F) Graph reporting the volume of ascitic fluids (E) and number of tumor spheroids (F) in NSG mice ($n = 5$/group of treatment) injected with PDX OV218.3 and treated or not with the specific anti-ITGA6 blocking antibody P5G10, with CBDCA or with the combination of both according to the scheme reported in Fig. 6H. (G) IGFBP6 mRNA expression in tumor cells in ascites of mice described in (E, F). mRNA expression was analyzed in triplicate and normalized to actin housekeeping gene expression. The mean (±SD) expression for each mouse is reported in the graph. (H) IF analyses evaluating the expression of pIGF1Rβ (green) and ITGA6 (red) on peritoneal metastases collected from mice treated as indicated (nuclei are in blue). White dashed lines indicate the boundary between tumor masses and the peritoneal wall. Yellow dashed boxes represent the areas magnified in the zoomed images, on the right. Scale bars = 20 μm. In the figure, statistical significance was determined by a two-tailed, unpaired Student's t-test (Exact p values were reported on graphs). Bars represent Standard Deviation.

