## [Peer Review File · EMBO Molecular Medicine]

Platinum-induced upregulation of ITGA6 promotes chemoresistance and spreading in ovarian cancer

Alice Gambelli, Anna Nespolo, Gian Luca Rampioni Vinciguerra, Eliana Pivetta, Ilenia Pellarin, Milena Nicoloso, Chiara Scapin, Linda Stefanatti, Ilenia Segatto, Andrea Favero, Sara D'Andrea, Maria Teresa Mucignat, Michele Bartoletti, Emilio Lucia, Monica Schiappacassi, Paola Spessotto, Vincenzo Canzonieri, Giorgio Giorda, Fabio Puglisi, Andrea Vecchione, Barbara Belletti, Maura Sonogo, and Gustavo Baldassarre

Corresponding author: Gustavo Baldassarre (gbaldassarre@cro.it)

Review Timeline:

Submission Date:	3rd Jul 23
Editorial Decision:	27th Jul 23
Revision Received:	21st Feb 24
Editorial Decision:	15th Mar 24
Revision Received:	27th Mar 24
Accepted:	4th Apr 24

Editor: Lise Roth

Transaction Report:

27th Jul 2023

Dear Dr. Baldassarre,

Thank you for submitting your work to EMBO Molecular Medicine. We have now heard back from the referees who agreed to evaluate your manuscript. As you will see below, the reviewers raise substantial concerns on your work, which unfortunately preclude its publication in EMBO Molecular Medicine in its current form.

The reviewers find that the question addressed by the study is of potential interest, however they remain unconvinced that some of the major conclusions are sufficiently supported by the data. Further cross-commenting with the referees allowed us to better define the requirements for revisions:

- Translational impact and novelty should be increased by testing the combination of chemotherapy with the anti-ITGA6
- Mechanistic understanding should be strengthened
- Point 2 from referee #2 (mechanism by which ovarian epithelial cancer cells adhere to the mesothelium) will NOT be required for further consideration of the manuscript

If you feel you can satisfactorily address these points and those listed by the referees, you may wish to submit a revised version of your manuscript. Please attach a covering letter giving details of the way in which you have handled each of the points raised by the referees. A revised manuscript will once again be subject to review, and we cannot guarantee at this stage that the eventual outcome will be favorable.

We are expecting your revised manuscript within six months. If you anticipate any delay, please contact us.

We require:

4) A .docx formatted letter INCLUDING the reviewers' reports and your detailed point-by-point responses to their comments. As part of the EMBO Press transparent editorial process, the point-by-point response is part of the Review Process File (RPF), which will be published alongside your paper.

5) A complete author checklist, which you can download from our author guidelines (<https://www.embopress.org/page/journal/17574684/authorguide#submissionofrevisions>). Please insert information in the checklist that is also reflected in the manuscript. The completed author checklist will also be part of the RPF.

6) It is mandatory to include a 'Data Availability' section after the Materials and Methods. Before submitting your revision, primary datasets produced in this study need to be deposited in an appropriate public database, and the accession numbers and database listed under 'Data Availability'. Please remember to provide a reviewer password if the datasets are not yet public (see <https://www.embopress.org/page/journal/17574684/authorguide#dataavailability>).

In case you have no data that requires deposition in a public database, please state so in this section (This study includes no data deposited in external repositories.). Note that the Data Availability Section is restricted to new primary data that are part of this study.

7) For data quantification: please specify the name of the statistical test used to generate error bars and P values, the number (n) of independent experiments (specify technical or biological replicates) underlying each data point and the test used to calculate p-values in each figure legend. The figure legends should contain a basic description of n, P and the test applied.

Graphs must include a description of the bars and the error bars (s.d., s.e.m.). Please provide exact p values.

8) Our journal encourages inclusion of *data citations in the reference list* to directly cite datasets that were re-used and obtained from public databases. Data citations in the article text are distinct from normal bibliographical citations and should directly link to the database records from which the data can be accessed. In the main text, data citations are formatted as follows: "Data ref: Smith et al, 2001" or "Data ref: NCBI Sequence Read Archive PRJNA342805, 2017". In the Reference list, data citations must be labeled with "[DATASET]". A data reference must provide the database name, accession number/identifiers and a resolvable link to the landing page from which the data can be accessed at the end of the reference. Further instructions are available at .

9) We replaced Supplementary Information with Expanded View (EV) Figures and Tables that are collapsible/expandable online. A maximum of 5 EV Figures can be typeset. EV Figures should be cited as 'Figure EV1, Figure EV2' etc... in the text and their respective legends should be included in the main text after the legends of regular figures.

10) The paper explained: EMBO Molecular Medicine articles are accompanied by a summary of the articles to emphasize the major findings in the paper and their medical implications for the non-specialist reader. Please provide a draft summary of your article highlighting

11) For more information: There is space at the end of each article to list relevant web links for further consultation by our readers. Could you identify some relevant ones and provide such information as well? Some examples are patient associations, relevant databases, OMIM/proteins/genes links, author's websites, etc...

12) Author contributions: CRediT has replaced the traditional author contributions section because it offers a systematic machine readable author contributions format that allows for more effective research assessment. Please remove the Authors Contributions from the manuscript and use the free text boxes beneath each contributing author's name in our system to add specific details on the author's contribution. More information is available in our guide to authors.

13) Disclosure statement and competing interests: We updated our journal's competing interests policy in January 2022 and request authors to consider both actual and perceived competing interests. Please review the policy <https://www.embopress.org/competing-interests> and update your competing interests if necessary.

14) Every published paper now includes a 'Synopsis' to further enhance discoverability. Synopses are displayed on the journal webpage and are freely accessible to all readers. They include a short stand first (maximum of 300 characters, including space) as well as 2-5 one-sentences bullet points that summarizes the paper. Please write the bullet points to summarize the key NEW findings. They should be designed to be complementary to the abstract - i.e. not repeat the same text. We encourage inclusion of key acronyms and quantitative information (maximum of 30 words / bullet point). Please use the passive voice. Please attach these in a separate file or send them by email, we will incorporate them accordingly.

15) As part of the EMBO Publications transparent editorial process initiative (see our Editorial at <http://embomolmed.embopress.org/content/2/9/329>), EMBO Molecular Medicine will publish online a Review Process File (RPF) to accompany accepted manuscripts.

In the event of acceptance, this file will be published in conjunction with your paper and will include the anonymous referee reports, your point-by-point response and all pertinent correspondence relating to the manuscript. Let us know whether you agree with the publication of the RPF and as here, if you want to remove or not any figures from it prior to publication. Please note that the Authors checklist will be published at the end of the RPF.

I look forward to receiving your revised manuscript.

Yours sincerely,

Lise Roth

***** Reviewer's comments *****

Referee #1 (Comments on Novelty/Model System for Author):

This paper shows that ITGA6 is upregulated when ovarian cancer becomes platinum-resistant, based on molecular mechanisms. It also states that enhanced ITGA6 plays a major role in the formation of peritoneal dissemination.

The authors have conducted sufficient cell lines and experiments for each hypothesis, and are considered to have advanced the theory carefully.

It is interesting to note that platinum administration itself causes ITGA6 upregulation in transcripts, which is compatible with the current clinical situation, as the acquisition of platinum resistance is very common in ovarian carcinoma. It is also interesting to note that there may be a correlation between the degree of tumor dissemination and resistance to chemotherapy.

In general, I feel that this is a well-designed draft.

However, there are a number of points that are difficult for the reader to understand because of the multiple substances and cells involved. I think it is necessary to add a clear description of which cells and which substances were used in each experiment, even though it may be troublesome for authors. The following points are generally made from this perspective.

Fig1.

They are looking at ITGA6 expression in Primary and Recurrence, but can they consider comparing them in the same patient? For example, it would be interesting to compare a biopsy sample before chemotherapy and a sample at IDS after NAC. Could this be considered?

Fig. 2

Three reporter plasmids are shown, but only the top plasmid was used in this figure. The structure of the mut-plasmids should be introduced in the Supplementary Figure.

Fig3.

They conclude that the activation of ITGA6 ultimately leads to the activation of Snail, which is one of the famous component of EMT and changes in it will almost certainly affect various cellular functions, Can they confirm whether inhibition of Snail or pMAPK can prevent the functional changes?

Fig. 4

It is difficult to understand the hypothesis that mRNA expression is up, but intracellular ITGA6 is unchanged, so it may be secreted. Rather, since there is an increase in CD63, it might be expected that there is exosome-mediated secretion.

Why not extract the exosome fraction and test whether ITGA6 in it increases?

Also, they show here that IGF1R and its downstream are altered, but it is difficult to understand which cells they used. It would be better to indicate which cell line was used in each experiment in the Figure.

Also, it would be easier to understand if the schema is written at the end of this Figure.

Fig5.

From this point on, mesothelial cells also appear, so it is necessary to more carefully describe which cells were used. It would be better to show the expression pattern of the original mesothelial cells, and compare it with that of PT-sen and PT-res.

They argue that the increased protein expression of IGFBP6 may be post-modification, but what exactly do you envision? They can check some famous mechanisms, such as ubiquitination and phosphorylation.

Fig6.
C, H, and L are very difficult to understand. It would be better to change the way.
Also, it would be desirable to examine whether ITGA6-IGF1R axis is observed at the tumor site in mice, and whether there is indeed an interaction with mesothelial cells, by immunohistochemistry.

Fig. 7.
Rather than looking at the expression of related genes, why not compare the scores of the ITGA6 control vs. KO genes as a signature?
Also, without using TCGA data, it would be more convincing to show that ITGA6-IGF1R axis is indeed very active in some tumors in IDS samples after NAC. For example, if ITGA6 is not very well expressed during disseminated PDS, but very well expressed during IDS after NAC, it is very convincing.

My question is, is ITGA6 basically negative in ovarian cancer unless chemotherapy is given? Am I correct in understanding that once chemotherapy is given, ITGA6 mRNA is upregulated and secreted from exosomes, starting a negative chain of events? The effect about secreted-ITGA6 on PT-sen has been discussed a bit, but I was not sure about the effect on PT-res, so I would like to know as much as you can tell me.

Referee #2 (Remarks for Author):

In their manuscript, Gambelli et al. define how increased expression of the alpha6 integrin subunit (ITGA6) induced by platinum treatment (PT) promotes chemoresistance, growth, and invasion of epithelial ovarian cancer (EOC). In addition, the Authors show how ITGA6 may represent a novel therapeutic target and biomarker of reduced survival of EOC patients. The data presented by Gambelli et al. are very interesting in principle, however several aspects need clarification or further experimental investigation.

Major issues

1. Although the Authors show that GoH3 blocking antibody counteracts the effects of conditioned medium (CM) of PT-resistant (PT-res) EOC cells on PT-sensitive (PT-sen) EOC cells and mesothelial cells, this is not enough. The authors need to analyze whether ITGA6 plays its role as a cleaved and released protein in the CM or as a component of extracellular exosomal vesicles that can be isolated and characterized (e.g., see Hoshino et al. Nature 2015). Furthermore, does depletion of extracellular exosomal vesicles attenuate or abolish the biochemical (e.g., signaling via IGF1R) or biological (e.g., control of PT-sen spheroid size and adhesion to the mesothelium) properties of the CM of PT-res EOC cells?
2. In addition to proposing that ITGA6 acts via its soluble/exosomal form released in the CM, the Authors show that PT-res EOC cells adhere to the mesothelial cell monolayer via ITGA6. What is the mesothelial ligand to which ITGA6 located on the surface of PT-res EOC cells binds? Do mesothelial cells also express ITGA6? Do mesothelial cells secrete laminin apically and does this apical laminin act as a bridge between mesothelial cells and PT-res EOC cells? Or do PT-res EOC cells sneak between mesothelial cells and employ ITGA6 to adhere to the laminin basally laid down by mesothelial cells?
3. Anti-ITGA6 (Bhol et al., J. Dent. Res., 2001, 80:1711-1715) or anti-ITGB4 (obligate partner of ITGA6; Chan et al., Invest. Ophthalmol. Vis. Sci. 1999, 40:2283-2290) (auto)-antibodies have been shown to result in the pathological separation of epithelial cells from the laminin-containing basement membrane. This could be a significant limitation to the real therapeutic applicability of anti-ITGA6 antibodies in EOC patients. The Authors should investigate whether intraperitoneal treatment with the anti-ITGA6 GoH3 antibody may induce detachment of mesothelial cells or epithelial cells of the small and large intestine from the basement membrane to which they physiologically adhere.
4. How ITGA6 expression level affects the cells sensitivity to cisplatin was only tested in TOV-112D cells. Are these results also observed in the OVSAHO cell line?
5. In Figure S1A, cell adhesion was tested in the pool of TOV-112D and OVSAHO cells. Authors should test the adhesion of the different clones of each cell line further used in the manuscript.
6. The manuscript indeed contains many findings that should be organized and linked together according to a more interconnected and sequential logic. If secreted ITGA6, likely in exosomes, plays the predominant role from the pathogenetic point of view, the whole text section concerning ITGA6-mediated adhesion aspects of EOC cells could be downsized and the corresponding experimental data transferred to new Supplementary Figures.

Minor issues

1. The abstract should also describe the in vitro work done by the Authors.
2. Numbering pages and lines in the manuscript would facilitate references in the Reviewer's comments. I will therefore refer to the page numbers as they appear in the PDF file.
3. Page 5. The sentence is too long and should be simplified and rephased. "These data were in line with the notion that $\alpha 6\beta 1$ and $\alpha 6\beta 4$ heterodimers mediate cell adhesion to laminins (Hynes, 2002) and were then confirmed in all PT-res clones derived from pools (Sonogo et al, 2020) (Figure 1A-B and Figure S1C) and in EOC cells collected from patients at disease recurrence, which displayed higher ITGA6 protein expression compared to cells derived from patients with primary disease (Figure 1C, and Table S1A), suggesting that upregulation of ITGA6 represents a quite common modification associated with acquisition of EOC PT-recurrence/resistance".
4. Page 5. The sentence referred in the previous point (3) indicates that "upregulation of ITGA6 represents a quite common modification associated with acquisition of EOC PT-recurrence/resistance". However, it does not seem that is quite common, as the significant could come from the three patients with the higher expression, rather from the main distribution of the cohort.
5. Page 5. "Intriguingly, when we sorted PT-sen TOV112D cell subpopulations expressing either high or low levels of ITGA6 (ITGA6HIGH/ITGA6LOW), we observed that ITGA6HIGH cells were as resistant to cis-platin (CDDP) as the PT-res cell clones (Figure 1D-F)." At the end of this phrase, only Figure 1D-F are referred, when it should be Figure 1D-G.
6. Page 6. "Overexpression of SP1 or c-Myc, and their pharmacological inhibition with Mithramycin (MTA, SP1 inhibitor) or 10058-F4 (c-Myc inhibitor), demonstrated that basal ITGA6 expression was stimulated by SP1 and inhibited by c-Myc expression/activity in PT-sen cells (Figure 2C-F and S2C-F)." Figure 2C-F and S2C-F showed the inhibition of SP1 and MYC or only the overexpression of SP1, while the text also mentions the importance of MYC overexpression.
7. Page 6-7 and Figure S2K. "Among nine epigenetic modulator drugs (EMDs) tested, only Panobinostat, a pan-inhibitor of Histone De-Acetylases (HDACs), consistently increased ITGA6 expression in PT-sen models and had no effect on PT-res cells (Figure S2K-M)." The main text indicate that Panobinostat had no effect on PT-res cells, yet it appears there is a decrease compared to the control.
8. Page 9. "To test whether secreted ITGA6 was biologically active, PT-sen cells were incubated with the CM of WT or ITGA6KO PT-res cells for 16 hours prior to challenge their ability to adhere on the mesothelial monolayer, form spheroids and activate specific intracellular pathways (Figure 4B)". However, the adhesion to mesothelial cells was not tested or not reported in Figure 4B.
9. Page 9 and Figure S4C. "Accordingly, we observed that CDDP treatment induced the secretion of ITGA6 and CD63 (extracellular vesicles marker) in a time-dependent manner, peaking at 24 hours of CDDP treatment, in both PT-sen and PT-res cells (Figure 4A and S4C)." In the main text, it is indicated that ITGA6 expression peaked at 24h post-cisplatin, but it is only tested up to 16h.
10. Page 10 and Figure 4E. "Among different tested signaling pathways, activating phosphorylations of the beta subunit of the Insulin-like Growth Factors 1 Receptor (IGF1R β) and Src tyrosine kinases were specifically enriched in cells stimulated with CM from WT, but not ITGA6KO, PT-res cell." The differences in the expression of these proteins between the two conditioned media are not that obvious, at least for Src, as indicated in the mentioned sentence.
11. Pages 12-13. The anti-ITGA6 blocking antibody was given twice in Figure 6G, but three times in the PDX model of Figure 6K. Based on what these treatments planning were decided? Why are they different? May Authors explain the rationale?
12. Page 13 and Figure 7C-D. "Importantly, high LAMA5 and/or IGF2 predicted shorter PFS and OS also in the whole EOC patients' population (Figure 7C and D)". The figures do not correspond to the text and there is no Figure 7D.
13. Please check the consistence in the numbering of PT-res cell clones. It appears that in Figure 1A-B and FigureS1C different PT-res clones were employed.
14. Figure 2J and S2I showed Spearman and Pearson analyses, but in each report a different one, 2J shows the Spearman r and p-value, while S2I reports the data from Pearson.
15. Figure 4E and 4F: both figures showed similar conditions, yet the result regarding Snail looks quite different at 15 and 30 minutes of conditioning. May Author check and comment?
16. Figure 4F lacks a NC control.

17. Figure 5D, it should be clearly specified that difference is detected at 15 minutes.
18. Figure 7C and 7D. The figures do not correspond to the text (page 13) and there is no Figure 7D.
19. Figure S1B, C and D seem to refer to completely different PT-res clones: TOV-112D #1, #2, #3; OVSAHO #1, #2, #39, #56. Is the numbering correct?
20. Figure S3D. The number of spheres is only reported for OVSAHO cells, but not for TOV-112D.
21. Figure S3C and F. The same axes scale should be used.
22. Figure S3 K and L. The only difference between these two graphs is the presence of ITGA6. The expression of ITGA6 should be tested in the same settings, as in Figure S3L.
23. Page 13. "(Figure 6H-J and S6D-E)". There is no figure S6E.
24. Figure S7G-H: these two graphs are not described in the figure legend and neither mention in the main body of the paper.
25. Table S1 and S2 do not indicate the number of patients, which should be around 30.
26. It would be interesting to include a third Supplementary Table like Table S2 but reporting the kinases analyzed in Figure S3N.

Referee #3 (Comments on Novelty/Model System for Author):

Technical quality and model systems are adequate. Novelty is an issue, since the link between ITGA6 and chemoresistance in ovarian cancer has been reported already (PMID 34600083, 30930993), including a causal link between ITGA6 and platinum resistance (PMID 28131812).

While the study is relevant for a better understanding of the biology behind ovarian cancer biology and drug responsiveness, the clinical implications are limited until the ability of ITGA6-targeted therapy to improve chemoresponse is shown in clinically relevant settings.

Referee #3 (Remarks for Author):

Gambelli et al. report that platinum-based chemotherapy induces the upregulation of ITGA6 in OC cells which, in turn, promotes chemoresistance, acquisition of stemness-related features, and mesothelial invasion. Moreover, they show that ITGA6 is required to sustain an EMT program that apparently relies on Lyn-dependent stabilization of Snail.

This is a well-conducted study, where the experimental strategy is clearly delineated and follows a logical sequence. The manuscript is well written and reader-friendly.

Overall, this manuscript describes the role of ITGA6 as a hub that in OC orchestrates response to chemotherapy, cancer stemness and EMT, and peritoneal spreading. This makes it in principle an interesting and relevant study. However, the current version raises several major concerns that should be addressed to improve the quality of the manuscript, to solve some conceptual issues, and to support some of the authors' interpretations and conclusions.

Major points:

1. The novelty of the study is somehow limited. Other reports have previously linked ITGA6 to chemoresistance (reviewed in PMID 37444576), including OC (PMID 34600083, 30930993). Villegas-Pineda et al. have also causally implicated ITGA6 in the platinum resistance of OC cells (PMID 28131812). Moreover, the role of SP1 in the transcriptional regulation of ITGA6 (PMID 34199886, 17652716), the crosstalk between ITGA6 and Lyn is also known (PMID 32219444), and the role of ITGA6 in cancer stemness and EMT (reviewed in PMID 37444576) have been described already. Of course, the authors here provide a unifying view of all these findings in the context of OC chemoresistance and malignancy, and this has its own value. Yet, I'm not sure that it is sufficient for a journal like EMBO Mol Med.
2. How does CDDP enhance SP1 recruitment to ITGA6 promoter in Pt-sen cells? This would be a novel mechanistic insight.
3. The ITGA6 protein migrates as a doublet in most immunoblots of cell lysates (not in CM). What do the two bands correspond to? This is not trivial, since in some experiments the two bands follow a different fate (which in some cases contradicts the authors' conclusion). For example, PT-sen ES-2 cells show high levels of the lower band, and in their PT-res counterpart this band is downregulated while the upper band is upregulated. The opposite occurs in TOV-21G cells (Fig. 1B). Adhesion of PT-res TOV-112D to LM10 caused reduction of the upper band and increase of the lower band (Fig. 3I). Always in TOV-112D cells, CHX treatment of PT-res cells is accompanied by no change (or even an increase) in the lower band and only the upper band decreases (Fig. S1E).

4. Page 5. Referring to three different isogenic PT-sen vs PT-res cell lines, the authors claim that "all PT-res counterparts increased their ability to adhere to laminins (Figure S1A)...". In fact, Fig. S1A shows only two cell lines, and in one of them, TOV-112D, the effect is quite minor and with high error bars.

5. Fig. 1D. The ITGA6-based FACS profile of PT-sen cells depicts two sharply distinct cell populations within an established cell line. Is that really the case or it has to do with the gating conditions or other technicalities? Is such a separation seen also in other OC cell lines? This is quite relevant and should be at least discussed.

1. Page 6. "Moreover, [in PT-res cells] SP1 binding to ITGA6 promoter was stably high and slightly reduced by CDDP treatment (Figure 2G) ...". This does not appear to be the case: CDDP reduces it quite remarkably (almost 3-fold), which is quite puzzling. This also raises another doubt: how comes that such a reduction of promoter-bound SP1 does not lead to a downregulation of ITGA6 expression, and we instead see an upregulation?

6.

7. Page 8. "Using the ovarysphere formation assay ..., we observed that PT-res cells formed more and/or bigger spheres than PT-sen ones". This does not appear to be the case: Fig. 3F shows that ovarysphere area is smaller in untreated, WT PT-res cells than PT-sen cells. The opposite applies only to CDDP-treated cells.

8. Page 9. "... upon adhesion to LM10, the phosphorylation at Y508 (inhibitory) was sharply increased in PT-res ITGA6KO cells..." and "... ITGA6 engagement to LM10 in EOC cells increases Snail protein stability...". In fact, the authors cannot formally ascribe these effects to the adhesion of cells to LM10, since their experiments in Figs. 3J and 3K lack the "time 0" condition (i.e., no LM10). Thus, the effect of adhesion to LM10 on these signaling events cannot be determined. In the same context, it is unclear why the authors data related to SRC if the relevant kinase is Lyn. Also, Fig. 3K should relate the regulation of Snail to phosphorylation status of Lyn, not SRC. This is crucial, because without it the regulation of Snail cannot be causally linked to Lyn activity. Finally, SRC data in Figs. 3J and 3K are discrepant: adhesion of PT-res to laminin shows no pSRC in 3J and detectable pSRC (reduced by Saracatinib) in 3K.

9. Page 9 and Fig. 4A. What is the rationale for including CD63 here? It is mentioned as an extracellular vesicle marker, yet no extracellular vesicle purification is performed here, so ITGA6 secretion in the CM remains unrelated to extracellular vesicles.

10. Pages 9-10 and Figs. 4C, D. The reduction in sphere formation when using the CM from ITGA6-KO cells as compared to WT cells is ascribed to the lack of secreted ITGA6. This conclusion cannot be formally drawn, since it could depend on an indirect effect of ablating ITGA6 on the secretome of PT-res cells (e.g., downregulation of other secreted factors that promote adhesion to mesothelium or the induction of factors that counteract it). Depleting ITGA6 from the CM of WT cells or using the blocking antibody shown in the following experiments would enable to determine the direct effect of secreted ITGA6.

11. Fig. 4E. The correct interpretation of these data appears to be that ITGA6-containing CM sustains the signaling (pSRC and Snail), while the authors claim a specific enrichment in cells treated with WT CM.

12. What is the ITGA6 function blocked by GoH3? If it is binding to ECM (laminin), how comes that the antibody blocks the signaling induced by soluble ITGA6 in the CM? For example, does the LM-binding site overlap with the binding site for IGF1R? This should at least be discussed.

13. Page 13 and Fig. 7/ Fig S7. The authors' interpretation of these data is somehow inconsistent. First, the association of high LAMA5 with shorter overall survival (Fig. 7D, once correctly labeled) can be considered a trend at maximum (log-rank $p=0.057$) and the Results should reflect this. Then, data in Fig. S7D are claimed to establish a correlation between high LAMA5 and shorter PFS, not considering a log-rank $p=0.093$. On the contrary, the association of high IGFBP6 with shorter PFS (log-rank $p=0.033$) "did not have any prognostic value".

14. Page 14 (Discussion) and Fig. 6K. "...targeting ITGA6 in combination with standard PT-based chemotherapy could be beneficial to elicit a better response to PT...". Why was this not tested experimentally? The authors appear well positioned to assess the effect of the GoH3 + CDDP combination, which would have relevant translational implications and would probably change the impact of the manuscript.

15. Page 15 (Discussion). "However, these studies did not investigate if and how ITGA6 participates in determining the phenotypes of PT-resistance". Actually, Villegas-Pineda et al (PMID 28131812) showed that silencing ITGA6 sensitizes to platinum-based chemotherapy.

Minor points:

1. Fig. 1C. Vinculin doesn't really serve as a loading control in that blot (as claimed in the figure legend. It's highly variable

among different samples.

2. Fig. 6 G-M. Does GoH3 cross-react with murine ITGA6? This is important to assess if the antibody can also affect the host response to tumor.
3. Fig. S6E is referenced to in the Results (page 13), but is not present in Fig. S6.
4. Fig. 7. Panels C and D are referenced to in the Results (page 13) but are not correctly labeled in the figure. Current panel C should be changed to E.
5. Page 13. PDX #OV215.3 is claimed to derive from "patient #8 in Table S1". Unfortunately, Table S1 does not list the patients, and the patients' history in the scheme at the end Materials and Methods does not include Patient #8. The patients' list should indeed be included, also to assess the data of Fig. 1C.
6. No information is provided on the origin, isolation and culture conditions of mesothelial cells. They are only described as "derived from pleural lavage".

Gambelli et al. Point by Point response

Editor Remarks

The reviewers find that the question addressed by the study is of potential interest, however they remain unconvinced that some of the major conclusions are sufficiently supported by the data. Further cross-commenting with the referees allowed us to better define the requirements for revisions:

We thank the Editor and the Reviewers for finding our work of potential interest and for their suggestions to improve the manuscript. As specified below, we largely agreed on their comments and responded in full. We are particularly grateful to the Editor and Reviewers since, in this case, their work helped us to identify and resolve some inaccuracies present in the previous version of the manuscript.

We hope that in the present form the manuscript will be acceptable for publication in EMM.

- Translational impact and novelty should be increased by testing the combination of chemotherapy with the anti-ITGA6

To answer this question, we used a EOC PT-Resistant PDX model, proving *in vivo* that the combination of anti-ITGA6 with carboplatin is highly effective in limiting growth and dissemination. These data are now included in the new Figure 6 and Expanded View 4.

- Mechanistic understanding should be strengthened

We improved the mechanistic understanding of our results following the suggestions of the three referees, as detailed in the point by point response. In particular, we have now proved that ITGA6 is secreted in exosomes by EOC cells and that ITGA6-containing exosomes are able to stimulate the acquisition of mesenchymal characteristics in PT-sen cells (e.g. spheroid formation and evasion/motility). We also proved that Snail is a critical mediator of ITGA6 activation in PT-res cells.

- Point 2 from referee #2 (mechanism by which ovarian epithelial cancer cells adhere to the mesothelium) will NOT be required for further consideration of the manuscript

As suggested we did not further investigate this point, while we better addressed all other points raised.

Referee #1 (Comments on Novelty/Model System for Author):

This paper shows that ITGA6 is upregulated when ovarian cancer becomes platinum-resistant, based on molecular mechanisms. It also states that enhanced ITGA6 plays a major role in the formation of peritoneal dissemination.

The authors have conducted sufficient cell lines and experiments for each hypothesis, and are considered to have advanced the theory carefully.

It is interesting to note that platinum administration itself causes ITGA6 upregulation in transcripts, which is compatible with the current clinical situation, as the acquisition of platinum resistance is very common in ovarian carcinoma. It is also interesting to note that there may be a correlation between the degree of tumor dissemination and resistance to chemotherapy.

In general, I feel that this is a well-designed draft.

However, there are a number of points that are difficult for the reader to understand because of the multiple substances and cells involved. I think it is necessary to add a clear description of which cells and which substances were used in each experiment, even though it may be troublesome for authors. The following points are generally made from this perspective.

We thank Referee #1 for finding of interest our manuscript and for Her/His general appreciation of our experimental design. We believe that Her/His suggestions and comments helped us to significantly improve the manuscript and we hope that S/He will find it acceptable for publication in EMM.

Fig1.

They are looking at ITGA6 expression in Primary and Recurrence, but can they consider comparing them in the same patient? For example, it would be interesting to compare a biopsy sample before chemotherapy and a sample at IDS after NAC. Could this be considered?

We agree with Referee #1 that this would be an important validation of our *in vitro* findings. Unfortunately, we do not have enough matched samples to compare the untreated biopsy with the tumor upon chemotherapy to properly answer this question. However, looking at the levels of ITGA6 in EOC cells retrieved by ascites from the same patient at

diagnosis and after (or during) chemotherapy, we observed an increased expression of ITGA6 in 2/3 cases available (see figure below). Since this is just an anecdotal observation, due to the very low number of cases available, we provide this information to the reviewer but we have not included it in the final version of the manuscript.

Figure for reviewers removed

Fig. 2

Three reporter plasmids are shown, but only the top plasmid was used in this figure. The structure of the mut-plasmids should be introduced in the Supplementary Figure.

As suggested, we moved the structure of plasmids in the new figure Expanded View 2G

Fig3.

They conclude that the activation of ITGA6 ultimately leads to the activation of Snail, which is one of the famous component of EMT and changes in it will almost certainly affect various cellular functions, Can they confirm whether inhibition of Snail or pMAPK can prevent the functional changes?

We thank Referee #1 for this suggestion. To answer Her/His question we silenced Snail in TOV112D PT-resistant cells using a shRNA approach validated in our lab in platinum sensitive ovarian cancer cells (PMID: 31086816). Compared to control-transduced cells, Snail silenced cells showed a decreased ability to form ovary-spheres on mesothelial cells and also a decreased ability to evade from Matrigel, confirming that increased Snail expression is a key step after ECM engagement of tumor cells (new figure Expanded View 3D-E).

Fig. 4

It is difficult to understand the hypothesis that mRNA expression is up, but intracellular ITGA6 is unchanged, so it may be secreted. Rather, since there is an increase in CD63, it might be expected that there is exosome-mediated secretion. Why not extract the exosome fraction and test whether ITGA6 in it increases?

Also, they show here that IGF1R and its downstream are altered, but it is difficult to understand which cells they used.

It would be better to indicate which cell line was used in each experiment in the Figure.

Also, it would be easier to understand if the schema is written at the end of this Figure.

We agree with Referee #1 on this point and thus verified if ITGA6 was secreted in exosomes. Our results showed that ITGA6 expression is mostly limited to the exosome fraction (Total Exosome Isolation Kit, Invitrogen) and that exosome-depleted conditioned medium does not express detectable levels of ITGA6 compared to *non*-depleted ones and that exosome isolated from PT-treated cells contains ITGA6 (new Figure 4B-C). We also showed that exosomes extracted from ITGA6 WT PT-Res cells better stimulate the formation and the growth of ovary-sphere (area) in PT-Sen cells plated on mesothelial cells, compared to exosomes from ITGA6 KO cells (new Figure 4D and Appendix S2C). Unfortunately, we could not compare the effects of CM containing or not exosomes since the latter resulted toxic when used on PT-sen cells, probably due to the buffer that is used during the extraction procedure (see images below, provided as an example).

To increase clarity, we moved the schema of the experiment at the end of Figure 5, since it includes the effect of both overexpressed and secreted ITGA6 (see new figure 5H), as suggested.

Figure 2R. Pictures show the effects of serum free conditioned medium, serum free exosome-depleted conditioned medium (deprived CM) and serum free + exosomes, on the viability of TOV112D cells. Cells, incubated for 16 hours with serum free exosome-depleted conditioned medium (deprived CM) started to die and could not be used for further experiments (lower left panel).

Fig5.

From this point on, mesothelial cells also appear, so it is necessary to more carefully describe which cells were used. It would be better to show the expression pattern of the original mesothelial cells, and compare it with that of PT-sen and PT-res.

We thank the reviewer for this suggestion and we have now included the description of mesothelial cells used in the methods section (see pag. 20). We also compared the expression of IGF1R-SRC-SNAIL axis in not-stimulated mesothelial PT-Sen and PT-Res cells as requested (new figure Appendix S4A). The mesothelial cell marker Cytokeratin 5 (CK5) and Calretinin were used to confirm mesothelial origin of the cells.

They argue that the increased protein expression of IGFBP6 may be post-modification, but what exactly do you envision? They can check some famous mechanisms, such as ubiquitination and phosphorylation.

We thank the reviewer for this suggestion. We tested if IGFBP6 expression could be modified in PT-Res WT cells blocking either the proteasome-dependent (MG-132 treatment) or the lysosome-dependent degradation (bafilomycin treatment). By these experiments, we observed that lysosomal-mediated degradation could be implicated but we also observed that it could not explain alone all the differences observed between ITGA6WT and KO cells.

Thus, we returned to the RNA analyses and found an increased mRNA expression of IFGBP6 in all tested ITGA6KO clones. We also confirmed an increased IGFBP6 expression in *in vivo* experiments with PDX.

We have included these data in the new figures Appendix S3D-F and Expanded view 4G and corrected the text accordingly.

Fig6.

C, H, and L are very difficult to understand. It would be better to change the way.

We agree that radar plots could be of difficult interpretation but we believe that this is the easiest way to graphically represent the complex metastatic dissemination of ovarian cancer in mice. We have now slightly modified the radar plots and explained better their interpretation in the text, to make them easier to be interpreted by the reader (new figure 6C, 6I, and Appendix S5E and S5J).

Also, it would be desirable to examine whether ITGA6-IGF1R axis is observed at the tumor site in mice, and whether there is indeed an interaction with mesothelial cells, by immunohistochemistry.

To address this point, we have now used the PT-resistant PDX 218.3 and evaluated the activation of the pathway described *in vitro*, by immunofluorescence, WB and qRT-PCR analyses. Combination treatment significantly increased the expression of the DNA-damage marker γ -H2AX and of IGFBP6. Immunofluorescence analyses showed that Snail expression was consistently upregulated by PT treatment, and co administration of ITGA6-Ab completely prevented this increase. Single ITGA6-Ab treatment also reduced basal expression of Snail. Co-staining of peritoneal metastases with anti-ITGA6 and anti-phosphoIGF1R β antibodies revealed a clear co-localization of active IGF1R β and ITGA6, especially at the site of tumor-mesothelium contacts. This co-localization further increased upon PT-treatment and almost completely abolished when the anti-ITGA6 Ab was used, mostly for a decreased expression of phosphoIGF1R β (Figure 6M and L and Expanded view 4H).

Fig. 7.

Rather than looking at the expression of related genes, why not compare the scores of the ITGA6 control vs. KO genes as a signature?

Since datasets are based on mRNA expression that not always parallel protein expression, we preferred to use the genes whose transcription is positively regulated by ITGA6 rather than the ones modified at protein levels. For instance, we do know that SNAIL (encoding for Snail) is not prognostic in EOC although it is critically implicated in EOC progression.

Also, without using TCGA data, it would be more convincing to show that ITGA6-IGF1R axis is indeed very active in some tumors in IDS samples after NAC. For example, if ITGA6 is not very well expressed during disseminated PDS, but very well-expressed during IDS after NAC, it is very convincing.

Yes, we agree with the Referee that this evidence would be very convincing. Unfortunately, we do not have enough tumor samples to test this possibility but we are proved their involvement and activation in novel PDX model used to verify the role of ITGA6 Ab with or without carboplatin treatment in vivo (see new Figure 6 and Expanded View 4).

My question is, is ITGA6 basically negative in ovarian cancer unless chemotherapy is given? Am I correct in understanding that once chemotherapy is given, ITGA6 mRNA is upregulated and secreted from exosomes, starting a negative chain of events?

The effect about secreted-ITGA6 on PT-sen has been discussed a bit, but I was not sure about the effect on PT-res, so I would like to know as much as you can tell me.

We thank the reviewer for Her/His interest. We do know and reported that a subset of PT-sen EOC cells express ITGA6 in basal condition and also that it is expressed in primary tumors, although at lower levels compared to the recurrent/resistant ones. So, we do not think that ITGA6 is not expressed in ovarian cancer.

However, our results clearly indicate that its expression increases upon platinum treatment and that PT-resistant ovarian cancer cells stably express higher levels of ITGA6 protein.

Our working hypothesis, at the end of our work, is that under the pressure of chemotherapy cell clones that express higher ITGA6 levels have a growth/survival advantage and, therefore, might be positively selected.

Of course, this hypothesis should be better validated with experiments aimed at study EOC clonal evolution under the pressure of chemotherapy. This is a different project that we have already started in collaboration with Dr Andrea Viale at MD Anderson Cancer Center, using a barcoding approach (PMID: 30726735) and will hopefully soon answer this open question.

We also observed that a transcriptional program is activated within 1-2 hours after PT treatment, leading to the transcription of ITGA6 in PT-sen cells, likely due to epigenetic regulation of its promoter activity. In particular, we believe that loss of Myc negative regulation of SP1 might be relevant in driving ITGA6 expression. Increased expression of ITGA6 could represent an adaptive survival mechanism exploited by PT-sen cells to survive.

This observation is in line with recent notions in breast and colon cancer, showing that the activation of specific transcriptional programs, also associated with decreased Myc-signatures, are necessary for cancer cells to survive the pressure of chemotherapy (see commentary in PMID: 33561394). Conversely, in PT-res cells this transcriptional program induced by PT is constitutively active and, thus, post-transcriptional regulation of ITGA6 could be relevant (e.g. its lysosomal degradation).

We have now better discussed the possible effects of ITGA6 in Pt-res ovarian cancer (see page 16-17).

Referee #2 (Remarks for Author):

In their manuscript, Gambelli et al. define how increased expression of the alpha6 integrin subunit (ITGA6) induced by platinum treatment (PT) promotes chemoresistance, growth, and invasion of epithelial ovarian cancer (EOC). In addition, the Authors show how ITGA6 may represent a novel therapeutic target and biomarker of reduced survival of EOC patients. The data presented by Gambelli et al. are very interesting in principle, however several aspects need clarification or further experimental investigation.

We thank Referee #2 for the general appreciation of our work and for Her/His stimulating suggestions. We hope that in the present form She/He will find our work improved and acceptable for publication in EMM.

Major issues

1. Although the Authors show that GoH3 blocking antibody counteracts the effects of conditioned medium (CM) of PT-resistant (PT-res) EOC cells on PT-sensitive (PT-sen) EOC cells and mesothelial cells, this is not enough. The authors need to analyze whether ITGA6 plays its role as a cleaved and released protein in the CM or as a component of extracellular vesicles that can be isolated and characterized (e.g., see Hoshino et al. Nature 2015). Furthermore, does depletion of extracellular vesicles attenuate or abolish the biochemical (e.g., signaling via IGF1R) or biological (e.g., control of PT-sen spheroid size and adhesion to the mesothelium) properties of the CM of PT-res EOC cells?

We agree with the Referee on this point and thus verified if ITGA6 was secreted in exosomes. Our results showed that ITGA6 expression is mostly limited to the exosome fraction (Total Exosome Isolation Kit, Invitrogen) and that exosome-depleted conditioned medium does not express detectable levels of ITGA6 compared to *non*-depleted ones and that exosome isolated from PT-treated cells contains ITGA6 (new Figure 4B-C). We also showed that exosomes extracted from ITGA6 WT PT-Res cells better stimulate the formation and the growth of ovary-sphere (area) in PT-Sen cells plated on mesothelial cells, compared to exosomes from ITGA6 KO cells (new Figure 4D and Appendix S2C). Unfortunately, we could not compare the effects of CM containing or not exosomes since the latter resulted toxic when used on PT-sen cells, probably due to the buffer that is used during the extraction procedure (see images below, provided as an example).

Figure 2R. Pictures show the effects of serum free conditioned medium, serum free exosome-depleted conditioned medium (deprived CM) and serum free + exosomes, on the viability of TOV112D cells. Cells, incubated for 16 hours with serum free exosome-depleted conditioned medium (deprived CM) started to die and could not be used for further experiments (lower left panel).

2. In addition to proposing that ITGA6 acts via its soluble/exosomal form released in the CM, the Authors show that PT-res EOC cells adhere to the mesothelial cell monolayer via ITGA6. What is the mesothelial ligand to which ITGA6 located on the surface of PT-res EOC cells binds? Do mesothelial cells also express ITGA6? Do mesothelial cells secrete laminin apically and does this apical laminin act as a bridge between mesothelial cells and PT-res EOC cells? Or do PT-res EOC cells sneak between mesothelial cells and employ ITGA6 to adhere to the laminin basally laid down by mesothelial cells?

Although these questions are all important and interesting, we did not investigate further the mechanism by which epithelial ovarian cancer cells adhere to the mesothelium, as indicated by the EMBO Mol Med Editor.

3. Anti-ITGA6 (Bhol et al., J. Dent. Res., 2001, 80:1711-1715) or anti-ITGB4 (obligate partner of ITGA6; Chan et al., Invest. Ophthalmol. Vis. Sci. 1999, 40:2283-2290) (auto)-antibodies have been shown to result in the pathological

separation of epithelial cells from the laminin-containing basement membrane. This could be a significant limitation to the real therapeutic applicability of anti-ITGA6 antibodies in EOC patients. The Authors should investigate whether intraperitoneal treatment with the anti-ITGA6 GoH3 antibody may induce detachment of mesothelial cells or epithelial cells of the small and large intestine from the basement membrane to which they physiologically adhere.

To answer this question, immunocompetent C57/Black6MJ female mice were treated twice/week with P5G10 (30 mg/kg) and sacrificed 48 hours after the last injection. P5G10 and not GoH3 is in fact the anti-ITGA6 blocking Ab we used for in vivo experiments, as stated in the previous version of the manuscript. Mice treated with P5G10 or control IgG, were examined macro- and microscopically. No signs of sufferance (e.g. food intake, body weight loss, movements impairments, ascites formation) were observed. When the detachment of mesothelial cells or epithelial cells of the small and large intestine from the basement membrane was examined by an expert pathologist, we did not observe significant alterations neither in mice treated with the ITGA6-Ab or in control mice. In general, no detachment was observed between basal membrane and epithelial cells in any of the mice and sporadic small detachments were only observed in some peritoneal areas (see new figure Appendix S5B and C).

4. How ITGA6 expression level affects the cells sensitivity to cisplatin was only tested in TOV-112D cells. Are these results also observed in the OVSAHO cell line?

We tested the IC50 of CDDP in PT-Sen, PT-Res and PT-Res ITGA6KO OVSAHO cells, plated on plastic or laminin, and observed that ITGA6 is necessary for adhesion-dependent resistance to PT (See new Appendix Figure S1C). We also confirmed these results in vivo using a PT-resistant ovarian cancer PDX (see new figure 6H-M and Expanded View 4E-H)

5. In Figure S1A, cell adhesion was tested in the pool of TOV-112D and OVSAHO cells. Authors should test the adhesion of the different clones of each cell line further used in the manuscript.

5. As suggested, we have added the missing data (see new figure Expanded view 1D).

6. The manuscript indeed contains many findings that should be organized and linked together according to a more interconnected and sequential logic. If secreted ITGA6, likely in exosomes, plays the predominant role from the pathogenetic point of view, the whole text section concerning ITGA6-mediated adhesion aspects of EOC cells could be downsized and the corresponding experimental data transferred to new Supplementary Figures.

We thank the Referee for this suggestion and we reorganized the manuscript focusing especially on the role of secreted ITGA6.

Minor issues

1. The abstract should also describe the in vitro work done by the Authors.

1. As requested, we have now better specified what was done in the in vitro experiments.

2. Numbering pages and lines in the manuscript would facilitate references in the Reviewer's comments. I will therefore refer to the page numbers as they appear in the PDF file.

2. We numbered the pages in the new version and apologize for this inaccuracy.

3. Page 5. The sentence is too long and should be simplified and rephased. "These data were in line with the notion that $\alpha\beta 1$ and $\alpha\beta 4$ heterodimers mediate cell adhesion to laminins (Hynes, 2002) and were then confirmed in all PT-res clones derived from pools (Sonogo et al, 2020) (Figure 1A-B and Figure S1C) and in EOC cells collected from patients at disease recurrence, which displayed higher ITGA6 protein expression compared to cells derived from patients with primary disease (Figure 1C, and Table S1A), suggesting that upregulation of ITGA6 represents a quite common modification associated with acquisition of EOC PT-recurrence/resistance".

3. We have rephrased, as suggested.

4. Page 5. The sentence referred in the previous point (3) indicates that "upregulation of ITGA6 represents a quite common modification associated with acquisition of EOC PT-recurrence/resistance". However, it does not seem that is quite common, as the significant could come from the three patients with the higher expression, rather from the main distribution of the cohort.

4. As suggested, we have better specified this point in the new version of the manuscript.

5. Page 5. "Intriguingly, when we sorted PT-sen TOV112D cell subpopulations expressing either high or low levels of ITGA6 (ITGA6HIGH/ITGA6LOW), we observed that ITGA6HIGH cells were as resistant to cis-platin (CDDP) as the PT-res cell clones (Figure 1D-F)." At the end of this phrase, only Figure 1D-F are referred, when it should be Figure 1D-G.

5. Figure 1G refers to Pt-Res cells and not the sorted ITGA6^{HIGH}/ITGA6^{LOW} populations.

6. Page 6. "Overexpression of SP1 or c-Myc, and their pharmacological inhibition with Mithramycin (MTA, SP1 inhibitor) or 10058-F4 (c-Myc inhibitor), demonstrated that basal ITGA6 expression was stimulated by SP1 and inhibited by c-Myc expression/activity in PT-sen cells (Figure 2C-F and S2C-F)." Figure 2C-F and S2C-F showed the inhibition of SP1 and MYC or only the overexpression of SP1, while the text also mentions the importance of MYC overexpression.

6. We rephrased since, as noted, Myc overexpression was only used in Figure EV2B.

7. Page 6-7 and Figure S2K. "Among nine epigenetic modulator drugs (EMDs) tested, only Panobinostat, a pan-inhibitor of Histone De-Acetylases (HDACs), consistently increased ITGA6 expression in PT-sen models and had no effect on PT-res cells (Figure S2K-M)." The main text indicate that Panobinostat had no effect on PT-res cells, yet it appears there is a decrease compared to the control.

7. We thank the Referee for highlighting this point. Although in the original Figure S2K (now Expanded view 2L) it seems that Panobinostat decreases ITGA6 protein expression compared to the control, other experiments (e.g. Expanded view 2N) did not confirm this observation in PT-res cells.

8. Page 9. "To test whether secreted ITGA6 was biologically active, PT-sen cells were incubated with the CM of WT or ITGA6KO PT-res cells for 16 hours prior to challenge their ability to adhere on the mesothelial monolayer, form spheroids and activate specific intracellular pathways (Figure 4B)". However, the adhesion to mesothelial cells was not tested or not reported in Figure 4B.

8. We partially disagree with the Reviewer here, since the assay of old Figure 4B (new Figure 4E) tested the ability to form spheres on mesothelial cells (lower images of Figure 4E). We have better specified this point in the text and in figure legend.

9. Page 9 and Figure S4C. "Accordingly, we observed that CDDP treatment induced the secretion of ITGA6 and CD63 (extracellular vesicles marker) in a time-dependent manner, peaking at 24 hours of CDDP treatment, in both PT-sen and PT-res cells (Figure 4A and S4C)." In the main text, it is indicated that ITGA6 expression peaked at 24h post-cisplatin, but it is only tested up to 16h.

9. Yes, the 24hours time-point refers to the PT-Res cells. PT-sen cells were stimulated up to 16 hours. We better specified this difference in the new text.

10. Page 10 and Figure 4E. "Among different tested signaling pathways, activating phosphorylations of the beta subunit of the Insulin-like Growth Factors 1 Receptor (IGF1R β) and Src tyrosine kinases were specifically enriched in cells stimulated with CM from WT, but not ITGA6KO, PT-res cell." The differences in the expression of these proteins between the two conditioned media are not that obvious, at least for Src, as indicated in the mentioned sentence.

10. Here, we observed the activation of IGF1R at 5' stimulation with both CM but this activation was stronger (about 2 folds) when CM from WT cells was used. Src activation was also higher and more sustained when cells were stimulated with CM from WT, compared to CM from KO cells. We better specified this point in the text.

11. Pages 12-13. The anti-ITGA6 blocking antibody was given twice in Figure 6G, but three times in the PDX model of Figure 6K. Based on what these treatments planning were decided? Why are they different? May Authors explain the rationale?

11. Based on pilot experiments, we decided to inject mice every 10 days and sacrificed them 10 days from the last treatment. We reasoned that PDX OV215.3 grew slower than TOV112D resistant cells and, thus, they should grow in vivo for at least 30 days.

Therefore, while PDX OV215.3 mice were treated at the time of injection, at day 10 and day 20, and then sacrificed at day 30, TOV112D-injected mice were treated at the time of injection and day10, then sacrificed at day 20.

12. Page 13 and Figure 7C-D. "Importantly, high LAMA5 and/or IGF2 predicted shorter PFS and OS also in the whole EOC patients' population (Figure 7C and D)". The figures do not correspond to the text and there is no Figure 7D.

12. We apologize for this inaccuracy. We have now corrected the text.

13. Please check the consistence in the numbering of PT-res cell clones. It appears that in Figure 1A-B and FigureS1C different PT-res clones were employed.

13. We have re-checked the number of clones and confirm that we used PT-res clones #2 and #3 for TOV112D cells and #39 and #56 for OVSAHO cells, in all experiments.

14. Figure 2J and S2I showed Spearman and Pearson analyses, but in each report a different one, 2J shows the Spearman r and p-value, while S2I reports the data from Pearson.

14. Yes this is true but, in the text, we reported both Spearman and Pearson analyses results, for both panels. We have now resolved this discrepancy in the figure.

15. Figure 4E and 4F: both figures showed similar conditions, yet the result regarding Snail looks quite different at 15 and 30 minutes of conditioning. May Author check and comment?

15. In this case, we think that the results are quite consistent showing a slight decrease of Snail in cells incubated with CM from WT PT-res cells. The decrease seems more consistent in Figure 4F (old 4E) only because of the different blot exposure, but it was similarly present in Figure 4G (old 4F), where longer exposure time was used.

16. Figure 4F lacks a NC control.

16. We resolved this inaccuracy (see new Figure 4G)

17. Figure 5D, it should be clearly specified that difference is detected at 15 minutes.

17. We better specified the results in the text.

18. Figure 7C and 7D. The figures do not correspond to the text (page 13) and there is no Figure 7D.

18. We apologize for this inaccuracy. We have now corrected the text.

19. Figure S1B, C and D seem to refer to completely different PT-res clones: TOV-112D #1, #2, #3; OVSAHO #1, #2, #39, #56. Is the numbering correct?

19. In Expanded view 1B (old Figure 1B) are reported the results obtained from PT-res pools (#1 and #2 for both TOV112D and OVSAHO). From these pools we retrieved the clones used in the manuscript that are TOV112D clones #2 and #3 and OVSAHO clones #39 and #56.

20. Figure S3D. The number of spheres is only reported for OVSAHO cells, but not for TOV-112D.

20. The number of spheres formed by TOV112D are visible from the representative picture, reported in Figure 3F, that clearly shows a reduction of sphere number/field in ITGA6KO cells. For this reason, we did not include a graph of sphere number. However, we reported the number of spheres formed by TOV112D PT-res cells, treated or not with Go3 Ab, in Appendix Figure S1F.

21. Figure S3C and F. The same axes scale should be used.

21. We do not think it is necessary to use the same scale, since in Figure S3C (now Appendix Figure S1D) we used OVSAHO cells (parental vs PT-res vs ITGA6KO), while in Figure S3F (now Appendix Figure S1G) we used sorted TOV112D cells, which form spheres of different shape and areas.

22. Figure S3 K and L. The only difference between these two graphs is the presence of ITGA6. The expression of ITGA6 should be tested in the same settings, as in Figure S3L.

22. The difference is that in Figure S3L we included the use of cycloheximide, CHX. In Figure S3K, ITGA6 expression was only used to confirm the KO of the protein.

23. Page 13. "(Figure 6H-J and S6D-E)". There is no figure S6E.

24. Figure S7G-H: these two graphs are not described in the figure legend and neither mention in the main body of the paper.

25. Table S1 and S2 do not indicate the number of patients, which should be around 30.

23-25. We apologize for these inaccuracies. We have now corrected the text, accordingly.

26. It would be interesting to include a third Supplementary Table like Table S2 but reporting the kinases analyzed in Figure S3N.

26. As suggested, we included the Appendix Table S2, reporting the kinases analyzed in Appendix Figure S1I (old Figure S3N).

Referee #3 (Comments on Novelty/Model System for Author):

Technical quality and model systems are adequate. Novelty is an issue, since the link between ITGA6 and chemoresistance in ovarian cancer has been reported already (PMID 34600083, 30930993), including a causal link between ITGA6 and platinum resistance (PMID 28131812).

While the study is relevant for a better understanding of the biology behind ovarian cancer biology and drug responsiveness, the clinical implications are limited until the ability of ITGA6-targeted therapy to improve chemoresponse is shown in clinically relevant settings.

Referee #3 (Remarks for Author):

Gambelli et al. report that platinum-based chemotherapy induces the upregulation of ITGA6 in OC cells which, in turn, promotes chemoresistance, acquisition of stemness-related features, and mesothelial invasion. Moreover, they show that ITGA6 is required to sustain an EMT program that apparently relies on Lyn-dependent stabilization of Snail.

This is a well-conducted study, where the experimental strategy is clearly delineated and follows a logical sequence.

The manuscript is well written and reader-friendly.

Overall, this manuscript describes the role of ITGA6 as a hub that in OC orchestrates response to chemotherapy, cancer stemness and EMT, and peritoneal spreading. This makes it in principle an interesting and relevant study. However, the current version raises several major concerns that should be addressed to improve the quality of the manuscript, to solve some conceptual issues, and to support some of the authors' interpretations and conclusions.

We thank Referee #3 for the general appreciation of our work and for Her/His comments and suggestions that helped us to improve the manuscript, resolving the major concerns and some conceptual issues present in the submitted draft. We hope that She/He will find this improved version of the manuscript now acceptable for publication in EMM.

Major points:

1. The novelty of the study is somehow limited. Other reports have previously linked ITGA6 to chemoresistance (reviewed in PMID 37444576), including OC (PMID 34600083, 30930993). Villegas-Pineda et al. have also causally implicated ITGA6 in the platinum resistance of OC cells (PMID 28131812). Moreover, the role of SP1 in the transcriptional regulation of ITGA6 (PMID 34199886, 17652716), the crosstalk between ITGA6 and Lyn is also known (PMID 32219444), and the role of ITGA6 in cancer stemness and EMT (reviewed in PMID 37444576) have been described already. Of course, the authors here provide a unifying view of all these findings in the context of OC chemoresistance and malignancy, and this has its own value. Yet, I'm not sure that it is sufficient for a journal like EMBO Mol Med.

1, Regarding the novelty issue raised, we were well aware of and, in fact, cited all mentioned publications (and others). However, we kindly disagree with the Referee interpretation.

- Those studies did not investigate if and how ITGA6 participates in determining the phenotypes of PT-resistance. Villegas-Pineda et al (PMID 28131812) only treated SKOV-3 cells (not the resistant clones) with morpholino oligos against ITGA6 showing that it improves the efficacy of carboplatin when administered at low doses. This marginal observation goes in the direction of our work but it does not demonstrate any causal link between ITGA6 expression and PT resistance in EOC.
- The mentioned manuscript did not explain how SP1 regulates ITGA6 expression and how this regulation is modified by PT treatment. They also did not show the role of HDACs and c-Myc in the regulation of ITGA6 transcription, nor they used ovarian cancer models.
- There is only one manuscript, that was already cited (PMID:32219444), that links Lyn phosphorylation to ITGA6 expression in murine ALL cells. However, as the Referee will certainly acknowledge, ALL is a completely different disease compared to EOC. Indeed, the role of Lyn and Src in the mouse ALL model was also completely different than the one we propose here, since in their model Kim and collaborators showed that Src signaling is necessary to regulate ITGA6 expression and not cell survival. Thus, although these results somehow confirm our observations, we think that they do not absolutely impact on the novelty of our findings.

We truly believe that the fact that other articles mention ITGA6 and drug resistance does not diminish the novelty of our manuscript, which investigates the key role of ITGA6 at mechanistic and translational level, as never done before.

If just mentioning 2 keywords would diminish the novelty of any finding, virtually no manuscript could be ever published in the field of cancer in a journal like EMM.

2. How does CDDP enhance SP1 recruitment to ITGA6 promoter in Pt-sen cells? This would be a novel mechanistic insight.

2. As shown in Figure 2 and Expanded View 2, we have demonstrated that, in PT-Sen cells, platinum treatment induce the displacement of Myc (likely in complex with HDAC1) from ITGA6 promoter. This, in turn, allows SP1 to bind and

positively regulate ITGA6 transcription. We have now better described these results in the text.

3. The ITGA6 protein migrates as a doublet in most immunoblots of cell lysates (not in CM). What do the two bands correspond to? This is not trivial, since in some experiments the two bands follow a different fate (which in some cases contradicts the authors' conclusion). For example, PT-sen ES-2 cells show high levels of the lower band, and in their PT-res counterpart this band is downregulated while the upper band is upregulated. The opposite occurs in TOV-21G cells (Fig. 1B). Adhesion of PT-res TOV-112D to LM10 caused reduction of the upper band and increase of the lower band (Fig. 3I). Always in TOV-112D cells, CHX treatment of PT-res cells is accompanied by no change (or even an increase) in the lower band and only the upper band decreases (Fig. S1E).

3. Although we have not shown any of the data, we have deeply investigated this point during our work. After many different approaches, we came to the conclusion that the upper band represents a glycosylated form ITGA6, as explained herein.

Integrins can be dynamically glycosylated (PMID: 30064662) and we have collected multiple evidences showing that ITGA6 undergoes N-Glycosylation, not least because the upper band disappears when cells are treated with Tunicamycin, which blocks N-linked glycosylation.

However, we found that this modification is quite dynamic, it is more evident in some of the cells and clones that we have used compared to others (although present in all tested cells) and more present on the ITGA6 isoform B. Yet, we still have not fully understood which is the role, if any, of ITGA6 N- Glycosylation in the acquisition of a PT-resistant phenotype and how N-Glycosylation is regulated in resistant EOC cells. Thus, we have decided to keep these experiments apart and work on this topic possibly for another manuscript.

Figure for reviewers removed

4. Page 5. Referring to three different isogenic PT-sen vs PT-res cell lines, the authors claim that "all PT-res counterparts increased their ability to adhere to laminins (Figure S1A)...". In fact, Fig. S1A shows only two cell lines, and in one of them, TOV-112D, the effect is quite minor and with high error bars.

4. We thank the Referee for highlighting this possible discrepancy. We have now provided more compelling data demonstrating that OVSAHO and TOV-112D PT-res clones better adhere to Laminins compared to the PT-sen counterparts (see new figure Expanded view 1D).

5. Fig. 1D. The ITGA6-based FACS profile of PT-sen cells depicts two sharply distinct cell populations within an established cell line. Is that really the case or it has to do with the gating conditions or other technicalities? Is such a separation seen also in other OC cell lines? This is quite relevant and should be at least discussed.

5. We actually did not use any technicalities to get this graphical separation of the two populations that was provided by our Cytometry facility. They did for us the sorting experiments and they used the BD Bioscience FACS Aria instrument, which is different from the FACS LS Fortessa (BD Bioscience) used for the analysis of Integrin expression provided in Figure 1 and Expanded view 1. We now provide a better description of the methods underlying the sorting experiments (page 22).

6. Page 6. "Moreover, [in PT-res cells] SP1 binding to ITGA6 promoter was stably high and slightly reduced by CDDP treatment (Figure 2G) ...". This does not appear to be the case: CDDP reduces it quite remarkably (almost 3-fold), which is quite puzzling. This also raises another doubt: how comes that such a reduction of promoter-bound SP1 does not lead to a downregulation of ITGA6 expression, and we instead see an upregulation?

6. We thank the Reviewer for highlighting this point. Following Her/His suggestion, we asked a different post doc to repeat the experiments one more time in duplicate and then analyzed all collected data together. These even more robust results (new Figure 2G) confirmed a slight, although significant, decrease of SP1 binding to the ITGA6 promoter in PT-res cells upon CDDP treatment. These results are in complete agreement with the observation that PT-treatment did not

stimulate the transcription of ITGA6 in PT-resistant cells and that the fold enrichment of SP1 in CDDP treated PT-res cells is still definitely higher than the one observed in PT-Sen cells not treated with PT. Thus, we believe that these ChIP data fully support a constitutive, and not PT-induced, transcription of ITGA6 in PT-Res cells.

7. Page 8. "Using the ovarysphere formation assay ..., we observed that PT-res cells formed more and/or bigger spheres than PT-sen ones". This does not appear to be the case: Fig. 3F shows that ovarysphere area is smaller in untreated, WT PT-res cells than PT-sen cells. The opposite applies only to CDDP-treated cells.

7. We thank the Reviewer for pointing out this inaccuracy. We completely agree with Her/Him and we used the expression "and/or" to clarify that in some case they could be bigger and in others could be more (see figure 3F images and graph). However, to avoid confusion we rephrased the sentence (page 8)

8. Page 9. "... upon adhesion to LM10, the phosphorylation at Y508 (inhibitory) was sharply increased in PT-res ITGA6KO cells..." and "... ITGA6 engagement to LM10 in EOC cells increases Snail protein stability...". In fact, the authors cannot formally ascribe these effects to the adhesion of cells to LM10, since their experiments in Figs. 3J and 3K lack the "time 0" condition (i.e., no LM10). Thus, the effect of adhesion to LM10 on these signaling events cannot be determined. In the same context, it is unclear why the authors data related to SRC if the relevant kinase is Lyn. Also, Fig. 3K should relate the regulation of Snail to phosphorylation status of Lyn, not SRC. This is crucial, because without it the regulation of Snail cannot be causally linked to Lyn activity. Finally, SRC data in Figs. 3J and 3K are discrepant: adhesion of PT-res to laminin shows no pSRC in 3J and detectable pSRC (reduced by Saracatinib) in 3K.

8. As requested, we checked the Lyn signaling in PT-Res WT and ITGA6KO cells, upon adhesion compared to *non* adherent cells. We now show that in WT, but not ITGA6KO cells, Lyn activating phosphorylation increased, confirming previous results. Regarding SRC activity, we did not refer to c-SRC activation but, generally, to SRC activity, since the so-called anti pSRC Tyr416 antibody cannot distinguish among the different SRC family members (see new Expanded view 3J).

9. Page 9 and Fig. 4A. What is the rationale for including CD63 here? It is mentioned as an extracellular vesicle marker, yet no extracellular vesicle purification is performed here, so ITGA6 secretion in the CM remains unrelated to extracellular vesicles.

9. We used CD63 expression as positive control of protein secretion, since it is a well-known secreted protein and it has been already reported that its secretion is induced by CDDP treatment.

Then, as also requested by the other Referees, we have also verified if ITGA6 was secreted in exosome. Our results showed that ITGA6 expression is mostly limited to the exosome fraction (Total Exosome Isolation Kit, Invitrogen) and that exosome-depleted conditioned medium does not express detectable levels of ITGA6 compared to *non*-depleted ones and that exosome isolated from PT-treated cells contains ITGA6 (new Figure 4B-C). We also showed that exosomes extracted from ITGA6 WT PT-Res cells better stimulate the formation and the growth of ovary-sphere (area) in PT-Sen cells plated on mesothelial cells, compared to exosomes from ITGA6 KO cells (new Figure 4D and Appendix S2C). Unfortunately, we could not compare the effects of CM containing or not exosomes since the latter resulted toxic when used on PT-sen cells, probably due to the buffer that is used during the extraction procedure (see images below, provided as an example).

Figure 2R. Pictures show the effects of serum free conditioned medium, serum free exosome-depleted conditioned medium (deprived CM) and serum free + exosomes, on the viability of TOV112D cells. Cells, incubated for 16 hours with serum free exosome-depleted conditioned medium (deprived CM) started to die and could not be used for further experiments (lower left panel).

10. Pages 9-10 and Figs. 4C, D. The reduction in sphere formation when using the CM from ITGA6-KO cells as compared to WT cells is ascribed to the lack of secreted ITGA6. This conclusion cannot be formally drawn, since it could depend on an indirect effect of ablating ITGA6 on the secretome of PT-res cells (e.g., downregulation of other secreted factors that promote adhesion to mesothelium or the induction of factors that counteract it). Depleting ITGA6 from the CM of WT cells or using the blocking antibody shown in the following experiments would enable to determine the direct effect of secreted ITGA6.

10. We agree with the Referee on this point. However, since we have already proved, as She/He also has acknowledged, that anti ITGA6 blocking antibody prevents ITGA6-IGF1R binding and IGFR phosphorylation (new Figure 4G and J), that phosphorylation of IGF1R in mesothelial cells (new figure 5D) and also the adhesion of PT-sen cells on mesothelial cells stimulated with ITGA6 containing ascites (new Figure 5G), we have decided to bypass the formal demonstration and focus our additional work on other more substantial points that were raised.

11. Fig. 4E. The correct interpretation of these data appears to be that ITGA6-containing CM sustains the signaling (pSRC and Snail), while the authors claim a specific enrichment in cells treated with WT CM.

11. We thank the Reviewer for the correct interpretation of our data and we have added this possible interpretation in the discussion section (page 10)

12. What is the ITGA6 function blocked by GoH3? If it is binding to ECM (laminin), how comes that the antibody blocks the signaling induced by soluble ITGA6 in the CM? For example, does the LM-binding site overlap with the binding site for IGF1R? This should at least be discussed.

12. One possible explanation is that the blocking Ab interfered with the activation of IGF1R-Src pathway, preventing ITGA6/IGF1R binding (Figure 4J) and reducing IGF1R and Src family member activation (Figure 4G and 5D). As suggested, we have better discussed this point in the new version of the manuscript (page 18).

13. Page 13 and Fig. 7/ Fig S7. The authors' interpretation of these data is somehow inconsistent. First, the association of high LAMA5 with shorter overall survival (Fig. 7D, once correctly labeled) can be considered a trend at maximum (log-rank $p=0.057$) and the Results should reflect this. Then, data in Fig. S7D are claimed to establish a correlation between high LAMA5 and shorter PFS, not considering a log-rank $p=0.093$. On the contrary, the association of high IGFBP6 with shorter PFS (log-rank $p=0.033$) "did not have any prognostic value".

13. We kindly disagree on this point, since we have a different interpretation of the results.

The prognostic effect on any biomarker in EOC is much better appreciable looking at the Progression Free Survival (PFS) rather than to the Overall Survival (OS), since the latter is largely influenced by multiple lines of treatments that could or could not (usually "not" in resistant tumors) include PT based therapy, and additional targeted therapies, such as Bevacizumab or PARPi.

To interpret this type of data, we always consider whether the biomarker in object has similar prognostic effects on PFS and OS.

In this is the case while IGFBP6 has no prognostic value when PSF is considered (HR= 1.03 $p=0.6$) and marginal prognostic value in OS (HR 1.1). Conversely, LAMA5 has a strong prognostic value in PFS (HR 1.29 $p=8.2 \times 10^{-5}$) and marginal prognostic value in OS (HR 1.1). Therefore, we considered that IGFBP did not have any prognostic value.

14. Page 14 (Discussion) and Fig. 6K. "...targeting ITGA6 in combination with standard PT-based chemotherapy could be beneficial to elicit a better response to PT...". Why was this not tested experimentally? The authors appear well positioned to assess the effect of the GoH3 + CDDP combination, which would have relevant translational implications and would probably change the impact of the manuscript.

We really thank the Reviewer for raising this important point. Following Her/His suggestion, we have tested this combination experimentally, although particularly long and laborious.

As shown in the new Figures 6 and Expanded view 4, we employed a PDX model (OV218.3) established in our lab from a EOC patient, who had relapsed during the maintenance treatment with the PARP inhibitor Olaparib after PT-based chemotherapy. NSG mice injected with PDX OV218.3 were treated with CBDCA and/or ITGA6-Ab i.p. for two weeks, starting 21 days after initial injection. All untreated mice formed ascites (median 2 ml; range 1.5-4 ml/mouse), colonized all abdominal and pelvic organs and metastasized to the lung. Only the combination of CBDCA+ITGA6-Ab reached the statistical significance in reducing the volume (median 0.8 ml; range 0.2-1 ml/mice, $p=0.01$) and the number of tumor spheroids of the ascites.

Pathological analyses demonstrated that each treatment alone reduced the number of metastasis/mice but that CBDCA+ITGA6-Ab combination controlled PDX spreading significantly better than the single treatments.

Combination treatment also significantly improved the expression of the DNA-damage marker γ -H2AX and of IGFBP6.

Further, we observed by immunofluorescence analyses that Snail was upregulated by CBDCA treatment, as we have recently reported (Sonogo et al, 2019), but co-administration of the anti-ITGA6-Ab completely prevented this increase. Single ITGA6-Ab treatment also reduced basal expression of Snail, again confirming in vitro data. Finally, by co-

staining peritoneal metastases with anti-ITGA6 and anti-phosphoIGF1R β antibodies, we observed clear clusters of ITGA6 co-localization with active IGF1R β , especially at the site of tumor-mesothelium contacts. These co-localizations were further increased upon PT-treatment and almost completely abolished when ITGA6-Ab was used. Overall, these data confirm and reinforce our previous observation that the combination of anti-ITGA6-Ab + CDDP has great translational implications.

15. Page 15 (Discussion). "However, these studies did not investigate if and how ITGA6 participates in determining the phenotypes of PT-resistance". Actually, Villegas-Pineda et al (PMID 28131812) showed that silencing ITGA6 sensitizes to platinum-based chemotherapy.

As discussed above, Villegas-Pineda et al. did not investigate the actual role of ITGA6 in the insurgence of PT-Resistance in EOC. They just tested the effect of a single morpholino oligo on a single PT-sensitive cell line in response to two doses of CBDCA treatment, obtaining even contrasting results. In fact, they observed a slight decrease of cell proliferation using 0.2 mM of CBDCA plus ITGA6 morpholino, compared to control cells, and no effect using 0.5 mM of CBDCA. This single experiment in our opinion is not the same of investigate if and how ITGA6 participates in determining the phenotypes of PT-resistance in EOC, as we did here with multiple models and approaches. In any case, these data are in line with our observations since they showed that ITGA6 expression is limited to few positive cells in PT-sen EOC (demonstrated by IF analyses in Villegas-Pineda et al.) and that ITGA6 silencing has variable effects on the response to PT-treatment when cells are plated on plastic.

Minor points:

1. Fig. 1C. Vinculin doesn't really serve as a loading control in that blot (as claimed in the figure legend. It's highly variable among different samples.

1. We agree with the Reviewer but, in our experience, it is difficult to identify a uniform loading control among primary samples. In this case we refer to Vinculin as loading control since the left graph in figure 1C reports the expression of ITGA6 normalized on the one of Vinculin.

2. Fig. 6 G-M. Does GoH3 cross-react with murine ITGA6? This is important to assess if the antibody can also affect the host response to tumor.

2. Yes, GoH3 it is reported that it should cross react with murine Itga6, but as specified in the text, for in vivo experiments we used P5G10 that is described to react to human ITGA6 and we did not check this point in detail.

3. Fig. S6E is referenced to in the Results (page 13), but is not present in Fig. S6.

4. Fig. 7. Panels C and D are referenced to in the Results (page 13) but are not correctly labeled in the figure. Current panel C should be changed to E.

3-4. We are sorry for these inaccuracies. We have now corrected the text.

5. Page 13. PDX #OV215.3 is claimed to derive from "patient #8 in Table S1". Unfortunately, Table S1 does not list the patients, and the patients' history in the scheme at the end Materials and Methods does not include Patient #8. The patients' list should indeed be included, also to assess the data of Fig. 1C.

5. We have now better described patients' data in Table S1. For PDX OV215.3 and PDX OV218.3 clinical history of EOC patients are now reported in Appendix Figure S5H and Expanded view 4D, respectively.

6. No information is provided on the origin, isolation and culture conditions of mesothelial cells. They are only described as "derived from pleural lavage".

We now provide more information on Mesothelial cells in the methods section, as requested.

15th Mar 2024

Dear Dr. Baldassarre,

Thank you for submitting your revised manuscript. We have now received the feedback from the three referees who who re-reviewed your manuscript. As you will see below, they are satisfied with the revisions, and I will therefore be able to accept your manuscript once the following editorial points will be addressed:

1/ Please address the minor comments from referee #1.

2/ Manuscript text:

- Please remove the blue font, and only keep in track changes any new modification.
- The following author's email address bounced: Andrea Vecchione.
- Please remove the "Highlights" on p.2.
- Please remove "Data not shown" (p.20): As per our guidelines, on "Unpublished Data" the journal does not permit citation of "Data not shown". All data referred to in the paper should be displayed in the main or Expanded View figures.
- Materials and Methods:
 - o Please check the text for spelling and grammar.
 - o Cells: please indicate whether the cells were tested for mycoplasma contamination.
 - o Antibodies: please make sure all concentrations/dilutions are provided.
 - o Mice: please indicate the housing and husbandry conditions of the mice.
 - o Human samples: please include a statement that the experiments conformed to the principles set out in the WMA Declaration of Helsinki and the Department of Health Services Belmont Report.
 - o Statistics: please include a statement on sample size, blinding and inclusion/exclusion criteria.
- The clinical timeline history on p. 30 should be made "Box 1".
- Data availability: This section is intended for providing the links to primary datasets generated in the study. Therefore, please remove the current text and list the accession numbers and database to primary datasets. In case you have no data that requires deposition in a public database, state "This study includes no data deposited in external repositories." Please fill in the checklist accordingly.
- Acknowledgements: The information provided in this section should match the information provided in the submission system (currently, CRO-Aviano CRO-Aviano 5% grants are missing in the submission system).
- Author contributions: CRediT has replaced the traditional author contributions section because it offers a systematic machine-readable author contributions format that allows for more effective research assessment. Please remove the Authors Contributions from the manuscript and use the free text boxes beneath each contributing author's name in our system to add specific details on the author's contribution. More information is available in our guide to authors.
- Please replace "Declaration of interests" by "Disclosure statement and competing interests". We updated our journal's competing interests policy and request authors to consider both actual and perceived competing interests (<https://www.embopress.org/competing-interests>).
- Please correct the nomenclature to "Figure legends" and "Expanded View Legends" as headings, and "Figure EV1" etc. as names for expanded view figures.

3/ Figures and Appendix:

- Appendix: please add page numbers, including in the table of content.
- Figure re-use is permitted but should be indicated in the figure legends (i.e. Figure 2D and Appendix Figure 2A).
- Please address the queries from our data editors in the figure legends:
 1. Please note that a separate 'Data Information' section is required in the legends of figures 1c, f-g; 2b-c, e-g, k-l; 3a-d, f-g; 5b, e, g; EV 1e, g-h; EV 2c, f, l; EV 3d-e; EV 4a-b, e-g.
 2. Please indicate the statistical test used for data analysis in the legends of figures 4k; 6b, d, k; EV 1a, d; EV 2a-b, h, m.
 3. Please note that information related to n is missing in the legends of figures 1c; f-g; 2b; 3a-d; 5e; 6b, d, k; EV 1a, d-e, g-h; EV 2a-b, h, m; EV 3d; EV 4a-b, e-f.
 4. Please note that the error bars are not defined in the legends of figures 1c; f-g; 2b-c, e, g, k-l; 3a-b, d; 5e; 6b, d, k; EV 1a, d-e, g-h; EV 2a-c, f, h, m; EV 3d, f; EV 4a-b, e-g.
 5. Please note that the scale bar needs to be defined for figure 4e.
 6. Please note that scale bar and its definition are missing for figures 3c; 6c, j.
 7. Please note that in figure 6m; EV 4h; the scale bar unit should be corrected from μM to μm (both in the figure legend and the figure file).

4/ Our source data coordinator contacted you to discuss which figure panels we would need source data for. Please provide the requested Source Data at next submission together with the completed checklist.

5/ Checklist: Please complete the section "Statistics/inclusion-exclusion criteria"

6/ I introduced minor modifications to your Paper Explained, please let me know if you agree with the following or amend as you see fit:

PROBLEM

Ovarian cancer remains a lethal disease mostly due to the development of chemo-resistant recurrence that metastasize in the abdomen and pelvis of patients. Understanding the molecular mechanisms behind resistance to chemotherapy and metastasis would help in identifying novel therapeutic approaches.

RESULTS

We generated several in vitro models of platinum-resistant (PT-res) ovarian cancer cells. PT-res cells displayed higher adhesion and invasive abilities compared to sensitive cells, which was mostly due to the higher expression of the adhesion molecule integrin $\alpha 6$ (ITGA6). In sensitive cells, ITGA6 expression was induced by PT treatment, while it was constitutively highly expressed in resistant cells. ITGA6 was involved in the formation of a pre-metastatic niche, acting both on bulk PT-sensitive cells and on mesothelial cells. Mechanistically, overexpressed ITGA6 in PT-res cells was secreted in the extracellular space and activated the IGF1R-Src pathway, leading to Snail protein stabilization in both PT-sensitive and mesothelial cells. Genetic or pharmacological inhibition of ITGA6 reduced PT-res metastatic abilities and increased their sensitivity to PT both in vitro and in several in vivo PDX models of ovarian cancer.

IMPACT

We propose that ITGA6 targeting by specific blocking antibody, could be exploited to improve the effects of PT-based chemotherapy in high risk ovarian cancer patients and prevent the appearance of recurrent/resistant diseases.

7/ For more information: thank you for providing useful links. Please only keep the URLs and remove the remaining text.

8/ Thank you for providing a nice synopsis picture. Please resize it to 550 px wide x 300-600 px high and make sure the text remains legible.

Please also provide a synopsis text that should include:

- a short stand first (maximum of 300 characters, including space)
- 2-5 one-sentences bullet points that summarizes the paper (maximum of 30 words / bullet point).

9/ As part of the EMBO Publications transparent editorial process initiative (see our Editorial at <http://embomolmed.embopress.org/content/2/9/329>), EMBO Molecular Medicine will publish online a Review Process File (RPF) to accompany accepted manuscripts.

This file will be published in conjunction with your paper and will include the anonymous referee reports, your point-by-point response and all pertinent correspondence relating to the manuscript. Let us know whether you agree with the publication of the RPF and as here, if you want to remove or not any figures from it prior to publication.

I look forward to receiving your revised manuscript.

Yours sincerely,

Lise Roth

***** Reviewer's comments *****

Referee #1 (Remarks for Author):

In general, I feel that the points raised have been adequately addressed.

In particular, it is easy to understand that ITGA6 is upregulated by CDDP, specifically that its secretion increases and affects the surrounding cells.

On the other hand, the appearance of IGFBP6 in the middle of the story is more difficult to understand for readers. In particular, the pattern in which IGFBP6 appears by depleting ITGA6 is difficult to understand: ITGA6 expresses many transcription factors such as Snail, while IGFBP6 expression is suppressed, although the factors involved are not well understood. Although it would be difficult to clarify this point, it would be easier for the reader to understand if this aspect was included in the schematic (Figure 5H). Can they somehow make one that includes IGFBP6 on the left side as well? It would be better to have a note in the figure.

We also believe that the fact that the text refers to ITGA6 but the schema refers to $\alpha 6$ makes it difficult to understand. It would be better to have a note in the figure.

A minor point is that although there are only 3 examples, I think the WB for the pre and post IDS samples should be included enough, even in the supplement.

Referee #2 (Remarks for Author):

The authors painstakingly worked and clarified conclusively all the points raised in the first stage of review.

Referee #3 (Remarks for Author):

The authors have addressed my comments.

The authors addressed the remaining formatting issues.

4th Apr 2024

Dear Dr. Baldassarre,

Thank you for submitting your revised files. I am pleased to inform you that your manuscript is accepted for publication and will shortly be sent to our publisher to be included in the next available issue of EMBO Molecular Medicine!

We note that a scale bar is still missing for Figure 4e. Please kindly address this in the manuscript file, and send me the corrected file (with accepted changes) via email, I'll upload it in the system.

Please also send me the original point-by-point rebuttal letter. If you would like to have figures removed, please let us know which one(s) and we will proceed (and replace by "figure for reviewer only"). You will be able to check and approve the RPF before final publication.

Your manuscript will then be processed for publication by EMBO Press. It will be copy edited and you will receive page proofs prior to publication. Please note that you will be contacted by Springer Nature Author Services to complete licensing and payment information.

With kind regards,

Lise Roth
